# Modelled dynamics of floating and grounded icebergs, with application to the Amundsen Sea

Yavor Kostov[1], Paul R. Holland[1], Kelly A. Hogan[1], James A. Smith[1], Nicolas C. Jourdain[2], Pierre Mathiot[2], Anna Olivé Abelló[2], Andrew H. Fleming[1], Andrew J. S. Meijers[1].

[1]British Antarctic Survey, Cambridge, UK

[2]Univ. Grenoble Alpes/CNRS/IRD/G-INP/INRAE, Institut des Géosciences de l'Environnement, Grenoble, France

*Correspondence to*: Yavor Kostov (yastov@bas.ac.uk)

**Abstract.** Icebergs that ground on the submarine Bear Ridge in the Amundsen Sea are known to block the drift of sea ice, playing a crucial role in maintaining shelf sea ocean conditions. This important iceberg—sea ice—ocean interaction is commonly observed around the Antarctic shelf seas. To better represent the drift, grounding, and ungrounding of icebergs in the vicinity of such seabed ridges, we introduce new dynamics into the iceberg component of the Nucleus for European Modelling of the Ocean (NEMO) ocean general circulation model. We implement a physically-motivated grounding scheme with parameter choices guided by observations from the Amundsen Sea. When the bergs are grounded, they now experience bottom sediment resistance, bedrock friction, and an iceberg acceleration due to gravity acting down topographic slopes. We also improve the representation of ocean turbulent drag and ocean pressure gradients, both for freely-floating and grounded icebergs, by incorporating the depth-dependence of these forces. We examine the diverse set of forces acting on simulated icebergs in the Amundsen Sea, and compare our simulations with iceberg observations near Bear Ridge. The new iceberg physics pave the way for future studies to explore the existence of possible feedback mechanisms between iceberg grounding, changing sea ice and ocean conditions, and iceberg calving from the ice shelves.

## 1 Introduction

Icebergs play a major role in the redistribution of freshwater in the polar oceans, contributing roughly half of the freshwater discharge from Antarctica and Greenland (Cendese and Straneo, 2022; Davison et al., 2023). Compared to basal melt of ice shelves, which causes localized freshening, icebergs tend to release meltwater farther offshore (Merino et al., 2016; Fox-Kemper et al, 2019). At the same time, there is diversity in the freshwater contribution among icebergs of different dimensions; large tabular bergs may become grounded in shallow shelf seas shortly after they calve and hence release melt water closer to the shore (Cenedese and Straneo, 2022; Olivé Abelló et al., 2025). Nevertheless, the largest icebergs have the longest life-span, allowing for a sustained contribution of meltwater (Cenedese and Straneo, 2022). Stern et al. (2016) and Fox-Kemper et al. (2019) point out that modelling the behaviour of the largest icebergs is particularly challenging, especially when they are represented as point particles because this ignores the impact of spatial variability in the ocean properties along the horizontal and vertical extent of the keels, as well as the interaction with bottom topography. The modelling approach of Stern et al. (2017) does capture the effect of the ocean and the atmosphere over the entire breadth and depth of large icebergs. Similarly, Martin and Adcroft (2010), as well as Merino et al. (2016), partially mitigate the issue of representing icebergs in NEMO as point particles by imposing that the latter respond to spatially averaged ambient properties, and that iceberg keels are aware of bottom topography.

Icebergs of all sizes are important not only for the thermodynamics of the polar oceans, but also for marine biology and ecology. For example, iceberg debris is a significant source of silica, whose supply to the ocean enhances the biological carbon pump by promoting the growth of diatom phytoplankton populations (Hawkings et al., 2017). Cefareli et al. (2016) identify diatom species attached to the submerged parts of "iceberg walls," providing further evidence for the biological relevance of icebergs. Tarling et al. (2024) argue that icebergs impact the distribution of phytoplankton and particulate matter

via changes in the salinity-driven stratification, while Lucas et al. (2024) suggest that iceberg melting allows for deep nutrient-
rich water to penetrate layers shallower than the stratification maximum. The latter promotes primary production and hence,
the carbon pump (Smith et al., 2007; Cenedese and Streaneo, 2022; Lucas et al., 2024). Last but not least, icebergs also
represent one of the largest sources of iron in the polar ocean (Duprat et al., 2016), while at the same time they are also very
heterogeneous and variable in their iron content (Hopwood et al., 2019).
As they drift, icebergs may run into topographic obstacles or plough into seafloor sediment, a phenomenon that has
been the subject of scientific inquiry as far back as the 19[th] century, when Charles Darwin published a short communication
on iceberg interaction with the ocean bottom (Darwin 1855). More recently, a number of studies on iceberg scouring have had
a focus on protecting underwater infrastructure (e.g., Chari 1979; Chari et al., 1980; and Lopez et al., 1981). Other literature
explores past iceberg scours in order to infer paleoclimatic changes in the North Atlantic (Hill and Condron, 2014) and the
Southern Ocean (Starr et al., 2021).
Topographic blocking by grounded icebergs can also significantly modify regional ocean conditions (Nakayama et
al., 2014; Stern et al., 2015; St-Laurent et al., 2015; St-Laurent et al., 2024), in shelf seas such as the Amundsen Sea
Embayment, where there are hundreds of icebergs present at any time and almost 90% of them are smaller than 2 km$^2$ (Mazur
et al., 2019). Bett et al. (2020) suggest the importance of iceberg grounding along Bear Ridge in the Amundsen Sea, Antarctica
(Figure 1; Mazur et al., 2019) to the oceanography of the region. In particular, Bett et al. (2020) point out that icebergs grounded
on Bear Ridge form a physical barrier to westward sea ice transport, and in their study, this is the dominant mechanism creating
a dipole in the sea-ice concentration (Figure 1). Icebergs blocking the supply of drifting sea ice creates the Amundsen Sea
Polynya to the west of Bear Ridge, which has a strong cooling effect on the water column and is important to the biological
productivity of the region (e.g., Arrigo et al., 2012; Person et al., 2019; St-Laurent et al, 2019). To the east of Bear Ridge,
icebergs blocking the export of sea ice suppresses polynya activity, reducing the local formation of Winter Water and thereby
promoting the intrusion of warmer modified Circumpolar Deep Water (mCDW; Bett et al., 2020). The drift and grounding of
icebergs in the Amundsen Sea is shown in the Supplementary Movie in Appendix A (Kostov et al., 2025a, an update of the
supplementary movie in Bett et al., 2020), which contains Copernicus Sentinel-1 SAR (synthetic aperture radar) images taken
over the time period 2017-2024. The impact on sea ice in the region is also clearly visible.
Crucially, these dynamics offer the possibility of a local positive feedback loop. The warm mCDW on the shelf in the
Amundsen Sea supports rapid melting of the surrounding ice shelves (Jacobs et al 2012; Dutrieux et al, 2014). Changes in the
supply of warm CDW has led to rapid thinning and acceleration of these ice shelves and their tributary ice streams (Mouginot
et al., 2016; Shepherd et al., 2019; Naughten et al., 2022). This has caused an overall increase in iceberg calving, punctuated
by several large ice shelf collapse events that released large numbers of icebergs (Miles et al., 2020; Joughin et al., 2021,
Davison et al., 2023). Changes in the supply of icebergs to Bear Ridge (and the thickness of those bergs) could affect the
characteristics of the grounded "iceberg wall," influencing sea ice blocking, hence CDW temperatures, and ultimately ice shelf
melting and calving (Bett et al., 2020). The sign and strength of this feedback mechanism and its future response to projected
climate change are unknown yet have important implications for the sea-level contribution from the West Antarctic Ice Sheet.
Therefore, the representation of iceberg grounding in models is essential not only for simulating the trends and variability of
the Amundsen Sea polynya but also for projecting major components of future sea-level rise.
These important roles of icebergs in the climate system necessitate their proper representation in ocean models (Fox-
Kemper et al, 2019). Some of the earliest efforts in modelling iceberg trajectories are attributed to Mountain (1980) and Smith
and Banke (1983), while many of the currently used iceberg models can be traced back to Bigg et al. (1997) and Gladstone et
al. (2001). Martin and Adcroft (2010) introduce icebergs in a fully coupled climate model, and Marsh et al. (2015) pioneer the
first version of the iceberg module in the state-of-the-art Nucleus for European Modelling of the Ocean (NEMO-v4.2.0) ocean
model (Madec, G. and NEMO-Team, 2016; Madec, G. and NEMO Systems Team, 2022). Bigg et al. (2018) use NEMO to
model iceberg hazard, while Merino et al. (2016) further develop the representation of icebergs in NEMO and, more

specifically, simulate the impact of depth-varying ocean properties on iceberg dynamics and thermodynamics. Since the developments of Merino et al. (2016), the keels of simulated icebergs interact with shallow bottom topography as an obstacle. However, icebergs in NEMO are unable to ground realistically and cannot remain trapped in the sediment. They stop and bounce off topographic obstacles instead. In turn, this has prevented the further development of algorithms simulating the blocking of sea-ice by grounded icebergs. The present work, which updates iceberg grounding in the model, is therefore a stepping-stone towards a future improvement of the interaction between icebergs and sea-ice in NEMO.

In this study, we introduce a new physically and observationally motivated iceberg grounding capability in NEMO. This is particularly designed and tuned for the Amundsen Sea, given the importance of icebergs grounding on Bear Ridge, but the underlying physics should be generally applicable to other areas. Sequences of satellite images from the Amundsen Sea region reveal complex patterns of iceberg motion near Bear Ridge with multiannual episodes of iceberg grounding (see the Supplementary Movie). Seafloor records of iceberg scours collected from multibeam bathymetry sounding in the vicinity of Bear Ridge also reflect the dynamics that arise from the combination of multiple forces acting on a grounded berg. A compilation of sediment density and shear strength data from representative sediment cores in the Amundsen Sea point to the important role that the so-called 'silt resistance' (Chari, 1979; hereafter referred to as 'sediment resistance') may play for the deceleration and arrest of grounded icebergs. These various observations, along with theoretical analysis of the underlying physical principles behind iceberg motion, inform our improvements to the NEMO iceberg module. This paper first develops the physical principles controlling the grounding of icebergs in NEMO, before applying these new physics in a regional model of the Amundsen Sea.

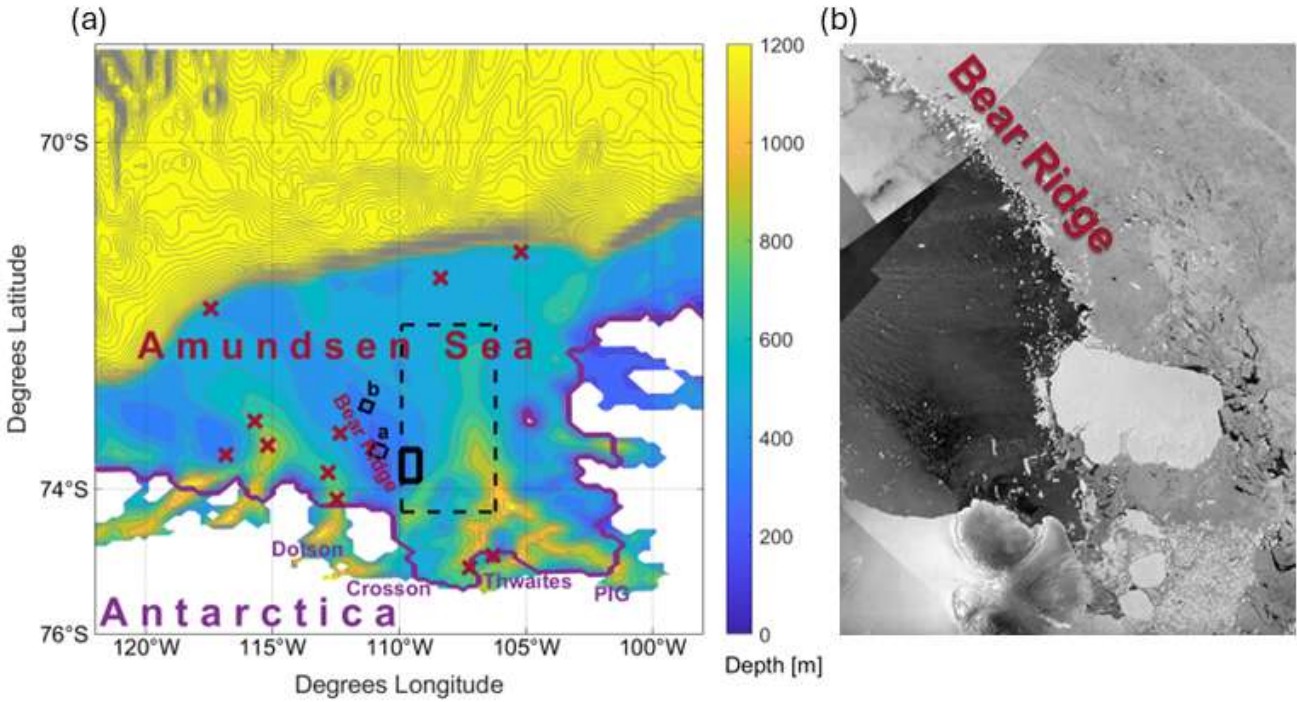

**Figure 1: (a) Amundsen Sea and Bear Ridge topography [m] based on the BedMachine Antarctica dataset (Morlighem et al., 2020) and re-gridded on the model configuration grid. Gray contours are spaced 80 m apart. Thin solid black contours delimit the bounds of boxes a and b in Figure 2, indicating the sites of observed iceberg scours in Section 2.1 with magenta labels matching boxes a and b in Figure 2. Dark red crosses mark the locations of sediment samples analysed in Section 2.2. The thick purple line shows the ice shelf edge, with major ice shelves labelled: Dotson, Crosson, Thwaites, and Pine Island Glacier (PIG). Thick solid (thin dashed) black lines show the boundaries of the boxes where icebergs are released in the SHORT (LONG) simulations. (b) Copernicus Sentinel-1 SAR (synthetic aperture radar) image from 12 February 2022 over Bear Ridge with white shades indicating the presence of icebergs and sea-ice.**

In Section 2 we present and discuss observations of iceberg scouring and sediment properties in the Amundsen Sea. Section 2 thus provides an overview of the relevant real-world conditions, which are essential to understand and model iceberg grounding as per Section 3. In Section 3, we describe the theoretical and modelling advances for iceberg dynamics and grounding and their implementation in NEMO. In Section 4 we test our modelling developments in an Amundsen Sea regional

configuration of NEMO and we describe the force balance of freely floating vs grounded icebergs. In Section 5 we summarise the results of our observationally-motivated numerical simulations with the updated NEMO model and then conclude.

## 2 Seafloor morphology and sediment properties in the Amundsen Sea

### 2.1 Observed iceberg scours

Iceberg scours (also called ploughmarks) are widespread in two Amundsen Sea regions and imply that thick icebergs can remain embedded in the bottom sediment for extended periods of time, a feature absent from previous versions of NEMO. Sediment scours also provide indirect evidence for the shape and size of the iceberg keels that interact with bottom topography, and this information is needed for the proper simulation of grounding in models. Here, we manually map more than 60 scours from existing multibeam-bathymetric data on the eastern flank and central part of Bear Ridge as a representative population of modern scours. We describe their morphology qualitatively with the aim of characterising modern iceberg grounding events, and then we provide metrics on their dimensions to calibrate the modelled scouring in Section 4.

First, scours are extensive across the middle—outer continental shelf where water depths shoal sufficiently to have allowed icebergs to touch the seabed (<~700 m), and second, they occur on the shallow banks between the landward-deepening glacial troughs on the continental shelf (Figure 1). Scour populations are dominated by single-keeled v- or u-shaped forms with relatively small widths and incision depths, although a few multi-keeled forms are present. A study of >10,800 scours in Pine Island Trough (East) by Wise et al. (2017) returned average single-keel scour widths and depths of 115 m and 2.8 m, respectively (ranges were 12-449 m and 0.1-22 m). Further west, on the banks around the Dotson-Getz glacial trough, typical scour widths and depths are 100-250 m and 4-10 m, respectively (Graham et al., 2009).

Most of the scours in deeper water (> 700 m) on the middle-outer shelf are likely to be relict forms incised during the last deglaciation, given the keel depths of modern iceberg populations and bathymetric impediments on the inner shelf (Dowdeswell & Bamber, 2007; Wise et al., 2017). In contrast, satellite imagery from the last decade shows clear accumulations of modern icebergs on the shallow banks to the west of calving glaciers in the eastern Amundsen Sea (see Supplementary Movie in Appendix A). After they are calved, the icebergs are moved westwards by prevailing winds and ocean currents and run aground wherever water depths shoal significantly. Bear Ridge, a N/NW-trending remnant of crystalline basement (Gohl et al., 2013) with water depths of 150-450 m, exhibits one of the most persistent collections of grounded icebergs today on its eastern margin, with icebergs of all sizes running aground there (Mazur et al., 2019).

Multibeam-bathymetric data from 17 research cruises with tracks across Bear Ridge were gridded with 30-m square grid cells and visualised in ArcGIS Pro 3.3.1 and QPS Fledermaus 8.6.1 for analysis. More than 60 iceberg scours were mapped, visualised with hillshading, and their cross-sectional widths and depths measured. Overall metrics are presented in Table 1.

Iceberg scours on Bear Ridge exhibit a variety of forms (Figure 2). Many incisions consist of single v- to u-shaped curvilinear depressions with shallow berms on either side; incision depths range from 2.5-15 m, widths are 90-360 m, and berm heights are typically 1-3 m (Figure 2c). The orientation of these scours is typically E-W or SE-NW (average orientation is 0.94°) and is thus consistent with iceberg transport from the Pine Island and Thwaites glacier fronts with the coastal current. However, the cross sections of some scours can change direction and vary along their length from a single v-shaped to wider u- or w-shaped, the latter indicating that the geometry of the iceberg base has varied as the iceberg was dragged through the sediment (consistent with either keel break-off or iceberg rotation). Tabular iceberg keels also have v-shaped protrusions along the rough surface of their bases. Occasionally, a large tabular iceberg can even plough multiple parallel scours. Some scours also terminate abruptly in rounded depressions, encircled by a shallow berm. Such iceberg grounding pits or iceberg plough ridges indicate where seafloor scouring terminated (e.g., Jakobsson et al., 2011). Most often there are no other incisions on the

seafloor that cross-cut the plough ridges, suggesting that the icebergs either lifted-off from the seafloor once enough melting had occurred or perhaps capsized (rotated vertically due to an unstable height:width ratio; e.g., Bass, 1980 and Ruffman, 2005) allowing them to float away freely without incising the seafloor further.

Detailed inspection of the scours reveals that some have either sinuous edges over length scales of a few hundred metres, variable depths along the centre of the scour, again at length scales of hundreds of metres, or exhibit both morphologies together. We interpret this variability as being formed by icebergs that have ploughed the seafloor and "wobbled", either from side to side or are close to flotation and have bobbed up and down on the seafloor under the influence of tides, currents and/or winds, or a combination of these factors (cf. Barnes and Lien, 1988; Lien et al., 1989). A particularly intriguing seafloor morphology of crossed u-shaped scours that appear to be linked together are observed on the southern part of Bear Ridge in water depths of ~275 m (Figure 2a). We suggest that these scours were formed by flat-bottomed icebergs that incised the seafloor as they were transported on to a shallower part of the ridge, where they halted. Then, under the influence of local currents, winds or tides, the bergs were dragged through the sediment in a perpendicular direction to their original pathway. Alternatively, two different icebergs may have ploughed scours perpendicular to each other, thus generating a crossed shape. Like the iceberg plough ridges, these scours have sharply defined and uninterrupted edges and berms suggesting that when the icebergs did eventually move off the seafloor it was due to melting or capsizing and no further marks on the seafloor were made.

**Table 1: Bear Ridge iceberg scour metrics**

| Av. depth (m) | Depth range (m) | Av. width (m) | Width range (m) | Av. orientation |
|---|---|---|---|---|
| 7 | 2.5-14 | 189 | 90-357 | 0.94° |

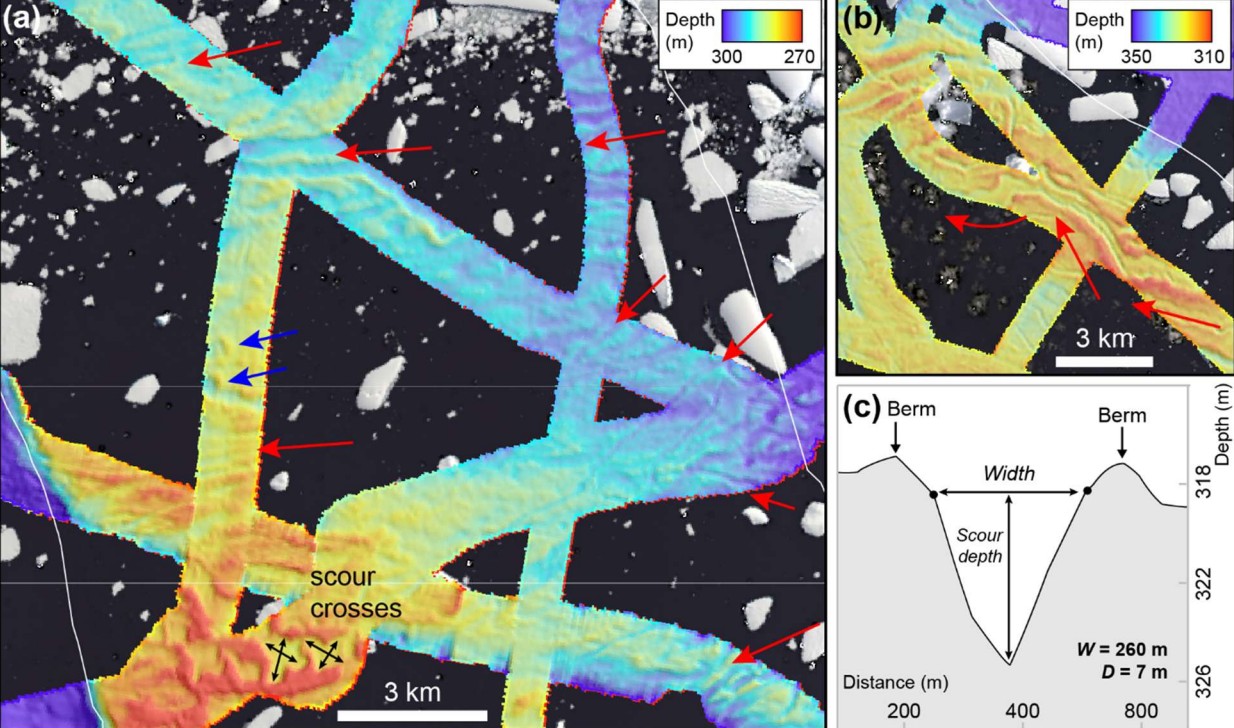

**Figure 2: Examples of iceberg scour types from Bear Ridge (boxes a and b located as shown on the map in Figure 1). The red arrows show likely iceberg transport paths westwards from the eastern Amundsen Sea and across Bear Ridge; blue arrows show iceberg plough ridges (berms); black arrows show the perpendicular direction of iceberg scouring in the crossed forms. Panel c) presents a schematic of a typical scour cross-section found on Bear Ridge.**

## 2.2 Sediment density and shear strength

In order to simulate the residence time of icebergs embedded in the bottom sediment, a model needs realistic values for the saturated density and shear strength, properties which determine resistance forces acting on grounded icebergs. In this section we present observations of saturated density and shear strength characteristic of the Amundsen Sea. Data from sediment cores recovered from the basin (Figure 3; Smith et al., 2011; Clark et al., 2024) provide a range of values for both the saturated density and shear strength. Whole core saturated density was determined using a GEOTEK multi-sensor core logger following standard methods (Niessen et al., 2007) while shear strength was measured with a shear vane (Smith et al., 2011; Clark et al., 2024).

Sediment thicknesses across different polar regions (and hence the maximum scour depth) differ markedly between areas of glacially eroded bedrock that are completely stripped of sediment and thick sequences of more deformable glaciomarine muds and subglacial tills. The composition of post-glacial seafloor (<15 m) sediments in the Amundsen Sea is variable, but typically consists of three broad lithological units. The uppermost unit, directly at the seafloor, consists of soft, water-rich muds with variable biogenic content. These sediments were deposited in an open or seasonally sea-ice covered glaciomarine environment distal (>25 km) from the ice sheet grounding line (Smith and Hogan, 2024) and are characterised by low shear strength and density values (<5 kPa, $<1.6 \times 10^3$ kg m$^{-3}$). Beneath these sediments, coarser-grained stratified to structureless, sandy to gravelly, terrigenous sediments occur. These 'transitional' sediments were deposited at or close to (<25 km) the ice sheet grounding line and lie directly above subglacial sediments below. The subglacial sediments in the basal unit consist of stiffer/denser, and largely structureless diamictons deposited subglacially during the Last Glacial Maximum, either as 'soft' deformation till (>5-20 kPa, 1.8-2.2 $\times 10^3$ kg m$^{-3}$) or 'stiff' compacted tills (>20 kPa, $>2.0 \times 10^3$ kg m$^{-3}$). Thicknesses of each unit vary, but postglacial glaciomarine mud tends to be thicker on the inner shelf, particularly where sedimentation is focussed in bathymetric troughs or depressions. Conversely, on shallow banks such as former pinning points (see Hogan et al., 2021), post-glacial sediments tend to be much thinner (<5 m), so that stiffer sediments and (crystalline) bedrock occur much closer to the seafloor. Figure 3 presents samples from all of the three units characterized by different saturated density and shear strength.

Compared to other sectors of the Amundsen Sea, comparatively few sediment cores have been recovered from Bear Ridge due to a combination of perennial sea-ice cover, and a general scientific focus on the glacial troughs. The seafloor drill rig MARUM-MeBo70 was used at three locations on the western flank of Bear Ridge (Gohl et al., 2017), revealing that indurated mudstones, likely of Eocene-Miocene age, crop out very close to the seafloor. Crucially, in areas such as Bear Ridge where post-glacial sediments are thin and crystalline basement/sedimentary strata crop out close to the seafloor iceberg ploughing depth will be retarded via solid body and static friction.

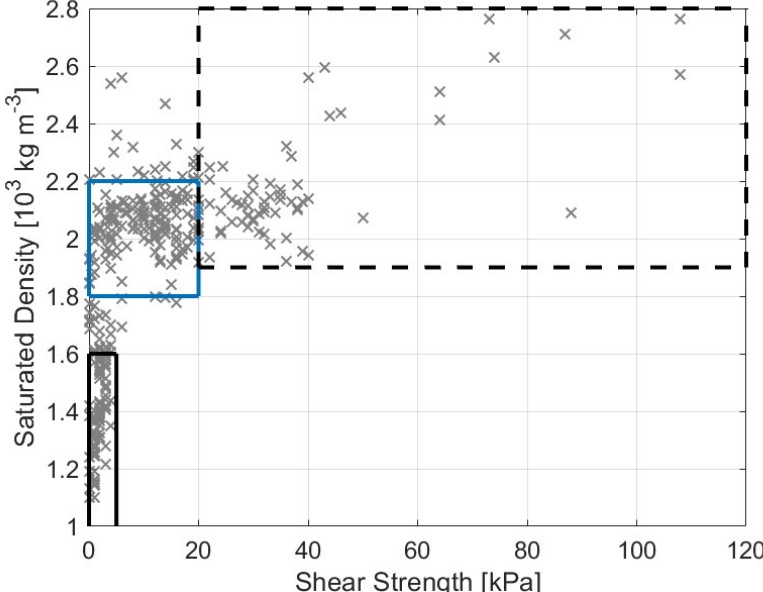


**Figure 3: Observed sediment properties for selected cores from the Amundsen Sea: the uppermost sediments consist of soft, water-**
**rich muds (solid black box); coarser-grained stratified to structureless diamictons and 'soft' deformation tills (solid blue box); stiffer**
**tills and/or denser subglacial diamictons (dashed black box). The soft/stiff till boundary follows Reinardy et al. (2009). Values outside**
**the boxes reflect other sediment types such as iceberg diamictons, glaciogenic debris flows etc.**

## 3 Modelling iceberg dynamics and grounding

We improve the capability to simulate iceberg drift, grounding, and ungrounding by building up a theoretical framework that
defines the relevant forces at play. We consider drivers of iceberg motion – such as surface winds and ocean pressure gradients
– as well as forces that dissipate mechanical energy via the effects of bottom sediment resistance and solid bedrock friction.

### 3.1 Forces acting on freely floating icebergs and their representation in NEMO

In order to fully understand the behaviour of grounded icebergs, we first explore their dynamics while they are in free flotation.
Floating icebergs are driven by the combined effect of relative winds (on the iceberg's upper surface and sides of the iceberg
freeboard), sea-ice drag, ocean waves, and ocean pressure gradients and relative ocean currents integrated over the depth of
the submerged keel and along the iceberg's basal surface (Merino et al., 2016). Icebergs are also subject to the Coriolis force
(Marsh et al., 2015; Cenedese and Straneo, 2022). The effect of tides (if represented) is indirect: tides alter the sea surface
height gradients, and hence, the water column thickness, ocean pressure gradients, and ocean drag. The acceleration of a freely-
floating iceberg as it moves (Bigg et al, 1996; Condron and Hill, 2021; Marsh et al., 2018) can be expressed as:
$$\frac{d\vec{u}}{dt} = \vec{a}_{pressure\ gradient} + \vec{a}_{Coriolis} + \vec{a}_{ocean\ drag} + \vec{a}_{atmos.\ drag} + \vec{a}_{sea-ice\ drag} + \vec{a}_{waves} \qquad (1)$$
where terms on the right-hand side denote accelerations due to the ocean pressure gradient, the Coriolis force, ocean drag,
atmospheric drag on icebergs, sea-ice drag, and wave drag (see the supplementary Table D1 for a list of all symbols used).
Importantly, grounded icebergs are also decelerated by an additional term, $\vec{a}_{ground}$, which represents the sum of all dissipative
forces acting on an iceberg. We assume that there is no continuous field of interacting icebergs, and so $\nabla \vec{u} = 0$ because icebergs
are treated as isolated rigid, solid bodies (in NEMO they are in fact represented as moving point particles when it comes to
advection).
The direct acceleration that the atmosphere and ocean flows impose on icebergs can be understood in terms of the
turbulent drag that the fluid exerts on the berg. This turbulent drag is assumed to be a quadratic function of the speed of the
iceberg relative to the fluid:
$$\vec{a}_{drag} = -C_{drag}|\vec{u} - \vec{u}_{fluid}|(\vec{u} - \vec{u}_{fluid}) \qquad (2)$$

where $C_{drag}$ represents an effective form drag (Wagner et al., 2022) coefficient that is different for air, seawater and sea-ice, and has units of an inverse length scale related to the horizontal extent of the iceberg (Martin and Adcroft, 2010). The effective drag coefficient takes into account the efficiency of momentum transfer between the fluid and the iceberg, the density ratio between the fluid and the iceberg, and the iceberg area exposed to the drag. The typical order of magnitude of this parameter when applied to ocean—iceberg drag is $10^{-4}$ $[m^{-1}]$ for thick tabular icebergs in the Amundsen Sea. Analogously to a turbulent drag, the sea-ice pack also exerts a drag on icebergs, especially in regions with a high sea-ice concentration (Marson et al., 2024). While the quadratic formulation of equation (2) is well established for high Reynolds numbers in the ocean and atmosphere, its application to sea-ice is used for simplicity, but little is known about its validity.

When considering ocean drag, it is important to account for the vertical shear in the upper ocean, which can significantly impact iceberg motion (FitzMaurice et al., 2016). The existing NEMO algorithms use the depth-averaged ocean velocity to estimate the ocean drag term:

$$Old \; \vec{a}_{ocean\;drag} = \frac{C_{drag}}{D_{keel}} \left| \vec{u} - \int_{D_{keel}}^{0} \vec{u}_{ocean}(z) \, dz \right| \left( \vec{u} - \int_{D_{keel}}^{0} \vec{u}_{ocean}(z) \, dz \right), \tag{3}$$

but the nonlinearity of the drag formulation means that equation (3) is inaccurate. In the update that we propose here, we consider the ocean currents at each depth level and average the resulting drag over the iceberg keel depth to arrive at an improved representation of ocean drag:

$$\vec{a}_{ocean\;drag} = \frac{C_{drag}}{D_{keel}} \int_{D_{keel}}^{0} |\vec{u} - \vec{u}_{ocean}(z)|\left( \vec{u} - \vec{u}_{ocean}(z) \right) dz, \tag{4}$$

where $D_{keel}$ is the keel depth.

Another significant source of acceleration for icebergs is the horizontal gradient in ocean pressure at different depth levels. Gradients in sea-surface height (SSH) form a large component of the horizontal pressure gradients, but density variations also modify pressure gradients within the water column. This latter effect was not represented in the existing iceberg dynamics in NEMO, and is introduced here. For example, in the x-direction, at each depth level z, we have a pressure gradient component:

$$\frac{\partial \left( \frac{P(z)}{\rho_0} \right)}{\partial x} = gh(z) \frac{\partial \left( \frac{\langle \rho'(z) \rangle}{\rho_0} \right)}{\partial x} + g \left( \frac{\langle \rho'(z) \rangle}{\rho_0} \right) \frac{\partial h(z)}{\partial x} + g \frac{\partial h(z)}{\partial x} \quad , \tag{5}$$

where $\langle \rho'(z) \rangle = \frac{1}{z} \int_{0}^{z} (\rho(z) - \rho_0) dz'$ is the density anomaly, relative to the reference Boussinesq density $\rho_0$, averaged over the overlying water column, $g$ is the magnitude of the acceleration due to gravity, and $h(z)$ is the water-column thickness above level $z$. Under the Boussinesq approximation, the contribution of the second term on the RHS may be neglected. In addition to large-scale dynamical features, tides also contribute to SSH gradients, while SSH anomalies are considered a major driver of iceberg motion, especially for weakly-grounded icebergs (Brown et al., 2017).

In addition to drag and pressure forces, icebergs furthermore respond to the Coriolis force due to the planetary rotation, which gives rise to an acceleration at a right angle relative to the velocity vector:

$$\vec{a}_{Coriolis} = -f\hat{k} \times \vec{u} \, , \tag{6}$$

where $f$ is the Coriolis parameter.

**3.2 Sediment resistance**

Grounded icebergs are subject to additional sources of acceleration and deceleration such as sediment resistance and solid-body friction, which have so far not been represented in the NEMO iceberg module and are introduced here for the first time. Here we make a distinction between several different cases: icebergs that plough only into the sediment and remain in motion or halt; icebergs whose base reaches the solid basement i.e., crystalline bedrock but remain in motion; and icebergs whose motion comes to a stop when grounded in the sediment and/or on the basement. Here we first describe the resistance experienced by a moving berg whose keel ploughs into the sediment. Guided by the most typical observed scour shapes in the Amundsen Sea, we assume that each iceberg ploughs a single v-shaped trench approximately 8-m deep.

There is a specialized body of literature discussing the interaction of icebergs with seafloor sediments. Empirical observations and theoretical considerations suggest that sediment resistance can be decomposed into three different mechanisms. An iceberg ploughing into sediments creates a wedge of sediment, composed of fracture plates (Chari et al., 1975; Chari et al., 1980). The weight of this wedge pushes back on the part of the iceberg's frontal face that has ploughed into the sediment with a force (in units of N):

$$\frac{\gamma'(H+D)^2 W}{2} \approx \frac{\gamma' D^2 W}{2} \tag{7}$$

where $\gamma'$ is the submerged unit weight of the bottom sediment (in units of N m$^{-3}$), $W$ is the width of the scour, $D$ is the depth of the scour, and $H \ll D$ is the height of the excavated sediment above the level of the surrounding seafloor (Chari et al., 1975; Chari et al., 1980). The submerged unit weight $\gamma'$ [N m$^{-3}$] in turn can be expressed as a function of the saturated sediment density $\rho_{sat}$ [kg m$^{-3}$] and the density of water $\rho_{water}$ [kg m$^{-3}$]:

$$\gamma' = g(\rho_{sat} - \rho_{water}). \tag{8}$$

Furthermore, there are shear stresses acting along the surfaces of the fracture plates within the wedge of sediment excavated by the iceberg:

$$Shear\ within\ wedge = 2\tau DW \tag{9}$$

where $\tau$ is the shear strength of the sediments in units of Pa (Chari, 1975; Chari et al., 1980). Lastly, as the excavated wedge is pushed forward along the path of the iceberg scour, the sides of the wedge experience lateral friction against the undisturbed surrounding sediments:

$$Lateral\ friction = \frac{\sqrt{2}}{2}\tau D^2. \tag{10}$$

This gives rise to a three-term expression for the maximum possible sediment resistance, which acts in the direction opposite to the iceberg drift if it is in motion, or opposite to the net driving force if the iceberg is static:

$$\vec{a}_{sediment\ max} = -\left[\frac{\gamma' D^2 W}{2} + 2\tau DW + \frac{\sqrt{2}}{2}\tau D^2\right] \cdot \begin{cases} \hat{u}/M_{iceberg} & , & |\vec{u_0}| > 0 \\ \hat{a}_{net\ drivers}/M_{iceberg} & , & |\vec{u_0}| = 0 \end{cases} \tag{11}$$

where $M_{iceberg}$ is the iceberg mass, $\hat{a}_{net\ drivers}$ indicates a unit vector in the direction of the net driving force (the sum of all nondissipative forces that drive iceberg motion) , $\hat{u}$ indicates a unit vector along the **projected** direction of future motion, which depends both on the present velocity and on the net driving force (see below), and $\vec{u_0}$ is the present velocity of the iceberg. In the model implementation (Section 3.4), the sediment resistance is not allowed to drive a net acceleration but only to decelerate iceberg motion or to oppose the forces trying to set an iceberg in motion, hence why the above is formulated as a maximum force.

The latter can be 10s to 100s of meters thick, particularly in bathymetric depressions such as glacial troughs. Following our Amundsen Sea example, the observed scours (Section 2.1) and compiled sediment properties (Section 2.2) inform our decision to assume the existence of a uniform 8-m sediment layer with a saturated density of $1.8 \times 10^3$ kg m$^{-3}$ and shear strength of 6 kPa. Beneath this single uniform sediment layer we assume the existence of a basement. Thus, iceberg ploughing is allowed to occur up to a maximum of 8-m depth while it remains in motion, but then other forces come into play, described next.

## 3.3 Gravity, solid-body friction, and static friction

When a moving grounded iceberg ploughs through all the sediment and reaches a sloping basement, it can move up the basement slope if the berg has enough momentum and/or sufficient sources of sustained acceleration (Chari 1975). As the berg moves up the slope, it is lifted upwards, its freeboard increases, and as a result, the buoyancy force of the ocean no longer balances the weight of the iceberg and a force directed into the basement supports the remainder (Figure 4). This gives rise to a gravitational force, which converts the kinetic energy of the iceberg motion into potential energy upon grounding, and also causes a solid body friction between the berg and the basement.

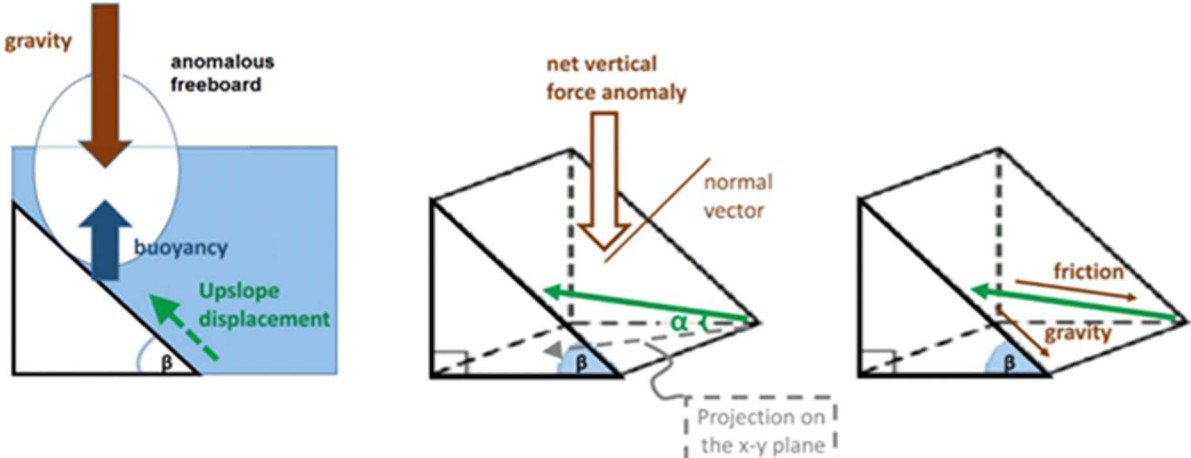

317

**Figure 4: Vertical force budget.** An idealized schematic of an iceberg trajectory (green arrow) going up a topographic slope. The angle of the trajectory (solid green arrow) relative to its projection on the horizontal x-y plane (dashed grey arrow) is denoted $\alpha$. The local maximum slope of the topography is denoted $\beta$. The vector normal to the topography is indicated with a brown line. Schematic angles not shown to scale with respect to the relevant topographic slopes.

Lifting the iceberg out of floatation means that a portion of the iceberg's weight is no longer supported by the buoyancy of ice relative to seawater. The vertical displacement yields a deficit of $\rho_{water} H_{freeboard\ anom.}$ in the mass of seawater displaced by the berg per unit area, and a corresponding net downwards gravitational force. Dividing this force by the berg mass per unit area, we arrive at an acceleration whose projection down the sloping topography is

$$Projection\ of\ gravitational\ acceleration = \frac{\rho_{water}}{\rho_{ice}} \frac{H_{freeboard\ anom.}}{H_{Total}} g \sin \beta. \tag{12}$$

where the above takes into account the density difference between the iceberg $\rho_{ice} = 875$ kg m⁻³ – including firn air trapped in the ice (Veldhuijsen et al., 2023) – and the ambient sea water $\rho_{water} = 1025$ kg m⁻³, and $\vec{g}$ is the gravitational acceleration vector, and the topography has a local maximum slope $\beta$ (Figure 4). It is the horizontal components of the force balance that are of interest here. We can furthermore express the projection of the gravitational force directed straight down the slope as a horizontal acceleration term:

$$\vec{a}_{gravity} = \frac{\rho_{water}}{\rho_{ice}} \frac{H_{freeboard\ anom.}}{H_{Total}} g\ (\sin \beta \cos \beta)\hat{s} \tag{13}$$

where $\hat{s}$ is a unit vector in the x-y horizontal plane aligned with the direction of the maximum topographic slope at the location of the iceberg, and $g$ is the magnitude of the gravitational acceleration.

There is another gap in the literature when it comes to the solid-body friction between the base of the iceberg and the basement beneath any sediments. One may, however, draw a qualitative analogy to the friction between glaciers and rock, whose representation in models can also be reduced to a solid-body (Coulomb type) sliding law (Fowler et al., 2010), which is independent of the speed in certain regimes. This term in the force balance is briefly discussed in Chari's dissertation (Chari, 1975). Generally, Coulomb friction is assumed to be proportional to the normal confining force at the interface between the two solid objects, in our case the seafloor basement and the base of the iceberg. When the iceberg is in contact with basement, this normal force is a projection of the net vertical imbalance between gravity and buoyancy. If an iceberg moves up a slope at an angle $\alpha$ relative to the horizontal plane, the magnitude of the Coulomb friction in the sloping plane can be expressed as

$$Coulomb\ friction = \frac{\rho_{water}}{\rho_{ice}} \frac{H_{freeboard\ anom.}}{H_{Total}} g\ \mu \cos \beta\ \cos \alpha \tag{14}$$

where $\mu$ is a dimensionless coefficient of solid body friction, whose value for the interaction between icebergs and the seafloor basement is not known. Equation (14) describes a deceleration term acting against the horizontal direction of motion $\hat{u}$ (Figure 4) in the case of moving icebergs. The kinetic friction has a vector orientation opposing the iceberg velocity:

$$\vec{a}_{solid} = -\frac{\rho_{water}}{\rho_{ice}} \frac{H_{freeboard\ anom.}}{H_{Total}} g\ \mu \cos \beta\ \cos \alpha\ \hat{u}\ . \tag{15}$$

In the model implementation (Section 3.4), the magnitude of the kinetic friction vector $\vec{a}_{solid}$ is capped, so that it
never drives a net acceleration. The value of the non-dimensional coefficient $\mu$ cannot be constrained directly and could only
be inferred from the resulting iceberg grounding behaviour. In this study, we test a range of values including $\mu = 0$, (which
corresponds to no Coulomb friction but only sediment resistance and gravity), $\mu = 0.002$, and $\mu = 0.2$, but we use $\mu = 0.002$
in simulations with a realistic Amundsen Sea model configuration (See Appendix B for a discussion of parameter sensitivity).
Our results are also robust with respect to the choice of value for the Coulomb coefficients $\mu$ and $\mu_{stat}$ (Appendix B). We
find that varying the magnitude of $\mu$ does not affect the episodes of motionless grounding in numerical simulations (Figure B1
in Appendix B). In contrast, the static episodes marked appear to be more influenced by the sediment resistance and the
processes that thin the icebergs.
We note that the coefficient of Coulomb friction $\mu$ is usually assumed to be different in the case of moving objects
(kinetic friction) versus objects at rest (static friction, $\mu_{stat}$). Usually, the coefficient of static friction $\mu_{stat}$ is greater than or
equal to the value of the kinetic friction coefficient for the same type of surfaces under similar conditions. Another noteworthy
difference between the kinetic and static solid body friction is related to the orientation of the frictional force vector. In the
case of static friction, there is no motion, so friction does not act in a direction opposite to the velocity vector, which is zero.
Instead, static Coulomb friction and sediment resistance jointly act in a direction opposing the net sum of all acceleration terms
that try to set an object into motion. Therefore, in the case of grounded icebergs, the maximum possible decelerating force per
unit mass due to static friction is:
$$\vec{a}_{static-sol} = -\frac{\rho_{water}}{\rho_{ice}}\frac{H_{freeboard\,anom.}}{H_{Total}}g\mu_{stat}\cos\beta\,\cos\alpha\,\widehat{a_{net}} \tag{16}$$
which is similar to the expression for kinetic friction, but here, were orient the vector against the sign of the net acceleration
$\vec{a}_{net\,drivers}$, which includes fluid and ice drag, the Coriolis force, and gravitational accelerations but none of the dissipative
accelerations, and is aligned with the unit vector $\widehat{a_{net}}$. In our study we impose a static coefficient $\mu_{stat}$ that is 100 times larger
than the corresponding kinetic coefficient $\mu$, which is an arbitrary choice tested below.
Last but not least, an iceberg moving along solid-basement topography with non-constant slopes experiences changes
in motion due to the surface geometry. For example, if an iceberg moves up a steeper slope, some of its horizontal momentum
is reoriented in the vertical direction, a "curvature term" $\vec{a}_{curvature}$ discussed in Appendix B. In total, then, the new iceberg
grounding model is made up of 4 terms, each of which applies under different conditions:
$$\vec{a}_{ground} = \vec{a}_{sediment} + \vec{a}_{gravity} + \vec{a}_{solid} + \vec{a}_{curvature}\,, \tag{17}$$
and when an iceberg is grounded the above terms (if present) are added to the full iceberg momentum budget.

## 3.3 Representation of iceberg melting and capsizing in NEMO and their role in ungrounding

Melting plays an important role in the ungrounding of icebergs from bottom topography. NEMO breaks down the iceberg
melting into three processes: basal melting, lateral melting, and wave erosion (Martin and Adcroft, 2010). Wave erosion is a
function of the relative wind velocity and the ambient ocean temperature. Basal melting is represented as a function of
temperature and velocity relative to the ocean, while the melting due to buoyant convection along the side walls is assumed to
depend only on temperature (Martin and Adcroft, 2010). Wave erosion and buoyant convection decrease the horizontal width
and length of a simulated iceberg, while basal melting reduces the vertical thickness, which is relevant to ungrounding.
However, the freshwater released by all three simulated processes, including basal melting, is injected at the NEMO ocean
surface.
For icebergs that float or follow the old grounding schemes, the rate of basal melting is proportional to the relative
velocity between the iceberg and the ocean current at the height of the iceberg base. However, we allow grounded icebergs to
plough into the sediment. In this case, for the purpose of calculating basal melting, we compute the relative velocity at the
deepest ocean level above the sediment.

When the iceberg's geometry becomes unstable and its centre of mass shifts, the iceberg can spontaneously roll over and capsize (Wagner et al., 2017 OM). The existing NEMO code uses a ratio between the iceberg's longer horizontal length and its keel depth as a criterion for stability against rolling (Weeks and Mellor, 1978). However, (Wagner et al., 2017 OM) point out that this approach used in models is unphysical. The appropriate indicator of vulnerability to capsizing is the ratio between the horizontal width and the full thickness (Wagner et al., 2017 OM).

Olivé Abelló et al. (2025), a companion study, updates the rolling criteria in NEMO by appropriately considering the ratio between the iceberg width and thickness. Olivé Abelló et al. (2025) build up on the approach used by MacAyeal et al. (2003) and Bigg et al. (1997 and 2018), while correcting errors highlighted by Wagner et al. (2017, OM). Our work and the Olivé Abelló et al. (2025) companion study test each of the new updates to the NEMO iceberg algorithms separately. Our study introduces the new representation of grounding but relies on the old rolling criterion. In comparison, Olivé Abelló et al. (2025) use the new rolling criterion combined with the old grounding algorithms. These coordinated efforts will contribute to the improved representation of icebergs in future versions of NEMO.

**3.4 Implementation of the new iceberg dynamics algorithms in NEMO**

We implement the relevant iceberg dynamics and grounding theory as an update (Kostov et al., 2025b) to the existing representation of icebergs in NEMO. The iceberg model in NEMO operates within the ocean domain. However, iceberg motion is not confined to the resolved ocean grid, and instead bergs are dynamically represented as Lagrangian mass particles that propagate across and within ocean grid cells. For the purposes of calculating melt rates, as well as turbulent drag, each iceberg is associated with horizontal and vertical dimensions: length, width, and thickness. The iceberg motion is calculated using a $4^{th}$-order Runge-Kutta scheme (Marsh et al., 2015; Merino et al., 2016). Acceleration, velocity, local SSH slope, and horizontal position are calculated at each stage of the Runge-Kutta scheme and the final estimate for a given iceberg timestep is a weighted sum of the result from each of the four Runge-Kutta stages.

As the icebergs move within the resolved grid cells, ocean properties such as velocity, temperature, SSH, and bathymetry are horizontally interpolated onto the sub-gridscale position of the bergs. We calculate vertically-averaged ocean drag terms that we update and apply as acceleration terms within each stage of the Runge-Kutta scheme for iceberg dynamics. At each Runge-Kutta stage, we also apply horizontal pressure gradients interpolated onto iceberg positions. The pre-existing code included only the gradient components due to SSH anomalies. Here we also account for the pressure gradient terms that arise due to water density anomalies, exactly as they are calculated and applied in the dynamics of the liquid ocean domain. Our calculation of the pressure gradient terms is therefore compatible with the multiple vertical coordinate systems that are available in NEMO, e.g, fixed z-level or z*-coordinates that stretch and contract matching the SSH anomalies. In our implementation, icebergs do not feel the component of the ocean pressure gradient due to the iceshelf load over iceshelf cavities, as icebergs themselves cannot enter the cavities. However, icebergs are often found near the iceshelf edge even long after they have calved. That is why, in algorithm, we mask the ice-shelves, such that icebergs in their vicinity can approach the shelf edge without being exposed to unphysical pressure gradients.

The NEMO model has some existing, crude representations of iceberg grounding (Olivé Abelló et al., 2025). At each of the four Runge-Kutta stages, these routines consider criteria under which iceberg velocity or components of the iceberg velocity are set to zero as topographic obstacles are encountered. The grounding routines use the projected forward motion of the iceberg and therefore have an implicit element. The pre-existing algorithms allow for two options: 1) icebergs come to a complete stop when their keel reaches a topographic obstacle; or 2) the component of an iceberg's velocity vector normal to a topographic obstacle is set to zero, while the tangential component is preserved.

In our new implementation, we also apply grounding routines at each stage in the Runge-Kutta scheme. However, we represent the interaction between icebergs and the seafloor in terms of acceleration terms due to gravity, solid-body friction,

and sediment resistance (Section 4). These acceleration vectors are projected along the x and y-axes (zonal and meridional) of iceberg motion, which are aligned with the ocean model grid (Figure 4). The acceleration/deceleration terms that we introduce depend on the local slope of the topography and on the angle at which an iceberg moves relative to the horizontal plane (which may be different from the maximum topographic gradient, Figure 4). In order to calculate these angles for each Lagrangian iceberg particle, we track and store interpolated bottom topography along an iceberg trajectory and calculate the corresponding slope angles.

In addition, we apply different solid-body friction coefficients depending on whether the grounded iceberg is in motion or at rest. For numerical purposes, we do not set a zero speed as the threshold for assuming no motion. Instead, when the speed drops below a small magnitude ($5 \times 10^{-5} \; m \; s^{-1}$), we set it to zero and assume a regime of static friction. This way we avoid situations where we see numerical stick and slip of icebergs alternating between no motion and brief periods of very slow motion.

While turbulent drag, geometric deflection, and gravity can act in any direction to accelerate or decelerate a berg, the dissipative sediment resistance and solid-body friction terms only act to decelerate motion. As described above, we do not allow the two dissipative terms, solid friction and sediment resistance, to drive motion but only to bring an iceberg to a stop and/or keep it at rest. Therefore, we impose that

$$\frac{d\vec{u}}{dt} = -\frac{\vec{u_0}}{\Delta t} \; \text{ for } \; |\vec{a}_{solid} + \vec{a}_{sediment}| \geq \left|\vec{a}_{net \; drivers} + \frac{\vec{u_0}}{\Delta t}\right|, \qquad (18)$$

where $\Delta t$ is the model timestep and $\vec{a}_{net \; drivers}$ is the net sum of all acceleration terms that do not dissipate mechanical energy. Even if the condition in (18) is achieved at only one of the four Runge-Kutta stages of the velocity calculation, the zeroing of an x- or y-velocity component is applied to the output of all four stages. Furthermore, if setting the velocity component to zero brings the total iceberg speed below $5 \times 10^{-5} \; m \; s^{-1}$, we force the iceberg to stop.

The fact that iceberg motion is computed using a separate numerical scheme within the NEMO ocean domain allows us to apply temporal substepping within the iceberg code while keeping the same time-step size for the ocean and sea-ice. We thus define a parameter that sets the ratio of the ocean timestep to the shorter iceberg timestep. We find that allowing a shorter iceberg timestep is numerically important when modelling the shape and evolution of iceberg scours because forces such as gravity along the sloping topography may act on much shorter timescales compared to other sources of acceleration. (See Appendix B, where the numerical relevance of sub-stepping is demonstrated in an idealized experiment with gravity but no dissipative forces along a sloping bottom). At the same time, when modelling iceberg residence times and locations on the resolved ocean grid, substepping becomes less critical (Appendix B).

In summary, the implementation of our updated algorithms affects the dynamics of both freely floating and grounded icebergs through the following set of changes: 1) we use a vertical average of the full profile of non-linear ocean drag along the keel; 2) we make the ocean pressure gradients on icebergs a function of both SSH and density; 3) we allow icebergs to experience sediment resistance when grounding; 4) we allow icebergs to experience Coulomb friction and gravitational acceleration when the keels reach the solid basement.

## 3.5 Amundsen Sea NEMO test cases

We test our updated iceberg dynamics in a regional Amundsen Sea configuration of the Nucleus for European Modelling of the Ocean (NEMO-v4.2.0, Madec et al. 2022). This includes the SI3 sea-ice component (Vancoppenolle et al., 2023), a representation of ocean–ice-shelf interactions (Mathiot et al., 2017), and a Lagrangian iceberg module (Marsh et al., 2015; Merino et al., 2016).

The extent of the regional domain, the horizontal resolutions (0.25°), and the topography are the same as in Caillet et al. (2023), except that we do not impose a "wall of icebergs" over Bear Ridge to block sea ice. We use 121 vertical levels as in Mathiot and Jourdain (2023), with a thickness increasing from 1 m at the surface to 20–30 m between 100 and 1000 m

depth. The cell thicknesses are slightly compressed or stretched to follow the SSH variations (Z* coordinates), and cells adjacent to the sea bed and ice shelf draft are described as partial steps (Adcroft et al., 1997).

The initial state and lateral boundary conditions are from 5-day mean outputs of the global simulation of Mathiot and Jourdain (2023). Tides are neither generated nor prescribed at the boundaries, which is expected to have a limited impact in the Amundsen Sea (Jourdain et al., 2019). Tides are relatively weak in the Amundsen Sea relative to other Antarctic sectors (Jourdain et al, 2019). The surface boundary conditions are calculated through the bulk formula of Large and Yeager (2008) from the JRA55-do atmospheric reanalysis (Tsujino et al., 2018).

We release icebergs in January of the first year of a simulation, i.e., 1979. Our aim with these simulations is to demonstrate the new iceberg grounding physics. We are interested in a realistic historical setting, but in this study, we do not aim to explore a broad range of long-term natural variability in the Amundsen Sea and its impact on icebergs. All the icebergs in the Amundsen Sea simulations are released east of Bear Ridge and then allowed to drift, ground, and unground according to the modelled ocean, atmosphere, sea ice and iceberg conditions. We conduct two sets of tests, each with a different iceberg population. In the first set, labelled THICK hereafter, all test icebergs are tabular, with the same mass, $3.9 \times 10^{11}$ kg, uniform density of 875 kg m$^{-3}$, and thickness of 395 m. In the second set of tests, labelled MEDIUM, all icebergs are 250 m thick, each with mass of $3.8 \times 10^{10}$ kg. However, MEDIUM icebergs are not thick enough to interact with the topography along Bear Ridge, and hence, they do not feature in our grounding analysis. We analyse the force balance of MEDIUM icebergs in free flotation.

In this study we explore the results of two types of simulations, where each type of simulation is performed with each type of iceberg thickness. In the short simulations, labelled 'SHORT' hereafter, we initially seed a total of 497 identical test icebergs one in each grid cell (~1/4° by ~1/4°) in the region east of Bear Ridge enclosed by 109.9° W, 106.2° W, 74.3° S, and 72.0° S (Figure 1). While it is not realistic to expect hundreds of thick tabular icebergs to simultaneously occupy this region, our icebergs do not interact with each other or block sea-ice (while sea-ice does impact icebergs). In our SHORT simulations, even when icebergs melt, we do not allow any iceberg meltwater to be injected in the ocean, so as not to perturb the background conditions with each set of 497 icebergs. We run this simulation with a large ensemble of icebergs but only over the course of one full year spanning all seasons. Each of the grid boxes that we seed is a possible location where an iceberg may be found, and hence, we explore the behaviour of thick tabular icebergs when exposed to the local conditions. This large population allows us to analyze the statistics of iceberg dynamics while focusing specifically on the thick tabular icebergs that are most likely to exhibit a long residence time when grounding on Bear Ridge. We aim to explore the differences in the dynamical behaviour of icebergs of different sizes by comparing the THICK and MEDIUM populations. In order to preserve the large thicknesses and horizontal areas in the THICK iceberg population, we do not allow the latter to melt during a SHORT 1-year simulation.

We also run longer simulations labelled 'LONG', which are extended farther in time over 4 years to enable a comparison between modelled and observed iceberg scours. We seed a much smaller subset of 14 identical icebergs in a narrower box, bounded by 109.9° W, 109.2° W, 73.9° S, and 73.5° S (Figure 1), on the eastern flank of Bear Ridge. We explore the simulated scours left by the THICK icebergs whose motion is arrested on Bear Ridge. In simulation LONG, we do allow THICK icebergs to melt, as that affects their motion and residence times along the bottom: in our experiments with tabular icebergs, melting (Section 3.3) is a mechanism that can allow icebergs to unground and resume free flotation. The released meltwater is injected in the ocean and added to the precipitation at the surface.

**4 Results: Force balance and behaviour of thick tabular icebergs in the Amundsen Sea configuration**

In our Amundsen Sea numerical simulations, we observe the behaviour of thick tabular icebergs that are less typical of the Arctic but often found near Antarctica and whose large volume and mass plays an important role in their dynamics

(Wagner et al., 2017). We are interested in the forces that act on these large icebergs in free flotation, upon grounding, and upon ungrounding. Although our main focus is the behaviour of grounded icebergs, we also aim to understand the forces acting on freely floating icebergs before they may ground and after they unground. In this analysis, we also test the impact of the new iceberg dynamics and grounding algorithms that we have implemented in NEMO in the important Amundsen Sea regional context.

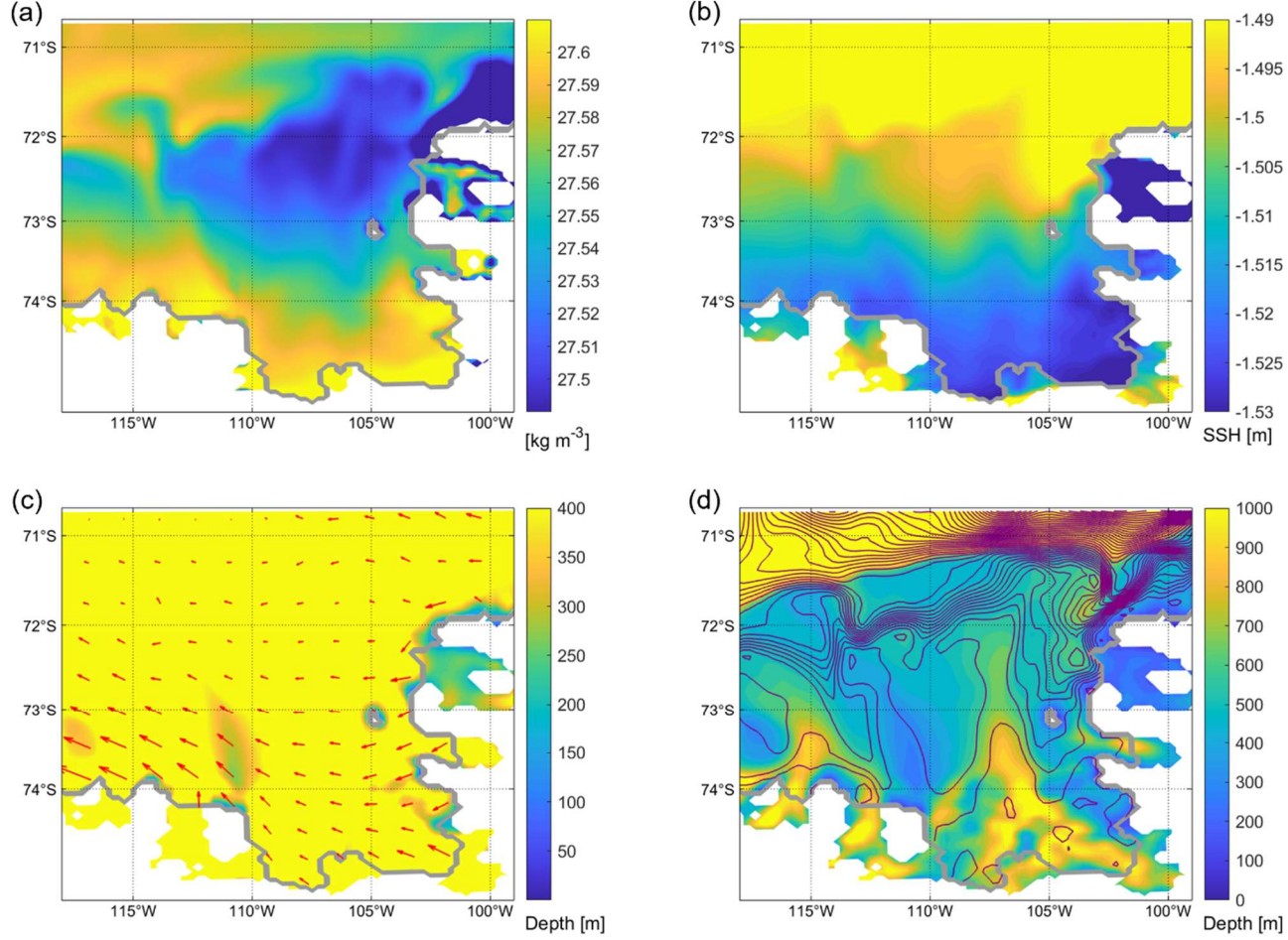

**Figure 5: Climatology (1979 through 1982) of the Amundsen Sea configuration. (a) potential density [kg m$^{-3}$] referenced to the surface, vertically averaged over the top 310 m with shallower regions masked; (b) sea-surface height [m]; (c) wind stress vectors [N m$^{-2}$]; (d) barotropic streamfunction contours [0.1 Sv intervals]. The bottom depth [m] is superimposed in c and d. The depth range in c chosen to highlight the location of the shallow Bear Ridge and a wider depth range in d to highlight the alignment of streamlines and topography. The ice shelf edge is marked by the thick gray line.**

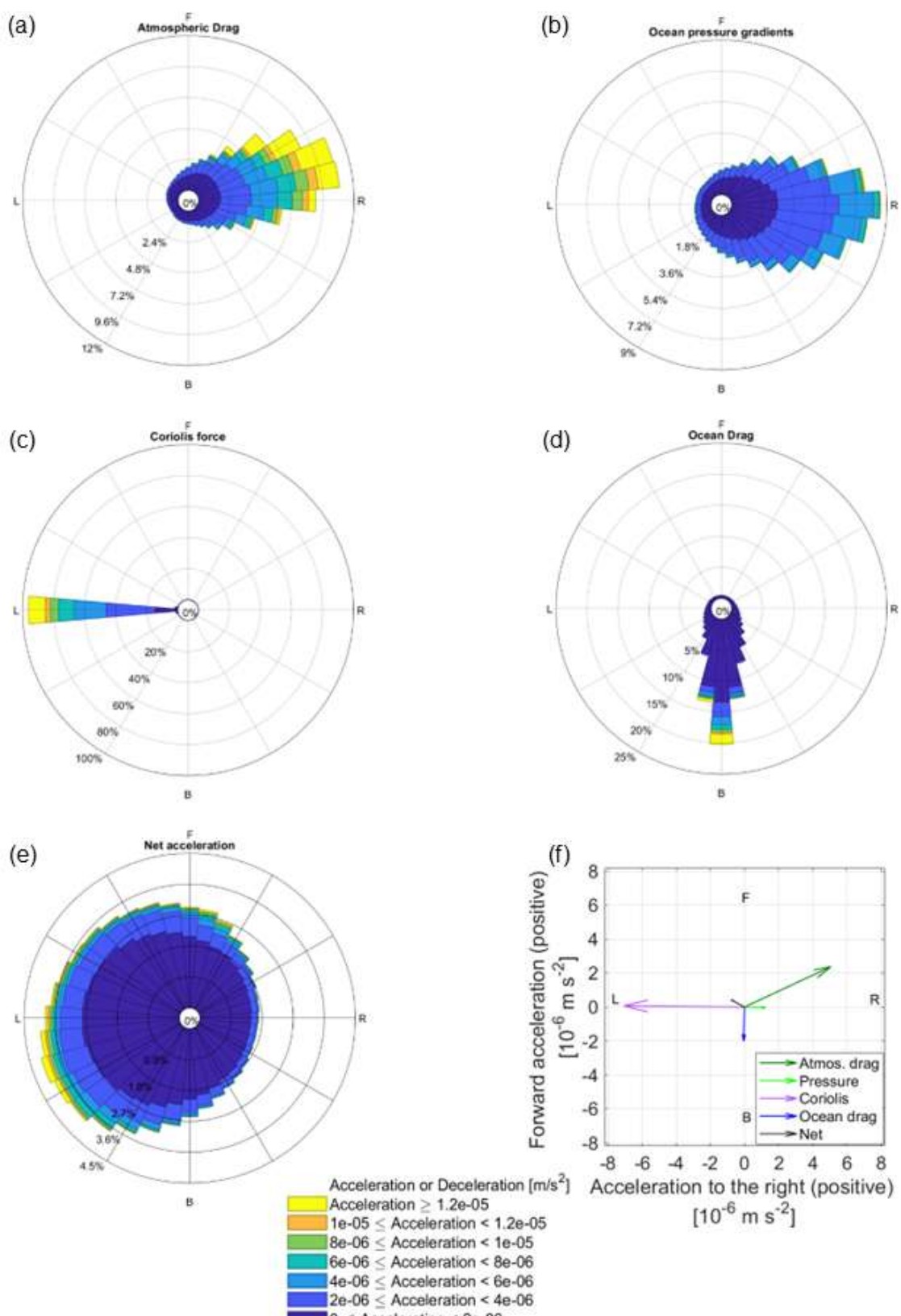

528

Figure 6: Contributions of atmospheric drag (a), the oceanic pressure gradients (b), the Coriolis force (c), the ocean drag (d), the net acceleration (e), and the average of a-d across all test icebergs and all timesteps (f) to the momentum budget of simulated freely floating THICK icebergs in the Amundsen Sea in the SHORT simulation. Directions are denoted as F (forward, along the iceberg's direction of motion), B (backward, against the direction of motion), L (to the left of the direction of motion), R (to the right). The full length of the radial bars indicates the probability density function of acceleration in a given direction. For each panel in a-e, the sum of all directions is 100%, but the bars along some of the directions are relatively small. For each bar in a-e, the range of acceleration magnitudes is indicated by the colours, and the probability of each magnitude class is indicated by the radial distance. Colour sections in a-e indicate the frequency distribution of the accelerations in each magnitude class. Note that the radial axis is different between panels a-e, while the colour scale is common to panels a-e. Panel f has a separate legend and a different colour scheme. The first 100 timesteps of the simulation, when the iceberg motion is initialized from rest, are omitted. Software written by Daniel Pereira was used to create the wind rose (https://www.mathworks.com/matlabcentral/fileexchange/47248-wind-rose, last access: November 2025, MATLAB Central File Exchange)

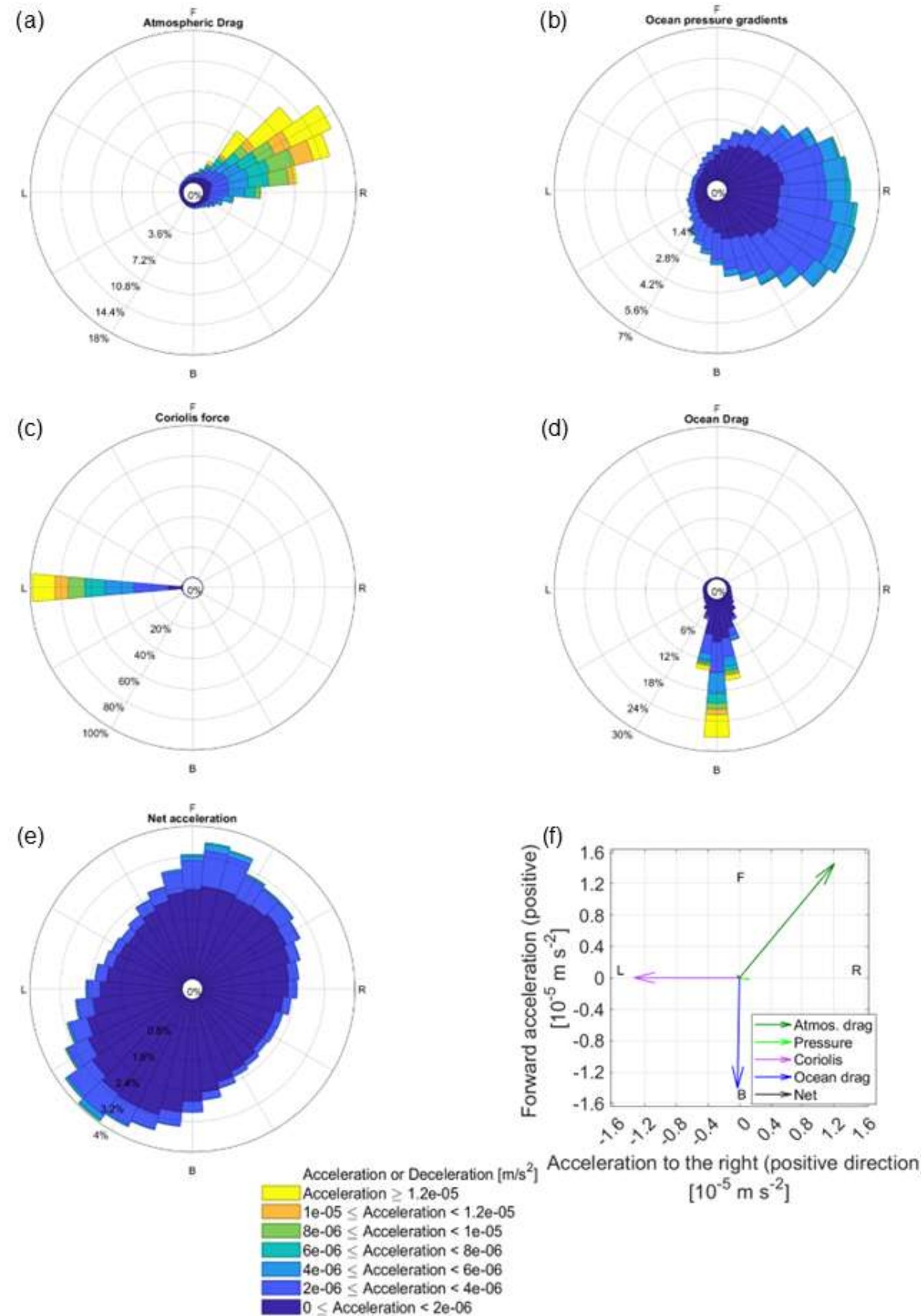

541

542 **Figure 7: Same as Figure 6 but for freely floating MEDIUM icebergs in the Amundsen Sea in the SHORT simulation.**

**4.1 Background conditions and force balance of freely-floating icebergs**

In order to understand the trajectories and behaviour of icebergs in the eastern Amundsen Sea, we first explore the wider climatic conditions, including the oceanic and atmospheric climatology and variability in the region. There is a relatively lower water column density to the east of the ridge compared to the water column over the ridge (Figure 5a), but this is compensated by a positive zonal SSH gradient between 112°W and 110°W (Figure 5b), which in combination with the Coriolis force would drive icebergs southward near the Ridge (Figure 5a). Overall, there is very little ocean transport across Bear Ridge although the prevailing winds are oriented north-westward across the ridge (Figure 5c,d).

Using model output from the large ensemble of freely-floating icebergs in Amundsen Sea Simulation SHORT, we compute statistics showing how often a given force acts along a particular direction relative to the iceberg motion (the length and orientation of the bars in the Figure 6 rose plots for THICK icebergs and Figure 7 for MEDIUM icebergs). We furthermore consider how often (the length of the individual color segments in Figures 6 and 7) a given force has a magnitude that falls within a particular range (the color scheme indicates the magnitude range). We point out that some forces represented in the iceberg model play a less prominent role compared to the main drivers. For example, the acceleration due to wave radiation is much smaller than $1.5 \times 10^{-6}$ m s$^{-2}$ and so is neglected from the figures, while the acceleration due to sea-ice drag is always smaller than the other terms (see Figure C1 in Appendix C).

We first consider the dynamics of freely-floating THICK icebergs in the period before they encounter the Bear Ridge or the shelf-edge, where they may ground or come to a stop. This helps us obtain a more comprehensive understanding of iceberg behaviour in the region. We also omit the first 100 timesteps of the simulation, when iceberg motion is initialized from rest. For freely-floating THICK icebergs, the most common situation by far is that the atmospheric form drag, the Coriolis force acting on icebergs, and the ocean form drag are nearly in a three-way balance (Figure 6) with very little net acceleration. The Coriolis force is oriented solely to the left, perpendicular to the direction of iceberg motion, while the acceleration due to wind stress and ocean pressure gradients is mostly oriented to the right of the direction of motion. On average, the corresponding forces on THICK icebergs are nearly orthogonal to the direction of motion and therefore do very little work on the freely floating large icebergs. This result is not peculiar to tabular icebergs in the Eastern Amundsen Sea but is more fundamental and agrees with the theoretical arguments of Wagner et al. (2017).

In our results, the (updated) ocean drag on freely floating icebergs plays a less prominent role compared to other forces and mostly opposes the forward motion of the bergs. Compare the small positive projection of the atmospheric drag along the forward direction of motion (Figure 6a and 7a) and the small deceleration due to ocean drag (Figure 6d and 7d). This result for large THICK icebergs may be expected on theoretical grounds. Here we briefly revisit the Wagner et al. (2017) arguments about the motion of large icebergs from a new perspective and with an attempt to recover the same result in fewer steps. We also specifically avoid making the assumption that the ocean's water column is in geostrophic balance, which is not necessarily applicable when modelling iceberg dynamics (e.g., Mountain, 1980). Unlike Wagner et al. (2017), we attempt to account for a realistic Ekman drift in our theoretical analysis and show that the same arguments remain valid.

We first consider the decomposition of the iceberg velocity $\vec{u}$ into a component that matches exactly the ocean's horizontal velocity $\vec{u}_{ocean}$ and a relative mismatch $\Delta\vec{u}$. The evolution, in an Eulerian sense, of the local ocean velocity over the upper ocean layer can be approximately expressed as:

$$\frac{\partial \vec{u}_{ocean}}{\partial t} \approx -f\hat{k} \times \vec{u}_{ocean} + \text{(Pressure gradient terms)} + \text{(Ekman ageostrophic acceleration due to wind stress)} \qquad (19)$$

where we have assumed that the main source of ageostrophic transport in the Ekman layer (of thickness $\delta$) comes from the tendency for Ekman drift due to surface wind stress $\vec{\tau}_{atm-ocean}$ (see Bigg et al., 1996). Under equilibrium conditions, the geostrophic component of $f\hat{k} \times \vec{u}_{ocean}$ balances the pressure gradient terms and the ageostrophic component of $f\hat{k} \times \vec{u}_{ocean}$ vertically averaged over the Ekman layer balances the wind-stress acceleration.

While the ocean water column contains an Ekman spiral, the icebergs do not, and this is a potential source of mismatch
between the velocity of icebergs and the surrounding ocean. In our study, we focus on thick icebergs, whose keel depth exceeds
the depth of the Ekman layer. If we consider the relative iceberg-to-ocean velocity $\Delta\vec{u}$ averaged over the keel depth, we can
express the Ekman acceleration term vertically integrated over the Ekman layer and then averaged over the full keel depth of
the icebergs as
$|\text{Ekman acceleration}| = \left|\frac{\tau_{atm-ocean}}{\rho_0 D_{keel}}\right|$ over the keel depth,                                           (20)
such that the vertically-averaged Ekman component of $\vec{u}_{ocean}$ is $\frac{\tau_{atm-ocean}}{f\rho_0 D_{keel}}$. The Ekman acceleration averaged over the depth
of the Ekman layer is oriented along the wind direction such that the average Ekman drift $\frac{\tau_{atm-ocean}}{f\rho_0 D_{keel}}$ is to the left of the wind
direction in the Southern Hemisphere.
The ocean pressure gradients exert the same force on the submerged portion of the iceberg as they would on an
equivalent water parcel. Therefore, we can express the evolution of the iceberg-to-ocean velocity mismatch $\Delta\vec{u}$ as arising from
a set of ageostrophic and dissipative processes. We can ignore the small terms due to sea-ice drag and wave drag. Furthermore,
assuming that the horizontal gradients of $\vec{u}_{ocean}$ are small along the iceberg trajectory, we can focus on the Eulerian evolution
of $\Delta\vec{u}$:
$\frac{\partial(\Delta\vec{u})}{\partial t} = \frac{\partial\vec{u}}{\partial t} - \frac{\partial\vec{u}_{ocean}}{\partial t} \approx$
$\approx -f\hat{k}\times\Delta\vec{u} - C_{drag-ocean}\Delta\vec{u}|\Delta\vec{u}| - C_{drag-atm}(\vec{u}-\vec{u}_{atmos})|\vec{u}-\vec{u}_{atmos}|$
$-\frac{\tau_{atm-ocean}}{\rho_0 D_{keel}}$                                                                (21)
Under the additional assumption that typically $|\vec{u}| \ll |\vec{u}_{atmos}|$, the above becomes
$\frac{\partial(\Delta\vec{u})}{\partial t} \approx -f\hat{k}\times\Delta\vec{u} - C_{drag-ocean}\Delta\vec{u}|\Delta\vec{u}| + C_{drag-atm}\vec{u}_{atmos}|\vec{u}_{atmos}| - \frac{\tau_{atm-ocean}}{\rho_0 D_{keel}}$            (22)
We assume that the main source of the mismatch $\Delta\vec{u}$, if present, can be attributed to the different atmospheric drag felt by
icebergs relative to the surrounding water column, which experiences Ekman drift. Notice that the last two RHS terms depend
primarily on winds as an external source of acceleration, while the first two RHS terms depend on the relative iceberg-to-ocean
velocity $\Delta\vec{u}$. When winds input momentum into iceberg motion, and into the surrounding water column, this creates a
differential acceleration between the iceberg and the ocean. If equation (22) is not in balance, the iceberg's relative speed $\Delta\vec{u}$
increases, and motion gets deflected to the left by the Coriolis force until the first two terms on the RHS balance the last two,
wind-dominated terms. In a steady state, the Coriolis force and ice-ocean drag balance the wind-dominated last two terms in
equation 22.
The question is whether the Coriolis term (the first on the RHS) is sufficiently large to balance the excess atmospheric
drag on icebergs (last two terms), or whether the atmospheric drag on icebergs and the Ekman acceleration on the water column
are balanced by a relative drag between the ocean and the iceberg (second RHS term). With $\tau_{atm-ocea}$ that is typically
$\sim 3\times10^{-1} Nm^{-2}$ in our simulated region of interest, for THICK icebergs, the Ekman acceleration is

615        $|\text{Ekman acceleration}| = \left|\frac{\tau_{atm-ocea}}{\rho_0 D_{keel}}\right| \sim \frac{\sim 3\times10^{-1} Nm^{-2}}{(1025\,kg\,m^{-3})(\sim 350\,m)} \sim 10^{-8}\,to\,10^{-7}\,m\,s^{-2}$         (23)

We point out that, following equation (2), both the acceleration due to wind drag and the acceleration due to ocean drag scale
proportionally to their corresponding effective coefficient of form drag $C_{drag}$. The latter has units of a characteristic inverse
length scale describing the iceberg $C_{drag} \propto 1/L_{iceberg}$ (see equation 2). In contrast, the Coriolis component of acceleration is
not proportional to the iceberg's characteristic lengthscale. Therefore, the ratios between the groups of acceleration terms are:



$$\frac{\left|C_{drag-atm}\vec{u}_{atmos}|\vec{u}_{atmos}|\right|}{|f\hat{k}\times\Delta\vec{u}|} \propto \frac{C_{drag-atm}|\vec{u}_{atmos}|^2}{f|\Delta\vec{u}|} \propto \frac{(|\vec{u}_{atmos}|^2)}{L_{iceberg}\,f|\Delta\vec{u}|}$$

$$\frac{|Ekman|}{|f\hat{k}\times\Delta\vec{u}|} \propto \frac{\tau_{atm-ocean}}{\rho_0 D_{keel}f|\Delta\vec{u}|}$$

$$\frac{|f\hat{k}\times\Delta\vec{u}|}{C_{drag-ocean}\Delta\vec{u}|\Delta\vec{u}|} \propto \frac{f/|\Delta\vec{u}|}{C_{drag-ocean}} \propto L_{iceberg}\,f/|\Delta\vec{u}| \tag{24}$$

However, these effective coefficients for the atmospheric and oceanic form drag (Wagner et al., 2022) on icebergs are dependent not only on the iceberg dimensions ($\sim L_{iceberg}$), but are also proportional to the density of the respective ambient fluid: water or air. Hence, the coefficient for ocean drag is much larger than the coefficient of atmospheric drag, while the characteristic lengthscales $\frac{1}{C_{drag-ocean}} < \frac{1}{C_{drag-atm}}$. For example, in our LONG simulation with THICK icebergs, direct output from the model shows that $\frac{1}{C_{drag-ocean}} \sim 10^4$ m, while the atmospheric drag coefficient $\frac{1}{C_{drag-atm}}$ is on the order of $10^7$ m, consistent with a 1:1000 density ratio between air and water. In our region, the southeastern Amundsen Sea, $f \sim 1.4 \times 10^{-4}s^{-1}$, and the surface wind speed $|\vec{u}_{atmos}|$ is typically on the order of $\sim 1\ to\ 10\ ms^{-1}$.

The scaling of terms (24) implies that if ocean drag were to match to the Coriolis force, the speed of icebergs relative to the ambient currents would have to reach unrealistically high values $|\Delta\vec{u}| \sim \frac{|f|}{C_{drag-ocea}} \sim 1\ ms^{-1}$. On the other hand, the set of proportionalities (24) implies that a relative iceberg-ocean speed

$$|\Delta\vec{u}| \sim \frac{|\vec{u}_{atmos}|^2}{1000} \sim 1\ cm\ s^{-1} \tag{25}$$

is enough for an appropriately oriented Coriolis force to balance the direct atmospheric drag on large tabular icebergs (order $10^{-6}\ m\ s^{-2}$) and also the smaller but nonnegligible Ekman term (up to order $10^{-7}\ m\ s^{-2}$). Assuming that this balance holds, the ocean drag that results from $\Delta\vec{u}$ can be estimated to be

$$C_{drag-ocean}\Delta\vec{u}|\Delta\vec{u}| \sim 10^{-8}\ to\ 10^{-7}\ m\ s^{-2} \tag{26}$$

consistent with our findings (Figure 6). This suggests that for large tabular icebergs, the fictitious Coriolis force is sufficient *in magnitude* to balance the excess atmospheric drag while the relative iceberg-ocean drag remains small (Figure 6) in agreement with our results for THICK icebergs. Notice that the iceberg-ocean drag is similar in magnitude to the impact of Ekman drift. Therefore, we are able to recover the force balance of large THICK icebergs even in the presence of small but nonnegligible ageostrophic Ekman flow.

In addition, the scaling proportionalities (24) explain the orientation of the force vectors needed to achieve balanced motion. Only the Coriolis force is large enough to balance the excess atmospheric drag on icebergs relative to the water column. However, for that to happen, the iceberg's motion relative to the ocean has to be oriented at 90° to the left of the atmospheric drag, so that the Coriolis force can oppose the atmospheric drag. In this case, ocean drag would act against the direction of motion relative to the ocean. In the case of freely-floating THICK icebergs, the icebergs' motion is on average at an angle of 63° relative to the wind direction, which means that the Coriolis force balances almost 80% of the acceleration due to wind-stress (Figure 6f). The remaining 20% are balanced by the relatively smaller ocean drag, which is indeed oriented against the direction of motion (Figure 6f), as predicted on theoretical grounds.

The Coriolis force forms part of the dominant balance and steers motion in a direction nearly perpendicular to the driving force due to winds (Wagner et al., 2017). As a result, most of the time, the orthogonal wind stress force does not do significant work along the pathway of motion of large tabular icebergs. The important Coriolis effect also projects onto a small net acceleration to the left of the iceberg direction of motion (Figure 6), which differs from the Wagner et al. (2017) assumption of zero net acceleration but still does very little work along the iceberg pathway.

We can contrast the above results for THICK icebergs with our simulated MEDIUM icebergs, where the iceberg-ocean relative speed and the associated ocean drag play a more prominent role (Compare Figure 7 for MEDIUM and 6 for THICK icebergs). The ratio between ocean drag and the Coriolis force is 2/7 for THICK icebergs (Figure 6), while it is closer to 1 for MEDIUM ones (Figure 7). At the onset of the SHORT simulations, the horizontal and vertical length scales of MEDIUM icebergs are 2.6 and 1.6 times smaller, respectively, compared to THICK ones. Consistently, we see that $\frac{1}{C_{drag-atm}}$ at the beginning of the simulations is 2.5 times smaller for MEDIUM icebergs relative to the wider THICK ones. MEDIUM icebergs experience further lateral and basal melting which causes additional reduction of their size relative to THICK icebergs.

Naturally, the Coriolis acceleration cannot change orientation and always remains directed to the left of the motion even as the balance of forces on MEDIUM icebergs is shifted. At the same time, for our MEDIUM icebergs, the atmospheric drag deviates from the orthogonality to the iceberg trajectory and projects more strongly along the axis of motion (Figure 7). Thus, atmospheric drag seems to drive a more pronounced forward motion for MEDIUM icebergs than for THICK ones during the course of the simulation. This in turn is matched by the enhanced ocean drag deceleration.

Both for THICK and MEDIUM freely-floating icebergs, the pressure gradient force is oriented mostly to the right of the direction of motion, but with larger and more noticeable deviations for MEDIUM icebergs (Compare Figure 6 and Figure 7). We can invoke a different line of reasoning to explain the orientation of the pressure gradient force. We expect on the aforementioned theoretical grounds (scaling arguments 24, and equations 25-26) that THICK large tabular floating icebergs move with velocities close to the ambient ocean currents (see also Wagner et al., 2017) in terms of both magnitude and direction. In our study, this notion is confirmed by the smaller ocean drag (Figure 6d), which suggests a small relative velocity between icebergs and the ambient water. This implies that the relationship between ocean pressure gradients and iceberg motion reflects the underlying relationship between ocean flow and pressure gradients. The consistent orientation of the ocean pressure gradient force nearly orthogonal to the direction of motion of THICK tabular icebergs (interpret Figure 6b) can be interpreted as an indication that geostrophic transport is a dominant component of the ambient ocean flow in the region of our simulated icebergs, when vertically averaging over the THICK keel depth. As in the case of wind-induced acceleration, the ocean pressure gradient force is mostly perpendicular to the iceberg trajectory and therefore does little work on the iceberg.

MEDIUM icebergs are released in the same region as THICK ones and over the same simulated historical period. So we expect that the ocean flow has a similarly important geostrophic component. However, in the case of MEDIUM icebergs (see Figure 7), we have a larger superimposed iceberg-to-ocean relative velocity. Therefore, the net direction of iceberg motion force is less clearly orthogonal to the ocean pressure gradient force. The second major difference between MEDIUM and THICK icebergs is that for MEDIUM ones, a larger fraction of their shorter keel is within the Ekman layer, and hence the ocean drag on MEDIUM icebergs is more strongly influenced by the ageostrophic Ekman spiral in the upper ocean. This contributes to further deviations of the iceberg motion away from the direction of the water column's predominantly geostrophic flow, which otherwise would have been purely orthogonal to the iceberg's trajectory. The resulting nonzero projection of the ocean pressure gradient force onto the MEDIUM iceberg trajectories means that this force does work on the icebergs.

We furthermore explore the background conditions that give rise to the ocean pressure gradient force. The pre-existing NEMO iceberg module represented the pressure gradients as a function of SSH alone. We analyse our new simulations with the updated algorithms. Our results show that under realistic historical initial and boundary conditions, horizontal density gradients in the ocean exert a noticeable impact on horizontal spatial variability in pressure (Figure 7 and Figure 8) and hence iceberg accelerations in the Amundsen Sea. For MEDIUM ones, horizontal density gradients contribute a smaller component of the full pressure gradient force. This decomposition of pressure, however, is averaged over multiple icebergs and timesteps. The orientation and dominance of the density and SSH gradients varies across timesteps of the simulation and individual icebergs.

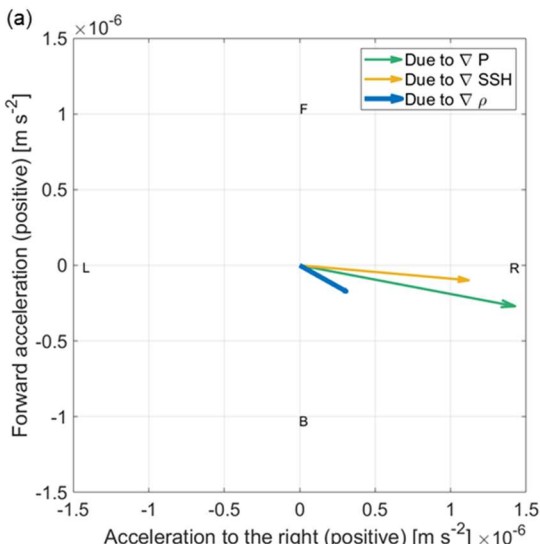
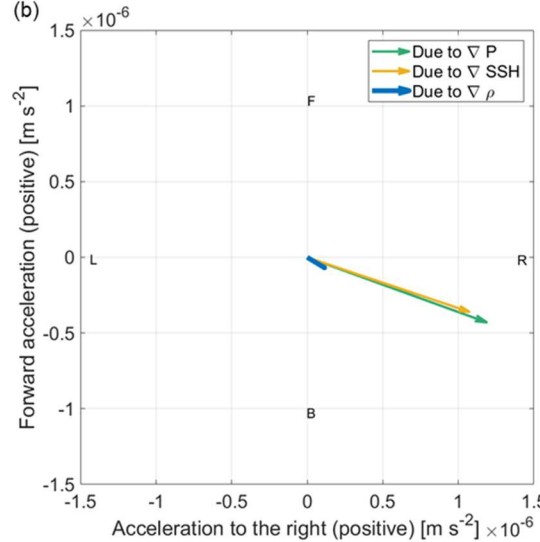

Figure 8: (a) same as Figure 6f but for the contributions of SSH and density gradients to the acceleration due to pressure gradients acting on THICK freely floating icebergs in the box where they were launched (Fig. 1), away from shallow topography and ice-shelves and (b) as in a but for freely floating MEDIUM icebergs.

## 4.2 Force balance of grounded icebergs

In the real Amundsen Sea, a fraction of the freely floating icebergs end up grounding along Bear Ridge. We now explore the force balance of grounded icebergs. In order to better understand the iceberg-bathymetry interactions, we first run a short NONDISSIPATIVE simulation without sediment resistance or solid-body friction along the bottom. In that case, icebergs whose keels reach the bottom topography are only subject to the force of gravity that pushes them down the slope. If the westward driving accelerations up the eastern flank of Bear Ridge are not large enough to overcome gravity, the icebergs slide eastward down the slope until they resume free flotation. The Coriolis force curves the trajectories of these eastward-moving icebergs northward and then westward, once again pushing them up the slope. This gives rise to a continuous repetitive motion of icebergs up and down the eastern slope of Bear Ridge but also a net northward motion of the bergs parallel to the Ridge (Figure 9). The shape traced out by the iceberg trajectory looks like a stretched-out spiral. Interestingly, such northward displacement of icebergs is indeed observed along Bear Ridge, although not in this unrealistic continuous fashion but instead with intermittent iceberg grounding (see Supplementary Movie).

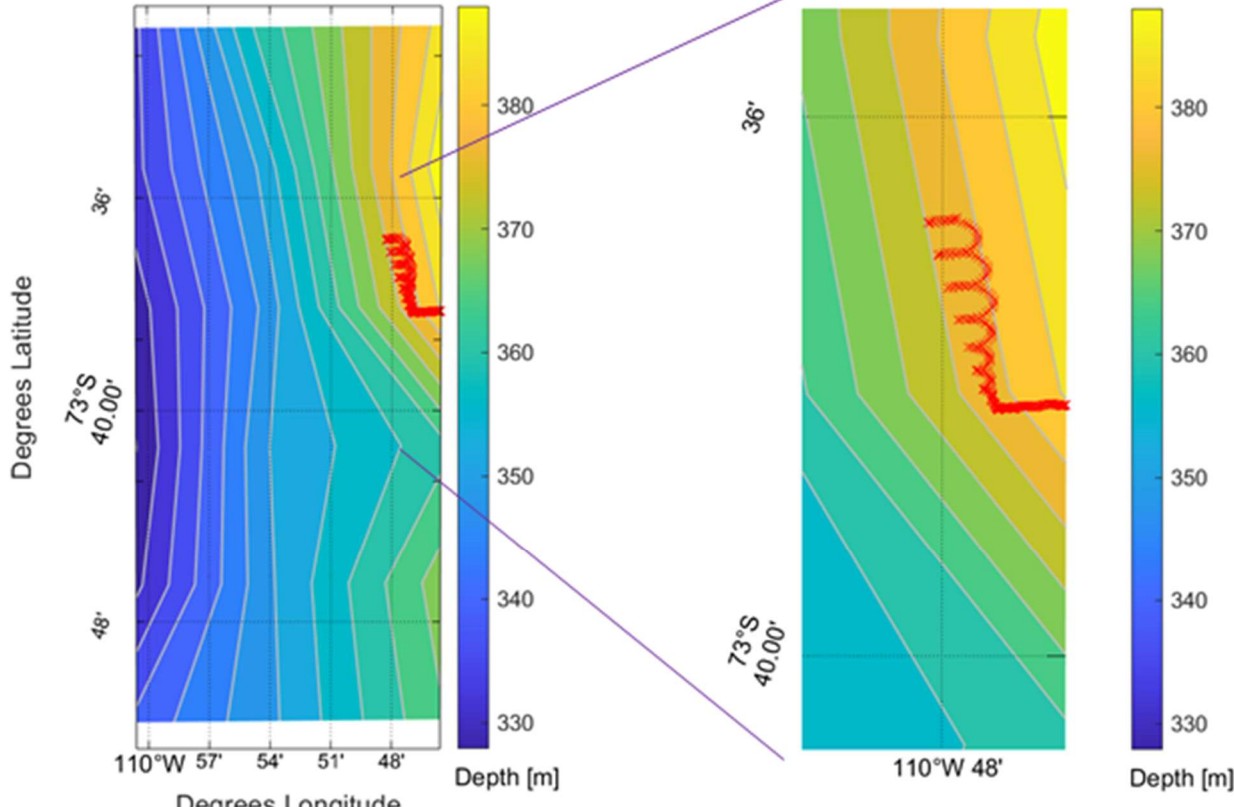

723

**Figure 9: Simulated motion of an iceberg along Bear Ridge in an Amundsen Sea configuration of NEMO without sediment resistance or solid-body friction along the bottom. The topography, with contours 4 m apart in elevation, and the superimposed iceberg trajectory are shown in the Mollweide projection. The right panel shows the enlarged iceberg trajectory. Most of the iceberg trajectory corresponds to a state of free flotation, and the iceberg's vertical displacement along the topographic slope is only on the order of centimeters.**

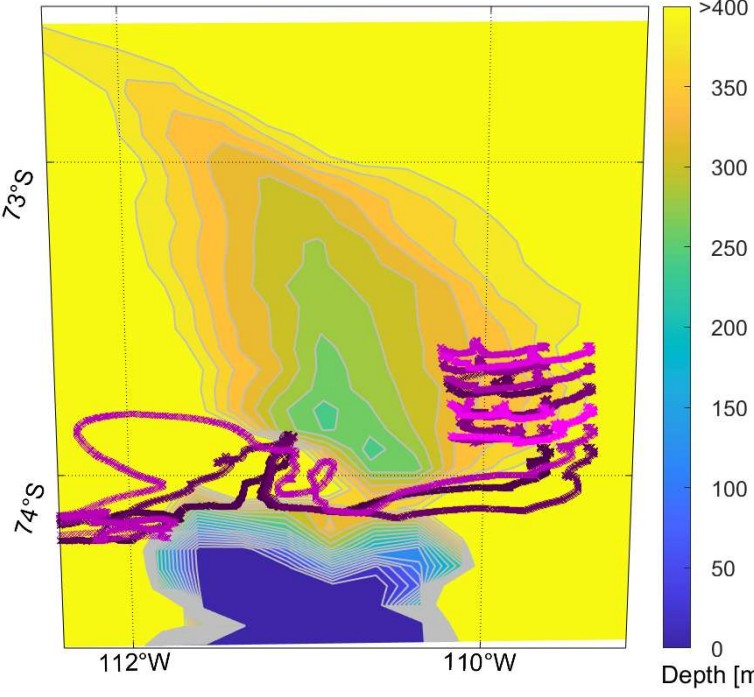

729

**Figure 10: Maps showing 14 iceberg trajectories at 20 minute time-intervals (different shades of purple corresponding to individual icebergs) over the four-year LONG simulation with THICK icebergs, where the colorbar indicates the seafloor depth along Bear Ridge highlighting regions shallower than 400 m. All icebergs in this test are launched east of 109.9° W and north of 74° S.**

The second scenario we explore is that along Bear Ridge we impose a uniform 8-m deep layer of sediment with properties observed in the Amundsen Sea, as described in Section 2. We explore this scenario in our Amundsen Sea Simulation

LONG. In the LONG simulation with 14 THICK icebergs, three are advected southward and avoid Bear Ridge (Figure 10). A total of 11 icebergs ground on the eastern side of Bear Ridge. In contrast, the keels of MEDIUM icebergs do not extend deep enough to ground along the bottom.

In contrast to our NONDISSIPATIVE simulation, in experiment LONG, we have THICK icebergs grounded on Bear Ridge that are subject to sediment resistance. The latter counteracts forces that may otherwise accelerate the icebergs. Occasionally, icebergs also come into contact with the basement and experience Coulomb friction. In this scenario, we do not see a clear northward displacement of the icebergs as they ground and unground. Instead, they come to a stop quite soon after their keels start ploughing scours into the seafloor sediment. The ploughed scours are mostly oriented perpendicular to Bear Ridge and along the zonal direction (Figure 11). The icebergs grounded in the seafloor sediment melt in place, which thins them. In addition, simulated icebergs sporadically receive a strong enough push by the ocean and/or the atmosphere, which allows them to resume their motion briefly and intermittently and to plough scours deeper into the soft sediment (Figure 11b). These ploughing episodes are marked by sporadic short-lived thrusts of acceleration into the sediment (Figure 11) while, most of the time, the icebergs actually remain static when they are embedded in the sediment. This combination of processes creates scours whose depth relative to the surrounding ocean bottom is not uniform or changing monotonically along the scour length but is marked by vertical undulations (Figure 11a), which is reminiscent of the complexity of the observed scours (Figure 2).

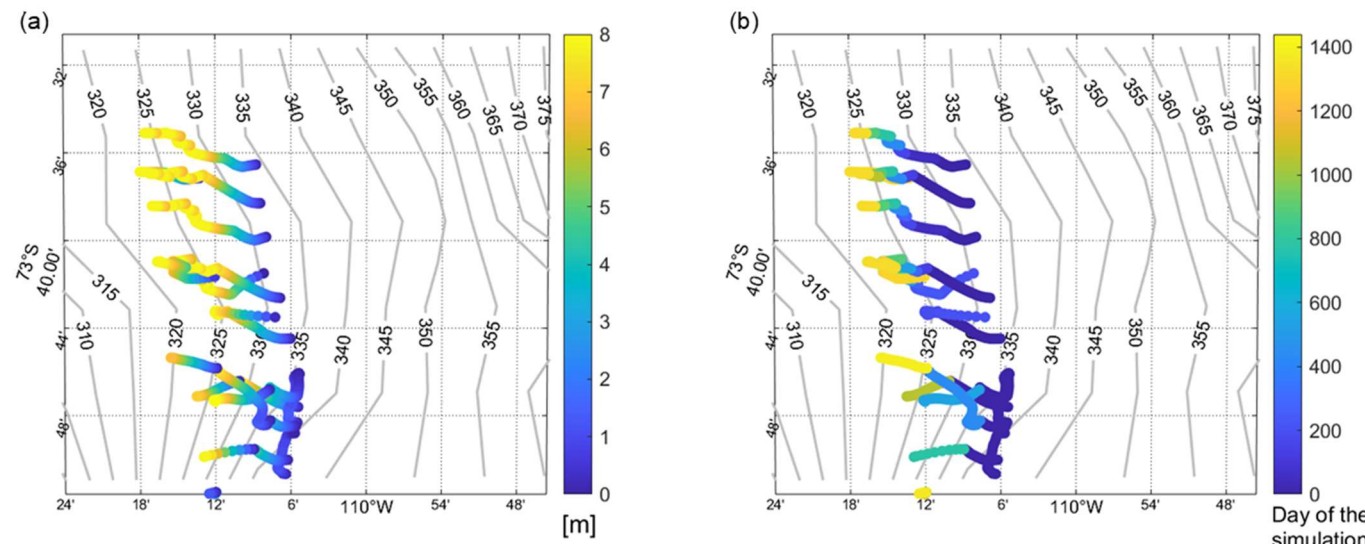

**Figure 11: Simulated scours along Bear Ridge from the LONG simulation with THICK icebergs. Panel a shows scour depth [m] and b shows the temporal evolution of the scours [days of the simulation].**

During the episodes of kinetic grounding, the dominant forces are the sediment resistance, which decelerates the icebergs, and atmospheric drag, which tries to push them forward (Figure 12). The ocean drag also makes a smaller relative contribution towards slowing down kinetically grounded icebergs that plough sediment scours. In the case of kinetic grounding, the force due to ocean drag is much weaker than the pressure gradient force and the Coriolis force. Moreover, the ocean drag does not exhibit a clear prevailing orientation against the direction of motion (Figure 12), and the ocean pressure gradients are not purely orthogonal to the iceberg trajectories. Instead of pointing mostly to the right, as is the case for freely-floating icebergs, the pressure gradient force on kinetically grounded icebergs can point in any of multiple directions. This can be compared with the very clear decelerating effect of ocean drag on freely floating icebergs (Figure 6d). The difference in the relative orientation of the sea-ice drag is noticeable, as well (Figure C1 in Appendix C). When the icebergs are grounded, sea-ice drag acts to push them forward rather than decelerate them or act in a direction orthogonal to their trajectory (Figure C1). Even more importantly, kinetically grounded icebergs experience a large forward push by atmospheric drag. However, the net acceleration, which is also positive, is smaller (Figure 12) due to the large dissipative forces (Figure 13 b,c) where sediment resistance plays a major role (Figure 13). The latter means that while in motion, these kinetically grounded icebergs moving within the sediment and along the bottom topography are rarely in force balance. Note that here we are considering the net

acceleration as a weighted average across all four Runge Kutta stages of a given model timestep but before the 'stop' flag described in Section 3.4 is applied. Therefore, the metric in Figure 12e does not include this additional deceleration imposed on icebergs as soon as the condition in equation 21 is fulfilled at any stage of the solver.

We furthermore explore the relative contribution of different physical mechanisms to the sediment resistance force and the dominant terms that affect its magnitude (Figure 14). We consider icebergs with an assumed keel width (and hence scour width) $W=90m$. We see that for scour depths $D$ shallower than 3 m, the shear strength term that is proportional to $D$ dominates over the term due to the weight of the ploughed sediment in accordance with equation 11. In the deeper scours, the density of the ploughed sediment determines to a large extent the total resistance experienced by a grounded iceberg. However, the leading density and shear terms remain comparable for scour depths of 8 m. These results indicate that knowledge of both the sediment density and the shear strength are necessary for an accurate estimate of the sediment resistance experienced by grounding icebergs. On the other hand, the third component of equation (11), which is independent of keel width, remains negligible for all scour depths when the scour width is $W=90m$. The scour width (which matches the keel width) must be comparable to the scour depth for all three sediment resistance terms to be characterized by the same order of magnitude, and this is not seen in the observations (Figure 2). However, such narrow scours are not resolvable in our observational dataset (Table 1) and could occur in some settings.

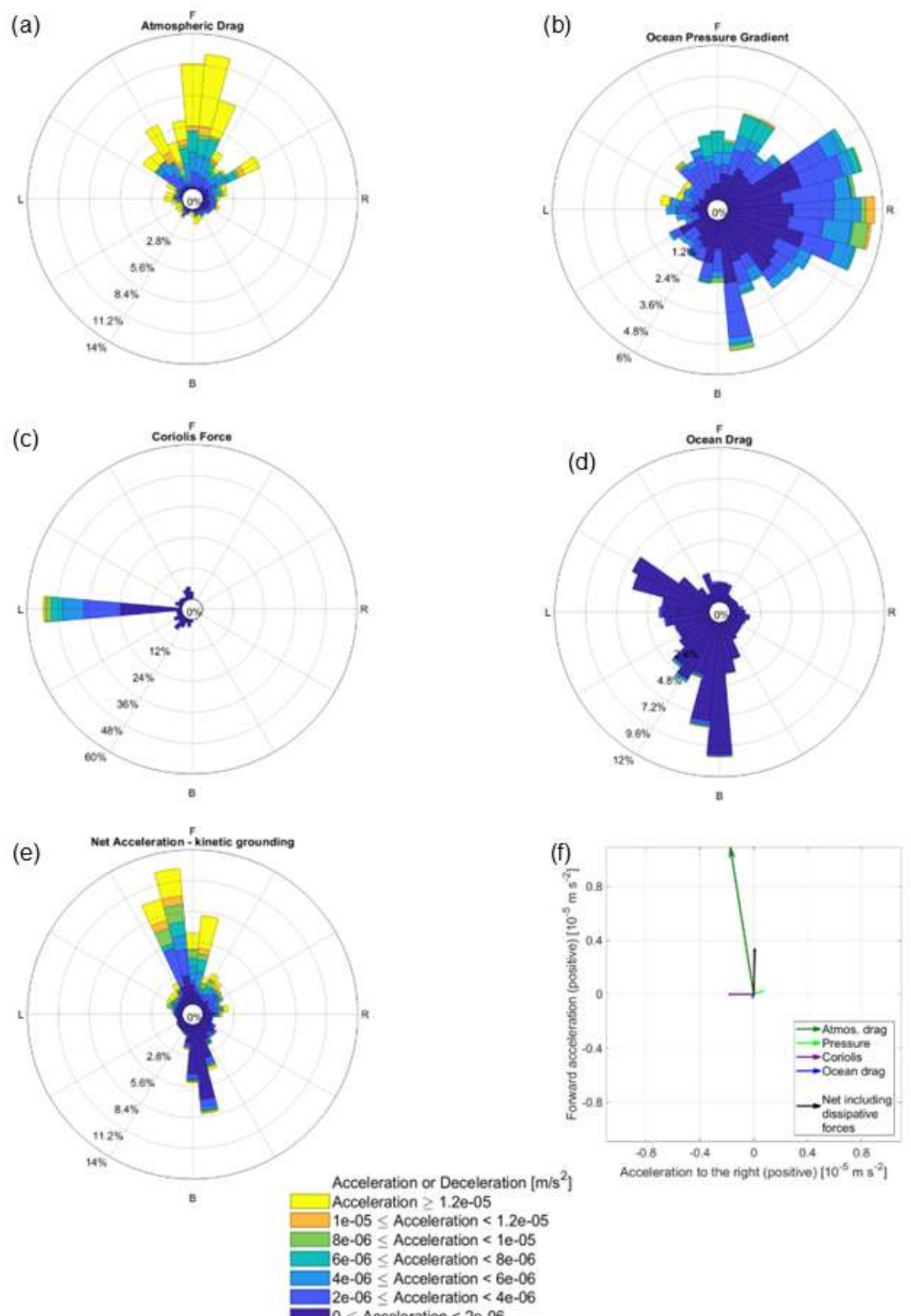

Figure 12: As Figure 6 and 7 and with the same colorscale but for the case of kinetic grounding of THICK icebergs. The net term in (e) is dominated by the dissipative forces shown in Figure 13. Occasionally, the orientation of the Coriolis force is not strictly to the left of the iceberg motion for numerical reasons: the force is calculated using the projected directions of future motion at each Runge-Kutta stage of the iceberg dynamics algorithm.

When icebergs plough all the way to the basement, they experience Coulomb friction and down-slope gravitational acceleration (Figure 13). These additional forces bring icebergs to a halt within one day of contact with the basement, and the icebergs are no longer kinetically grounded.

We draw a distinction between the term "kinetic grounding" which may still allow for motion along the bottom and
a static regime which implies no motion of the grounded iceberg. Icebergs do come to a complete stop even before their keels
plough deep enough to reach the solid basement beneath the sediment. In that case, sediment resistance balances the net
potential drivers of iceberg motion.
When the iceberg keels plough deep into the sediment and eventually reach the basement beneath, they start to feel
the effect of Coulomb friction and gravity. Our chosen kinetic Coulomb coefficient value of $\mu = 0.002$ is large enough to
bring icebergs rapidly to a halt once they reach the basement along Bear Ridge. In our simulation, we find that when the
icebergs reach the basement, they come to a complete stop within the same day and enter a static regime. In this regime, to a
first order, static Coulomb friction ($\mu_{stat} = 0.2$) balances the gravitational force that tries to pull the icebergs down the slope
(Figure 15). When the icebergs are statically grounded on the solid basement, Coulomb friction dominates, while sediment
resistance and other forces play a less prominent role (compare the orders of magnitude in Figures 13 and 15).
During the first year of the LONG simulation with THICK icebergs, the curvature term $\vec{a}_{cuvature}$ in each horizontal
direction only has a nonnegligible magnitude ($> 1 \times 10^{-6}$ m s$^{-2}$) over less than 23 hours. During the times when the
geometric curvature term is greater than $1 \times 10^{-6}$ m s$^{-2}$ in magnitude, solid-body friction and gravitational forces dominate.
We also explore a range of values for the Coulomb coefficient $\mu$, and they also give qualitatively similar results (Appendix B).
**4.3 Comparison with observations**
We next consider simulated iceberg residence times along Bear Ridge and qualitatively compare them with observations of
the real Amundsen Sea. This allows us to test the fidelity of our new algorithms and the validity of the assumptions we have
made regarding sediment and scour properties along Bear Ridge. We furthermore compare our new results with the output of
a pre-existing iceberg grounding algorithm in NEMO.

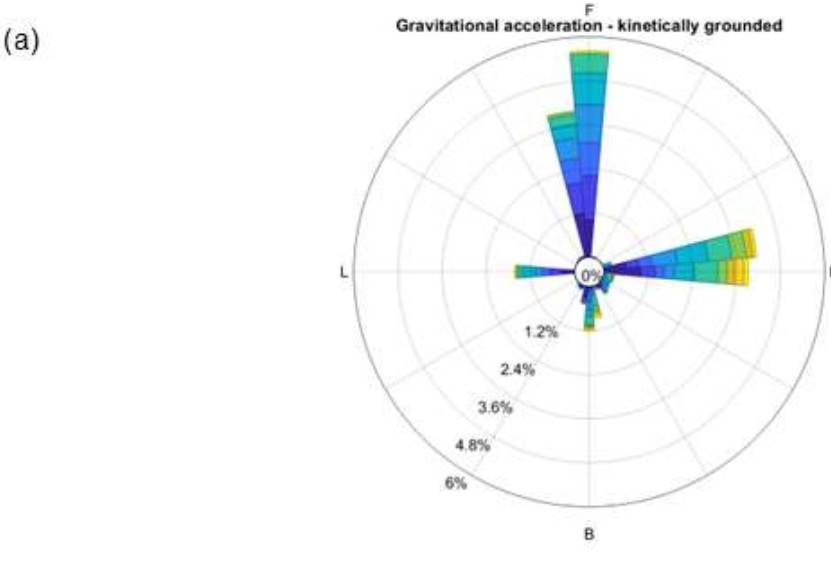

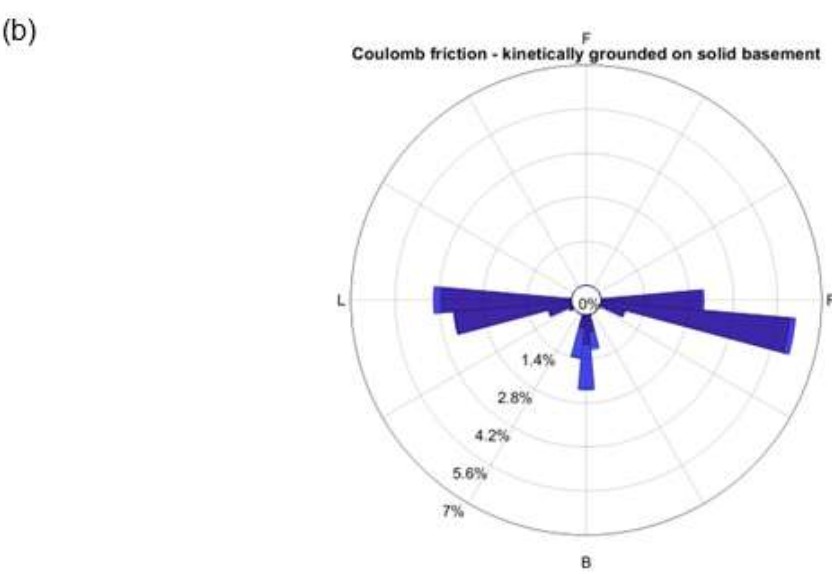

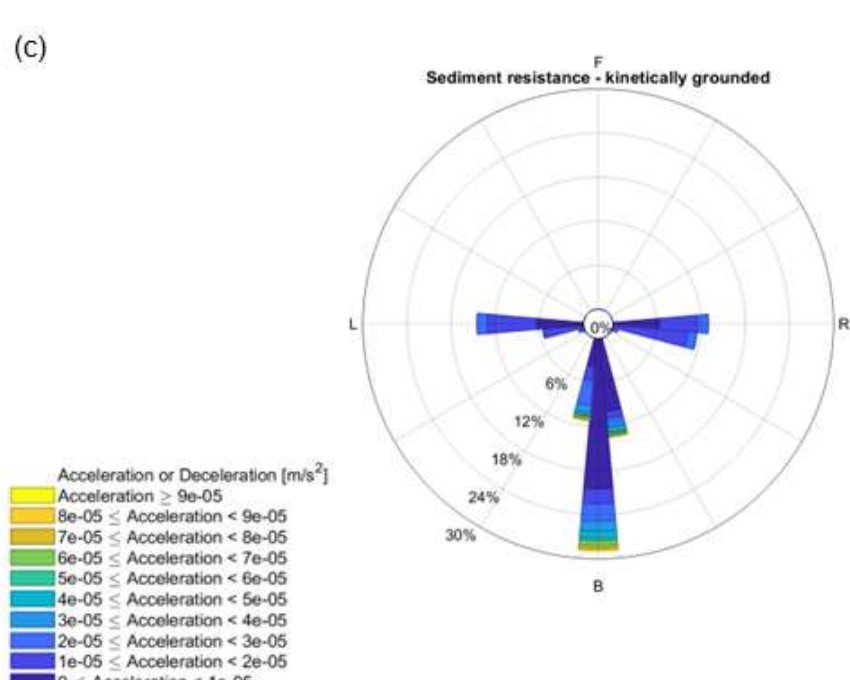

**Figure 13: As Figure 12 but for the gravitational acceleration and the deceleration due to dissipative forces in the case of kinetic grounding of THICK icebergs: (a) gravitational acceleration present in a small fraction of the iceberg timestep samples (b), the Coulomb solid-body frictional deceleration (c), deceleration due to kinetic sediment resistance. Occasionally, the orientation of the deceleration due to sediment resistance force is not strictly against the iceberg motion for numerical reasons: the force is calculated using the projected directions of future motion at each Runge-Kutta stage of the iceberg dynamics algorithm.**

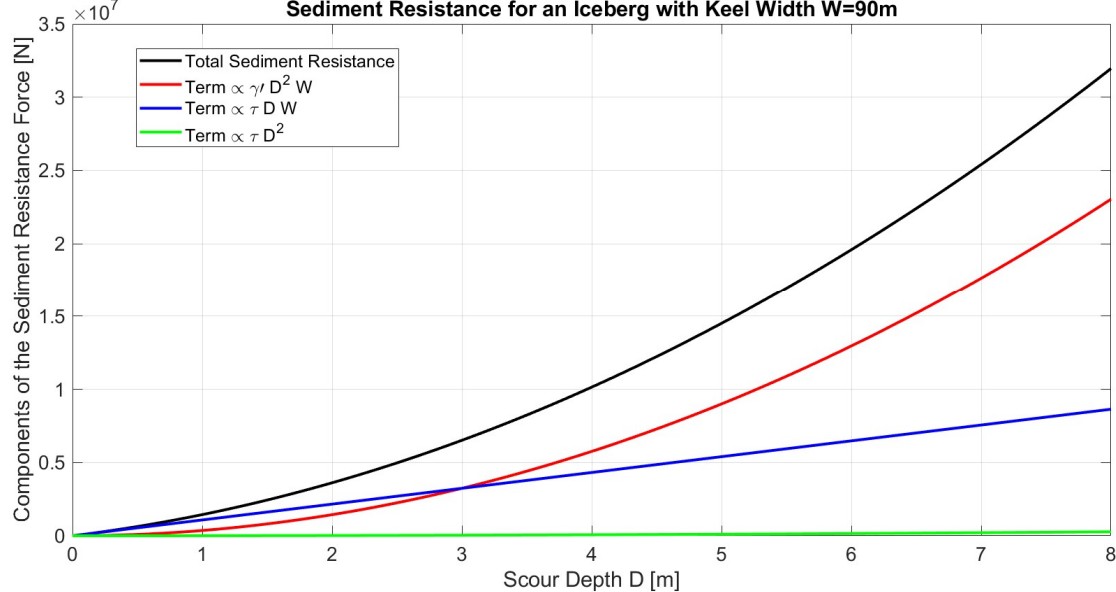

**Figure 14: Components of the sediment resistance force [N] as a function of the scour depth *D* [m] for icebergs with keel width W=90 m and sediment properties typical of the Amundsen Sea. The individual terms and their sum refer to equations 6 through 11.**

In particular, we consider a pre-existing very simple iceberg grounding scheme in which icebergs whose keels reach topography are moved back to their previous floating position and have their horizontal velocity set to zero (Olivé Abelló et al., 2025 and Figure 16a). We apply this algorithm in the SIMPLE GROUNDING simulation with THICK icebergs, and we find that bergs along Bear Ridge quickly leave the region and float away or are moved back to their previous floating position and have their horizontal velocity set to zero, consistent with the simulations of Olivé Abelló et al. (2025). If the icebergs do 'ground', that grounding is often near the ice shelf fronts. If our pressure gradient algorithm masks the ice shelves, as described in section 3.4, then we see that we avoid trapping as many icebergs near the ice shelf edges (Figure 16b). In that case most icebergs that we launch go around the northern side of the shallow Bear Ridge and quickly end up in the western Amundsen Sea, within a month. Five icebergs circumvent the shallowest part of Bear Ridge by going around its southern end. However, the SIMPLE GROUNDING scheme still remains fundamentally unable to keep the icebergs grounded along Bear Ridge for long enough (Figure 16b).

In contrast, with our new updated grounding algorithm, THICK icebergs remain trapped along Bear Ridge for years (Figure 16c). Crucially, assuming a deformable sediment layer above the basement is what allows the icebergs to remain grounded along Bear Ridge (Figure 16 and Appendix B). This stands in contrast to the rapid northward export of icebergs in the scenario without seafloor sediments or solid-body friction (compare Figures 9 against 10, 11 and 16). While icebergs in the real Amundsen Sea do propagate northward, they also remain grounded for long periods of time suggesting that frictional forces are an important consideration and should be retained in the model. Our results are in qualitative agreement with the observations suggesting that thick icebergs may be trapped on Bear Ridge for months or years.

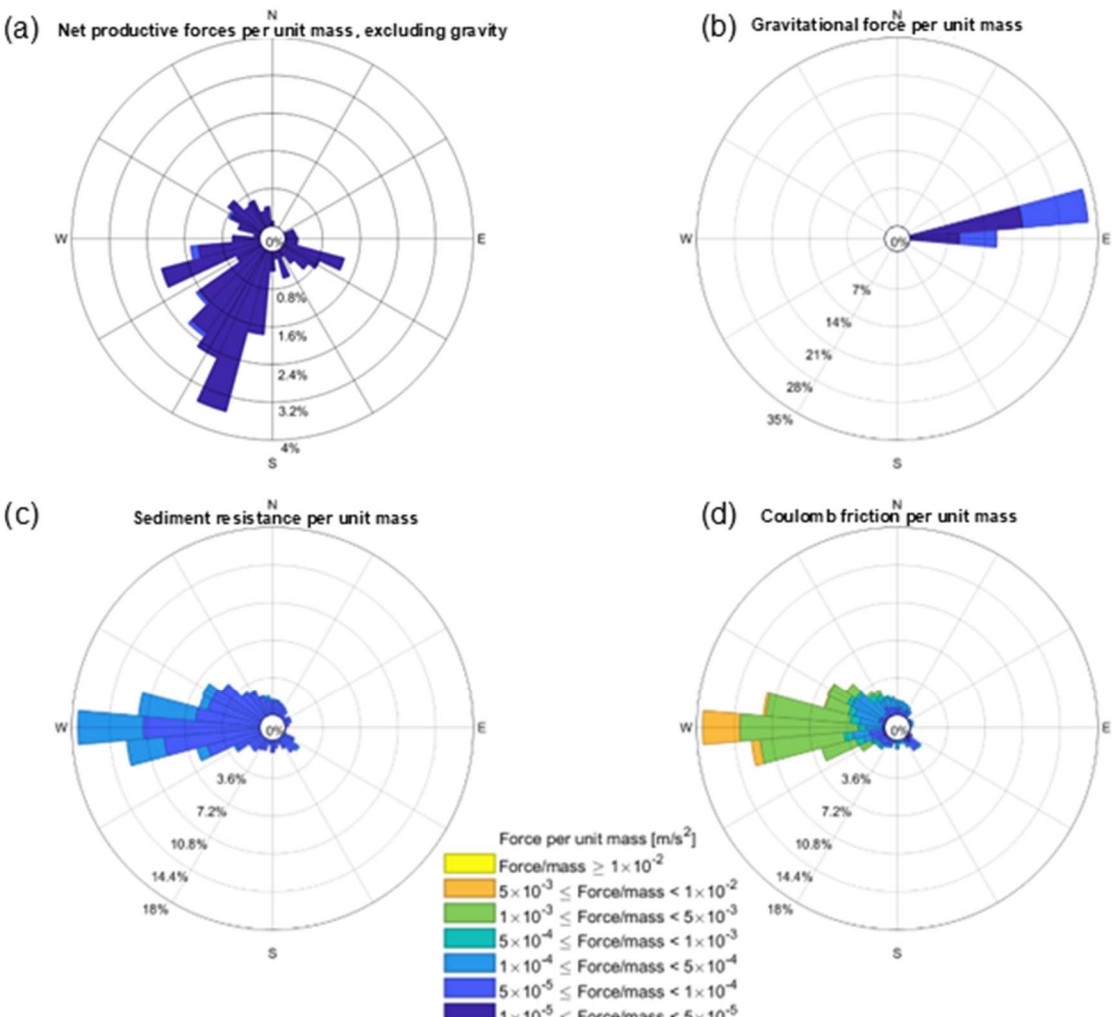

847

Figure 15: As Figure 12 and 13 but in the case of static grounding of THICK icebergs on solid basement with (a) showing the net productive force per unit mass excluding gravity, (b) the gravitational force per unit mass, (c) the sediment resistance per unit mass and (d) static Coulomb friction in the case of static grounding of THICK icebergs. The scale is non-linear and shows the order of magnitude. Note that unlike Figures 12-13, the directions are here the cardinal points as the icebergs do not move. In order to show meaningful geographical directions, here we include only icebergs grounded on Bear Ridge and nowhere else. In the case of grounding along the basement, the icebergs in the simulation always come to rest and enter a static regime within a day. The static Coulomb frictional force per unit mass shown here is the maximum achievable magnitude, but the effective friction does not exceed the sum of the other sources of acceleration, which are dominated by gravity.



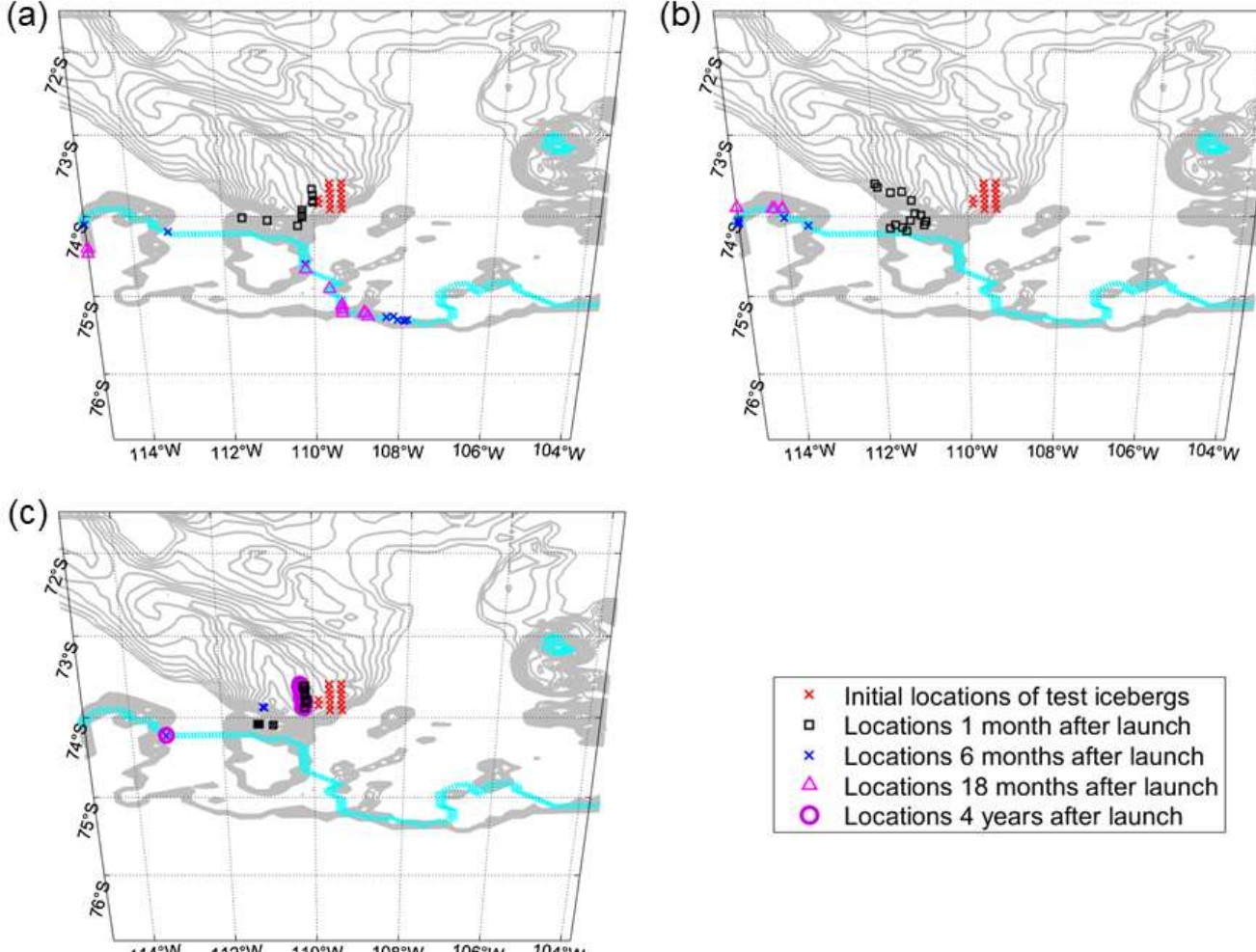

**Figure 16: Locations of THICK icebergs during the course of the (a) pre-existing SIMPLE GROUNDING; (b) the pre-existing**
**SIMPLE GROUNDING scheme and with masked pressure gradient forces along the iceshelf edge; and (c) LONG dissipative**
**simulation with the new grounding scheme. The topography shallower than 560 m is superimposed with gray contours, 20 m apart.**
**The blue line denotes the ice shelf fronts, where many icebergs come to a stop under the pre-existing SIMPLE GROUNDING scheme.**
**Iceberg locations 4 years after launch are shown for the simulation with new grounding scheme only.**

864        While the transition between static and kinematic grounding and scouring is determined by basal melt rates and winds,

the full ungrounding of icebergs is a much slower process. In the real ocean, grounded icebergs may capsize if their horizontal
dimensions shrink faster than the decrease in their vertical thickness and their new geometric orientation becomes unstable. In
order to explore the potential for capsizing, we have extended our four-year LONG simulation with THICK icebergs up to 20
years by applying a repeat cycle of the same four-year surface boundary conditions. We see that in our extended LONG
simulation, the rate of lateral melting indeed exceeds the rate of basal melting, with a potentially unrealistically large
contribution of wave erosion to the former. One may expect that this causes widespread capsizing of grounded icebergs.
However, the pre-existing NEMO capsizing criterion incorrectly compares horizontal length rather than horizontal width
against vertical thickness. As a result, the excessive lateral melting of individual grounded icebergs eventually reduces the
horizontal width to zero within 17 years and completely destroys the iceberg before the horizontal length decreases enough for
the iceberg to capsize (not shown). Olivé Abelló et al. (2025) correct the unphysical capsizing criterion in NEMO, a change
that will be implemented in new model configurations. On the other hand, eliminating the relevant bias in the horizontal wave
erosion remains an important outstanding issue left to future studies.

## 5 Summary and conclusions

We have introduced a new and improved representation of iceberg dynamics and iceberg grounding capability in NEMO. These updates to the model are verified using idealized test configurations before being applied to an eddy-permitting regional simulation of the Amundsen Sea forced with historical boundary conditions. We note that in this initial study our specific intention is to develop the new model physics, and so we delay a full multi-year realistic simulation of iceberg calving and grounding to a subsequent study. A companion paper presents an improved iceberg thickness distribution in NEMO based on the thickness of the Antarctic ice-shelves from which icebergs calve (Olivé Abelló et al., 2025). The companion paper also introduces an objective definition of iceberg size classes implemented in NEMO (Olivé Abelló et al., 2025). Future work will combine these two approaches. Once we are in possession of a full realistic simulation, a much fuller comparison to observed iceberg grounding can take place, with additional tuning of the iceberg grounding parameters. This would require the application of advanced techniques for tracking iceberg grounding episodes from satellite imagery, perhaps involving artificial intelligence (AI) techniques. A full comparison of simulated and observed iceberg scours would also be extremely powerful in constraining the model. In addition, future radar sounding observations may reveal the distribution of iceberg keel shapes.

The Amundsen Sea experiments suggest that in this region the ocean pressure gradients acting on icebergs are not determined solely by spatial variability in SSH. Instead, our results highlight the important direct contribution of horizontal gradients in the ocean density, which are separate from the contribution of density through steric SSH variability. In addition, we show that large freely drifting tabular icebergs often enter a dynamical regime characterized as a balance between surface winds and the Coriolis force. In contrast, smaller icebergs experience stronger turbulent ocean drag. Overall, freely floating tabular icebergs in the Amundsen Sea are not subject to strong net acceleration along or against their direction of motion, but tend to get deflected to the left by Coriolis force. This is consistent with the theoretical arguments presented by Wagner et al. (2017) who suggest a similar force balance for large tabular icebergs when the ocean is in a purely geostrophic regime. In our analysis, we derive more general scaling arguments that describe the momentum budget of large and medium icebergs even in the presence of ageostrophic Ekman flow.

Our results for the force balance of freely floating icebergs also highlight the potential dynamical importance of iceberg fragmentation. If large tabular icebergs get fragmented, the smaller fragments will get advected in a different direction compared to a parent iceberg. Under identical oceanic and atmospheric conditions, large and small icebergs originating from the same place drift towards and melt in different geographical locations. England et al. (2022) highlight this effect by directly imposing the Wagner et al. (2017) analytical result on simulated icebergs. Furthermore, our findings for the dependence of the turbulent iceberg-ocean drag on iceberg size have implications for the rate of iceberg subsurface melting. Jenkins et al. (2010) and Davis et al. (2023) point out that the rate of ice shelf basal melting depends on the velocity beneath the shelf. By analogy, the rate of iceberg subsurface melting, especially for large tabular icebergs, exhibits a similar dependence on the horizontal length scale through the latter's impact on the relative velocity. Note that the formulas in NEMO explicitly represent the direct effect of iceberg lengthscale on basal melting through a -0.2 power law, while the potentially large impact of horizontal size via changes in relative velocity is an emergent phenomenon. This once again points to the importance of the fragmentation of large icebergs into smaller ones, a first order process, which is not currently represented in NEMO but is implemented in other models such as GFDL (Huth et al., 2022).

Our updated algorithms show the force balance of icebergs interacting with the ocean bottom. The new grounding representation in NEMO allows us to reproduce realistic bottom scours and grounding along Bear Ridge in the Amundsen Sea. We compare our simulations against observations in the region. Grounding in deformable sediments is marked by alternating periods of slowdown and arrest followed by episodic acceleration that thrusts the icebergs deeper into the sediment (Figure 11). The sediment resistance force is characterized by comparable contributions due to the density and the shear strength of the deformable layer. Consistent with observations, the presence of a sediment layer in our simulations allows thick icebergs to remain trapped along Bear Ridge on a timescale of years (Figure 16c) despite their susceptibility to basal and lateral melting.

The long residence time of icebergs along the Ridge and their eventual northward motion affect the distribution of iceberg
meltwater (Bett et al., 2020) but also, more importantly, create the Amundsen Sea Polynya and the dipole in the distribution
of sea ice growth on either side of Bear Ridge (Figure 1). In contrast, in the absence of sediment resistance and solid-body
friction with the basement, icebergs would continuously move northward along the eastern slope of Bear Ridge.
The model algorithms in this study will allow us to explore the existence of possible feedback mechanisms between
iceberg grounding along Bear Ridge in the Amundsen Sea and the calving of new icebergs from the shelf under changing
climate conditions, as suggested by Bett et al. (2020). Ultimately, these processes may be more accurately represented
circumpolarly, with consequent improvements in regional ocean modelling, particularly over the continental shelves. This is
a region that models struggle to accurately recreate, notably the interactions between melt and sea ice distribution, polynyas,
dense shelf water formation and wider ocean vertical circulation (e.g. Aguiar et al., 2024). The strength of this mechanism
and its future response to climate change have major implications for the freshwater budget of the Amundsen Sea, for the
stability of ice shelves, and for the Antarctic contribution global sea level rise. Therefore, the proper representation of Antarctic
iceberg grounding and residence times in models is an important prerequisite for developing reliable future climate projections.

**Appendix A. Supplementary Movie**

We enclose a hyperlink (https://doi.org/10.5446/70447) to a supplementary movie composed of Copernicus Sentinel-1 SAR
(synthetic aperture radar) images taken over the time period 2017-2024, an extension of the movie provided in Bett et al.
(2020). The movie highlights the locations of grounded icebergs of various sizes along the shallow Bear Ridge. Many of the
still images in the movie also show the dipole in sea-ice concentration with larger values to the east of the Bear Ridge.

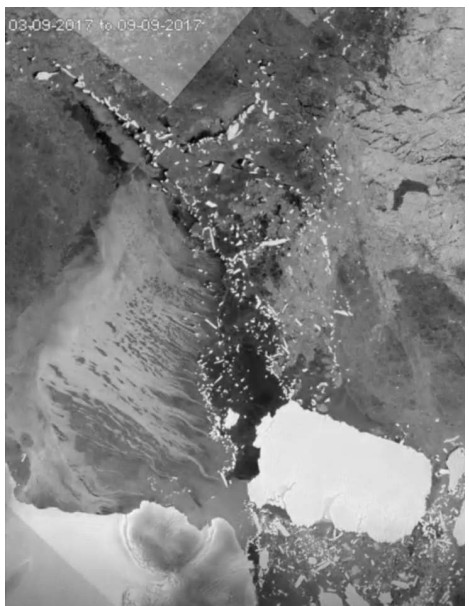

**Supplementary Movie A. Available at the TIB AV Portal (Kostov et al., 2025a) : https://doi.org/10.5446/70447**




**Appendix B. Effects of topographic curvature, substepping parameters, and the Coulomb coefficient of solid-body friction**

We explore the parameter space of possible values for the unconstrained Coulomb coefficient of solid-body friction $\mu$ across different shorter repetitions of the Amundsen Sea Simulation B that extend for more than 350 days. In each case, we vary the coefficient of kinetic friction, while the coefficient of static friction is always set to be 100 times larger than the kinetic one.

We find that varying the value of the kinetic $\mu$ and static coefficient $\mu_{stat}$ by different orders of magnitude does not affect the statistics for episodes of motionless grounding (Figure B1). We infer two conclusions from this result. First, our results are robust with respect to the unconstrained value of the Coulomb coefficient for solid body friction. Second, the duration of episodes of grounding along the bottom seem to be governed by sediment resistance and melting processes that thin the icebergs.

We furthermore test the importance of the "curvature term" from equation 17. When an iceberg moves upwards from a gentler slope to an even steeper slope, some of its horizontal momentum is reoriented in the vertical direction. This gives rise to an apparent horizontal deceleration term $\vec{a}_{cuvature}$ that is purely geometric, independent of gravity and buoyancy, unrelated to turbulent drag, and unrelated to dissipative forces such as sediment resistance or solid body friction. For example, if the topographic slope $\beta$ changes at a rate $\partial\beta/\partial x$ along the pathway of a grounded iceberg moving in the x-direction, then $u$, the x-component of the iceberg's horizontal velocity also changes as:

$$\vec{a}_{cuvature}\cdot\hat{x} = -|u^2\tan\beta\,|\frac{\partial\beta}{\partial x}\hat{x} \tag{B1}$$

and analogously for a component $\vec{a}_{cuvature}\cdot\hat{y}$ along the y-direction. The term $\vec{a}_{cuvature}$ represents the direct impact of seafloor basement geometry on horizontal velocity and can be understood through dimensional analysis. For example, in equation (B1), the factor $u^2$ has dimensions of length squared over time squared $\left[\frac{L^2}{T^2}\right]$, while the horizontal spatial derivative $\frac{\partial}{\partial x}$ has dimensions of inverse length $[L^{-1}]$. Therefore, $\vec{a}_{cuvature}$ is an acceleration term with dimensions of length over squared time $\left[\frac{L}{T^2}\right]$. Qualitatively, the tangent $\tan\beta$ of the topographic slope $\beta$ tells us how changes in the slope along the horizontal direction re-orient the velocity vector along the vertical direction. For instance, when the slope gets steeper, the iceberg's velocity vector is deflected more towards the vertical direction. Numerical tests demonstrate that the term $\vec{a}_{cuvature}$ becomes non-negligible compared to other acceleration terms when icebergs slide along realistic topography. However, this happens only when the angle of the interpolated topography changes along the iceberg's pathway. The curvature factor is non-negligible if we focus on the small-scale shape and evolution of iceberg bottom scours, but does not play a first-order role in the overall residence times of grounded icebergs on the scale of the ocean domain resolution as revealed by the tests in Section 4. We also see that the effect of topographic curvature does not contribute to qualitative differences in the shape and evolution of iceberg scours (Figure B1).

We also show that in the presence of a sediment layer above the crystalline basement, the time-stepping of the interaction with the shape of the topography becomes less important. Using or not using temporal sub-stepping does not play a first order role when icebergs ground in the sediment above the crystalline base (Figure B1).

We do find that having no Coulomb friction (Figure B1) or having neither friction, nor sediment resistance (Figure 9 in the main text) are extreme cases where the absence of processes makes a noticeable difference. If there is no solid-body friction but there is sediment resistance, icebergs plough swirls and half-loops or exhibit a repetitive back and forth motion along the slope (Figure B1). If no dissipative force acts on a grounded iceberg, then its trajectory can trace out a spiral motion (Figure 9 in the main text). Interestingly, iceberg scours in the shape of swirls and spirals are indeed observed along the ocean bottom in the real world, but they are traditionally attributed to the effect of tides (See Figure 2 and Section 2). In our model configurations, we do not represent tidal effects, yet we observe such features and point out the important role of the Coriolis force for driving oscillatory iceberg motion.

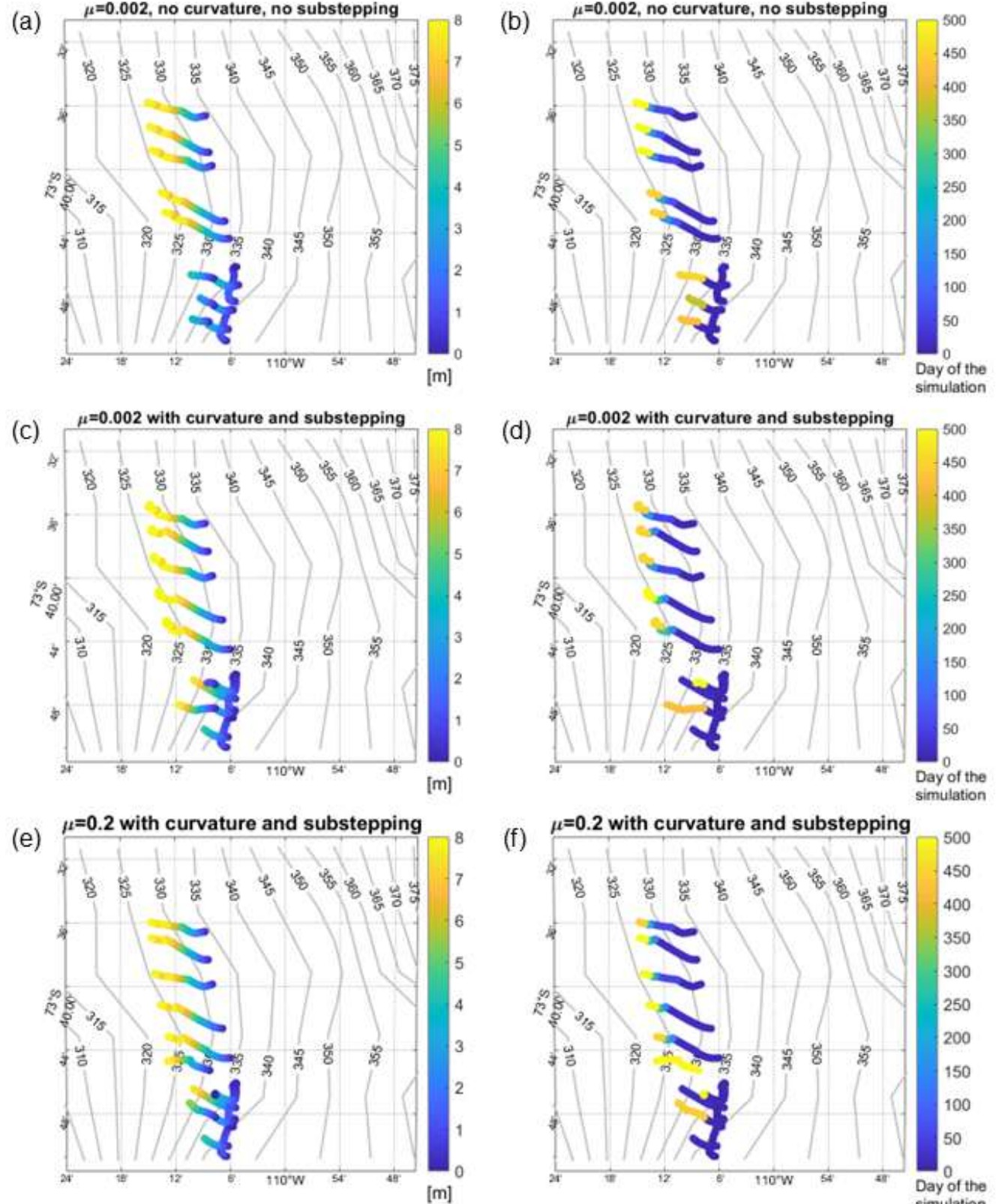

**Figure B1: As in Figure 11 but in simulations with a range of kinetic Coulomb coefficient values μ, sub-stepping choices, and representations of the topographic curvature effect.**




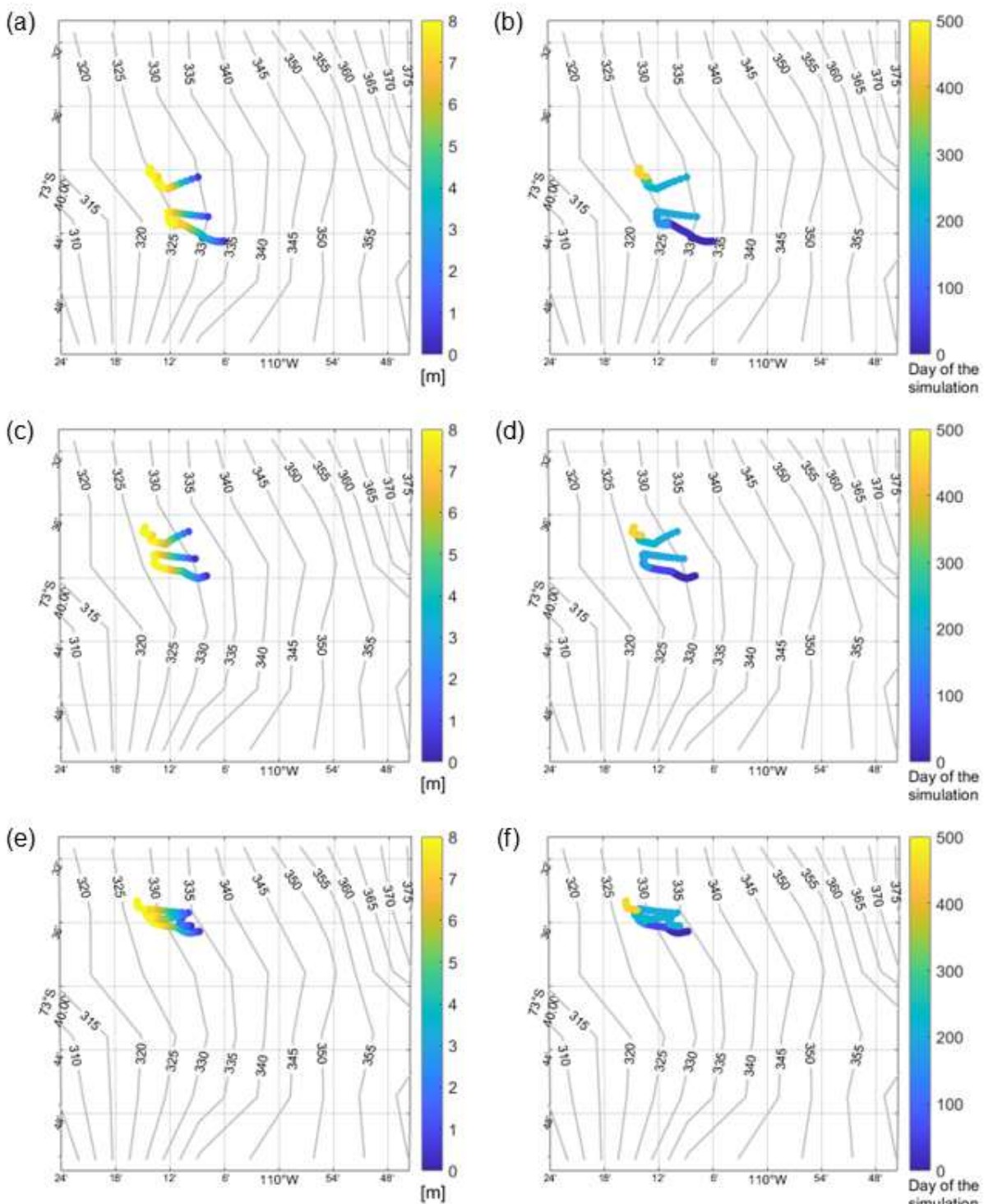


**Figure B2: As in Figure B1 but in simulations with zero Coulomb friction and 8-m sediment thickness. Each row represents an individual iceberg. The left column shows scour depths, and the right column shows their evolution in time during the LONG simulation with THICK icebergs.**

We also probe the impact of temporal substepping. In the absence of sediment resistance forces, simulating the effect of gravity on grounded icebergs does require shorter timesteps and may necessitate substepping for iceberg dynamics. Here we present an important test of the iceberg substepping and its impact on gravitational acceleration/deceleration. In this test, we release a 97-m thick iceberg in an idealized rectangular basin that is not in a rotating frame of reference (the Coriolis force is set to zero). All wind/ocean/sea ice drag terms are set to zero. Iceberg melting is switched off, and so is solid-body friction. The sediment depth along the bottom is set to zero, eliminating any sediment resistance. Under these conditions, the only term remaining in the momentum balance is the gravitational disequilibrium term $\vec{a}_{gravity}$. The iceberg is initialized with south-eastward velocity above a sloping bottom whose depth decreases by 10m per 1 km in the southward direction, going from

100m to 80m. As the iceberg moves up the bottom slope and out of free flotation, its kinetic energy is not dissipated but rather converted to potential energy. The gravity term should then convert this back into kinetic energy, accelerating the berg back down the slope, and this descent should exactly mirror the berg's ascent up the slope, like an idealized perfectly un-damped pendulum.

We first test a case where the iceberg timestep equals the ocean timestep. The iceberg travels south-eastward up the slope, decelerating under gravity, and then reverses direction to slide north-eastward, accelerated by the same term. However, when the iceberg starts sliding back down the slope, its descent is not symmetric with respect to its initial ascent, despite the absence of external driving or dissipative forces (Figure B3); there is an asymmetry in the iceberg's meridional velocity going up the slope and coming back down (Figure B3).

In our numerical tests, we found that this error is caused by an inadequate time-resolution of the iceberg deceleration and acceleration by the gravitational acceleration, which can be much larger than other sources in realistic cases. When we introduce temporal substepping within the iceberg module (four iceberg steps per one ocean step), we see a better symmetry between the iceberg's ascent and descent along the slope (Figure B3). This indicates an improved conservation of mechanical energy which is converted between kinetic and potential energy and this was found to be important in realistic cases.

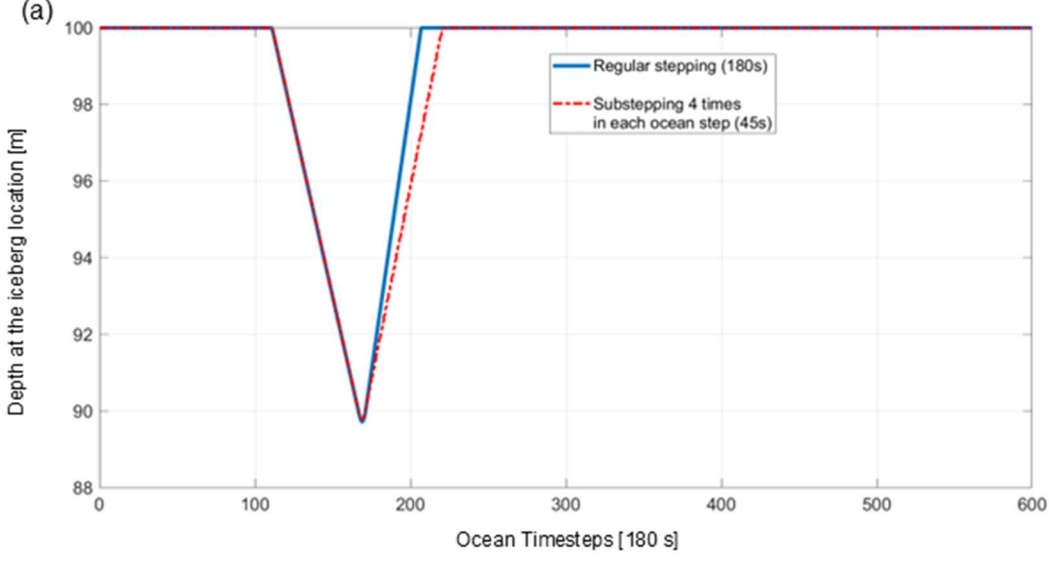

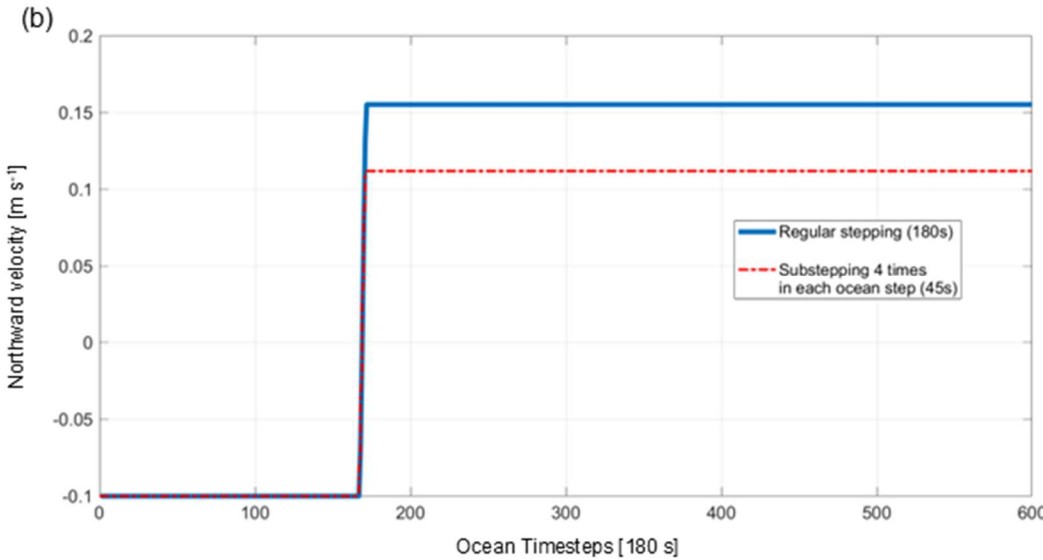


**Figure B3: Asymmetry between the ascent and descent (a); and deceleration and acceleration (b) of an iceberg along a frictionless**
**slope with and without 1075 sub-stepping.**
















## Appendix C. Sea-ice drag on freely-floating and grounded icebergs

We calculate the average magnitude and orientation of the sea-ice drag for the case of freely-floating THICK and MEDIUM icebergs, as well as, kinetically grounded THICK icebergs. In the case of statically grounded icebergs, the solid-body friction and gravity per unit mass are four orders of magnitude larger than the sea-ice contribution, and we do not consider that case here. Figure C1 highlights the different orientation of the sea-ice drag for freely-floating icebergs and grounded ones:. the drag projects onto the forward direction of motion of kinetically grounded icebergs.

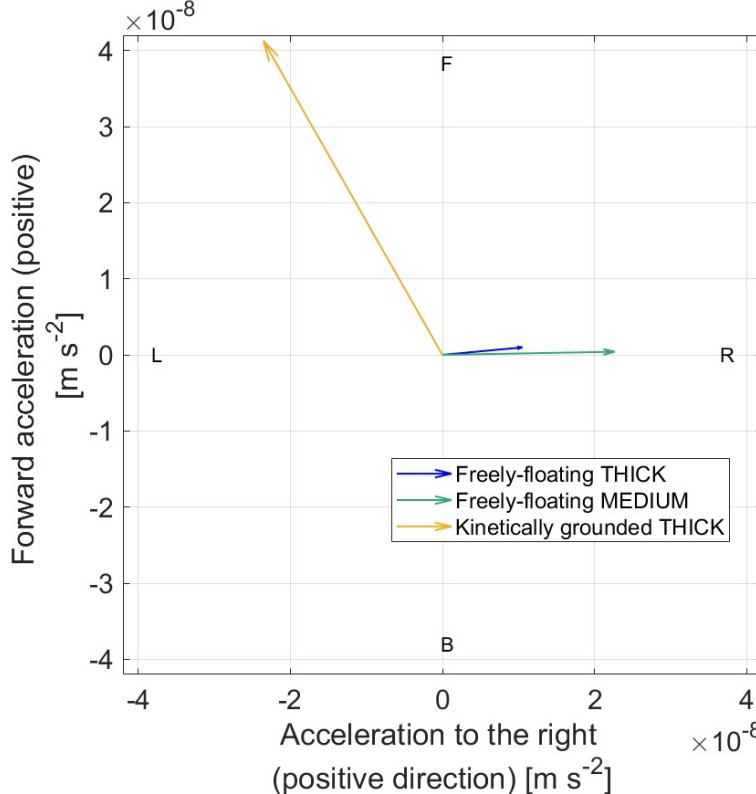

**Figure C1: As Figure 6f and 7f in the main text but for sea-ice drag on freely-floating THICK icebergs (dark blue), freely-floating MEDIUM icebergs (cyan), and kinetically-grounded THICK icebergs (purple).**

## Appendix D.

### D1. Table of Symbols

| Symbol | Unit | Description |
|---|---|---|
| $\vec{a}_{ocean\ drag}$ | $ms^{-2}$ | Acceleration due to ocean drag |
| $\vec{a}_{atmos.drag}$ | $ms^{-2}$ | Acceleration due to atm. drag |
| $\vec{a}_{pressure\ gradient}$ | $ms^{-2}$ | Acceleration due to horizontal oceanic pressure gradient forces |
| $\vec{a}_{Coriolis}$ | $ms^{-2}$ | Acceleration due to Coriolis |
| $\vec{a}_{sea-ice\ drag}$ | $ms^{-2}$ | Acceleration due to sea-ice drag |
| $\vec{a}_{waves}$ | $ms^{-2}$ | Acceleration due to ocean waves |
| $\vec{a}_{sediment}$ | $ms^{-2}$ | Deceleration due to sediment resistance |
| $\vec{a}_{gravity}$ | $ms^{-2}$ | Acceleration due to gravity |
| $\vec{a}_{ground}$ | $ms^{-2}$ | Deceleration due to all forces active only on grounded icebergs |
| $\vec{a}_{solid}$ | $ms^{-2}$ | Deceleration due to solid-body friction |
| $\vec{a}_{static-solid}$ | $ms^{-2}$ | Deceleration due to solid-body friction in a static regime |
| $\vec{a}_{curvature}$ | $ms^{-2}$ | Effective horizontal acceleration due to topographic curvature |
| $\vec{a}_{net}$ | $ms^{-2}$ | Net acceleration |
| $\vec{a}_{net\ drivers}$ | $ms^{-2}$ | Acceleration due to all nondissipative forces |
| $C_{drag}$ | $m^{-1}$ | Drag coefficient parameter |
| $C_{drag-ocn}$ | $m^{-1}$ | Ocean drag coefficient parameter |
| $C_{drag-atm}$ | $m^{-1}$ | Atmospheric drag coefficient over icebergs |
| $C_{drag-atm-to-ocean}$ | $m^{-1}$ | Atmospheric drag coefficient over ocean |
| $D$ | $m$ | Scour depth |
| $D_{keel}$ | $m$ | Keel depth |
| $f$ | $s^{-1}$ | Coriolis parameter |
| g | $ms^{-2}$ | Magnitude of the acceleration due to gravity |
| $\vec{g}$ | $ms^{-2}$ | Acceleration due to gravity |
| $H_{Total}$ | $m$ | Iceberg thickness |
| $H_{freeboard\ anom.}$ | $m$ | Iceberg freeboard anomaly (out of buoyancy-gravity equilibrium) |
| $\vec{k}$ | | Vertical direction |
| $L_{iceberg}$ | $m$ | Characteristic iceberg lengthscale |
| $M_{iceberg}$ | $kg$ | Iceberg mass |
| $\hat{n}$ | | Unit vector normal to the topographic plane |

| | | |
|---|---|---|
| $P$ | $Pa$ | Ocean pressure |
| $\hat{s}$ | | Unit vector in the direction of the topographic slope |
| $\Delta t$ | $s$ | Timestep length |
| $\Delta \vec{u}$ | $ms^{-1}$ | Difference between the iceberg velocity and the velocity of the ambient ocean |
| $\vec{u}_{fluid}$ | $ms^{-1}$ | Velocity of ambient fluid |
| $\vec{u}_{atmos.}$ | $ms^{-1}$ | Atmospheric velocity |
| $\hat{u}$ | $ms^{-1}$ | Unit vector in the direction of the iceberg motion |
| $W$ | $m$ | Scour width |
| $x, y, z$ | | Horizontal dimenstions |
| $\alpha$ | $rad$ | Angle of the trajectory relative to the horizontal plane |
| $\beta$ | $rad$ | Maximum local angle of the topographic slope |
| $\gamma'$ | $Nm^{-3}$ | Submerged unit weight |
| $\delta$ | $m$ | Thickness of the Ekman layer |
| $\mu$ | | Coefficient of Coulomb friction |
| $\mu_{static}$ | | Coefficient of static Coulomb friction |
| $\rho_0$ | $kg\ m^{-3}$ | Reference density of ocean water |
| $\rho_{sat}$ | $kg\ m^{-3}$ | Saturated density of sediment |
| $\rho_{water}$ | $kg\ m^{-3}$ | Water density |
| $\rho_{ice}$ | $kg\ m^{-3}$ | Iceberg density |
| $\tau$ | $Pa\ (Nm^{-2})Pa$ | Shear strength resistance of sediment |
| $\tau_{atm-ocean}$ | $Pa\ (Nm^{-2})$ | Magnitude of the wind stress at the ocean surface |


## Code availability

The updated code for the NEMO iceberg module is publicly available at the following repository
https://doi.org/10.5281/zenodo.15484879

## Acknowledgements

The multibeam-bathymetric data were compiled from the NCEI (National Centers for Environmental Information; USA) and
the NERC PDC (Polar Data Centre; UK) data centres. We thank the masters and crew of all ships involved in data collection;
we thank Frank Nitsche for his help collating the bathymetric data. This research was supported by Ocean Cryosphere
Exchanges in ANtarctica: Impacts on Climate and the Earth system, OCEAN ICE, which is funded by the European Union,
Horizon Europe Funding Programme for research and innovation under grant agreement Nr. 101060452, 10.3030/101060452.
OCEAN ICE Internal contribution number 26. This work was funded by UK Research and Innovation (UKRI) under the UK

government's Horizon Europe funding Guarantee 10048443. A. Olivé Abello, N. Jourdain and P. Mathiot received funding from Agence Nationale de la Recherche - France 2030 as part of the PEPR TRACCS programme under grant numbers ANR-22-EXTR-0008 (IMPRESSION-ESM)  and ANR-22-EXTR-0010 (ISClim).

**Author contribution**

YK developed new model algorithms in coordination with AOA's companion manuscript, conducted numerical simulations, and analysed output. KAH, JAS, and AHF provided observationally-based data and images. YK, PRH, KAH, JAS, NCJ, PM, AOA, AHF, and AJSM contributed content and commented on the manuscript.

**Competing interests**

NCJ is a member of the TC editorial board. The authors declare that they have no other conflict of interest.

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

**Table 1: Bear Ridge iceberg scour metrics**

| Av. depth (m) | Depth range (m) | Av. width (m) | Width range (m) | Av. orientation |
|---|---|---|---|---|
| 7 | 2.5-14 | 189 | 90-357 | 0.94° |

**D1. Table of Symbols**

| Symbol | Unit | Description |
|---|---|---|
| $\vec{a}_{ocean\ drag}$ | $ms^{-2}$ | Acceleration due to ocean drag |
| $\vec{a}_{atmos.drag}$ | $ms^{-2}$ | Acceleration due to atm. drag |
| $\vec{a}_{pressure\ gradient}$ | $ms^{-2}$ | Acceleration due to horizontal oceanic pressure gradient forces |
| $\vec{a}_{Coriolis}$ | $ms^{-2}$ | Acceleration due to Coriolis |
| $\vec{a}_{sea-ice\ drag}$ | $ms^{-2}$ | Acceleration due to sea-ice drag |
| $\vec{a}_{waves}$ | $ms^{-2}$ | Acceleration due to ocean waves |
| $\vec{a}_{sediment}$ | $ms^{-2}$ | Deceleration due to sediment resistance |
| $\vec{a}_{gravity}$ | $ms^{-2}$ | Acceleration due to gravity |
| $\vec{a}_{ground}$ | $ms^{-2}$ | Deceleration due to all forces active only on grounded icebergs |

| | | |
|---|---|---|
| $\vec{a}_{solid}$ | $ms^{-2}$ | Deceleration due to solid-body friction |
| $\vec{a}_{static-solid}$ | $ms^{-2}$ | Deceleration due to solid-body friction in a static regime |
| $\vec{a}_{curvature}$ | $ms^{-2}$ | Effective horizontal acceleration due to topographic curvature |
| $\vec{a}_{net}$ | $ms^{-2}$ | Net acceleration |
| $\vec{a}_{net\ drivers}$ | $ms^{-2}$ | Acceleration due to all nondissipative forces |
| $C_{drag}$ | $m^{-1}$ | Drag coefficient parameter |
| $C_{drag-ocn}$ | $m^{-1}$ | Ocean drag coefficient parameter |
| $C_{drag-atm}$ | $m^{-1}$ | Atmospheric drag coefficient over icebergs |
| $C_{drag-atm-to-ocean}$ | $m^{-1}$ | Atmospheric drag coefficient over ocean |
| $D$ | $m$ | Scour depth |
| $D_{keel}$ | $m$ | Keel depth |
| $f$ | $s^{-1}$ | Coriolis parameter |
| g | $ms^{-2}$ | Magnitude of the acceleration due to gravity |
| $\vec{g}$ | $ms^{-2}$ | Acceleration due to gravity |
| $H_{Total}$ | $m$ | Iceberg thickness |
| $H_{freeboard\ anom.}$ | $m$ | Iceberg freeboard anomaly (out of buoyancy-gravity equilibrium) |
| $\vec{k}$ | | Vertical direction |
| $L_{iceberg}$ | $m$ | Characteristic iceberg lengthscale |
| $M_{iceberg}$ | $kg$ | Iceberg mass |
| $\hat{n}$ | | Unit vector normal to the topographic plane |
| $P$ | $Pa$ | Ocean pressure |
| $\hat{s}$ | | Unit vector in the direction of the topographic slope |
| $\Delta t$ | $s$ | Timestep length |
| $\Delta\vec{u}$ | $ms^{-1}$ | Difference between the iceberg velocity and the velocity of the ambient ocean |
| $\vec{u}_{fluid}$ | $ms^{-1}$ | Velocity of ambient fluid |
| $\vec{u}_{atmos.}$ | $ms^{-1}$ | Atmospheric velocity |
| $\hat{u}$ | $ms^{-1}$ | Unit vector in the direction of the iceberg motion |
| $W$ | $m$ | Scour width |
| $x, y, z$ | | Horizontal dimenstions |
| $\alpha$ | $rad$ | Angle of the trajectory relative to the horizontal plane |

| | | |
|---|---|---|
| $\beta$ | $rad$ | Maximum local angle of the topographic slope |
| $\gamma'$ | $Nm^{-3}$ | Submerged unit weight |
| $\delta$ | $m$ | Thickness of the Ekman layer |
| $\mu$ | | Coefficient of Coulomb friction |
| $\mu_{static}$ | | Coefficient of static Coulomb friction |
| $\rho_0$ | $kg\ m^{-3}$ | Reference density of ocean water |
| $\rho_{sat}$ | $kg\ m^{-3}$ | Saturated density of sediment |
| $\rho_{water}$ | $kg\ m^{-3}$ | Water density |
| $\rho_{ice}$ | $kg\ m^{-3}$ | Iceberg density |
| $\tau$ | $Pa\ (Nm^{-2})Pa$ | Shear strength resistance of sediment |
| $\tau_{atm-ocean}$ | $Pa\ (Nm^{-2})$ | Magnitude of the wind stress at the ocean surface |

**Figure 1:** (a) Amundsen Sea and Bear Ridge topography [m] based on the BedMachine Antarctica dataset (Morlighem et al., 2020) and re-gridded on the model configuration grid. Gray contours are spaced 80 m apart. Thin solid black contours delimit the bounds of boxes a and b in Figure 2, indicating the sites of observed iceberg scours in Section 2.1 with magenta labels matching boxes a and b in Figure 2. Dark red crosses mark the locations of sediment samples analysed in Section 2.2. The thick purple line shows the ice shelf edge, with major ice shelves labelled: Dotson, Crosson, Thwaites, and Pine Island Glacier (PIG). Thick solid (thin dashed) black lines show the boundaries of the boxes where icebergs are released in the SHORT (LONG) simulations. (b) Copernicus Sentinel-1 SAR (synthetic aperture radar) image from 12 February 2022 over Bear Ridge with white shades indicating the presence of icebergs and sea-ice.

**Figure 2:** Examples of iceberg scour types from Bear Ridge (boxes a and b located as shown on the map in Figure 1). The red arrows show likely iceberg transport paths westwards from the eastern Amundsen Sea and across Bear Ridge; blue arrows show iceberg plough ridges (berms); black arrows show the perpendicular direction of iceberg scouring in the crossed forms. Panel c) presents a schematic of a typical scour cross-section found on Bear Ridge.

**Figure 3:** Observed sediment properties for selected cores from the Amundsen Sea: the uppermost sediments consist of soft, water-rich muds (solid black box); coarser-grained stratified to structureless diamictons and 'soft' deformation tills (solid blue box); stiffer tills and/or denser subglacial diamictons (dashed black box). The soft/stiff till boundary follows Reinardy et al. (2009). Values outside the boxes reflect other sediment types such as iceberg diamictons, glaciogenic debris flows etc.

**Figure 4:** Vertical force budget. An idealized schematic of an iceberg trajectory (green arrow) going up a topographic slope. The angle of the trajectory (solid green arrow) relative to its projection on the horizontal x-y plane (dashed grey arrow) is denoted $\alpha$. The local maximum slope of the topography is denoted $\beta$. The vector normal to the topography is indicated with a brown line. Schematic angles not shown to scale with respect to the relevant topographic slopes.

**Figure 5:** Climatology (1979 through 1982) of the Amundsen Sea configuration. (a) potential density [kg m$^{-3}$] referenced to the surface, vertically averaged over the top 310 m with shallower regions masked; (b) sea-surface height [m]; (c) wind stress vectors [N m$^{-2}$]; (d) barotropic streamfunction contours [0.1 Sv intervals]. The bottom depth [m] is superimposed in c and d. The depth range in c chosen to highlight the location of the shallow Bear Ridge and a wider depth range in d to highlight the alignment of streamlines and topography. The ice shelf edge is marked by the thick gray line.

**Figure 6:** Contributions of atmospheric drag (a), the oceanic pressure gradients (b), the Coriolis force (c), the ocean drag (d), the net acceleration (e), and the average of a-d across all test icebergs and all timesteps (f) to the momentum budget of simulated freely floating THICK icebergs in the Amundsen Sea in the SHORT simulation. Directions are denoted as F (forward, along the iceberg's direction of motion), B (backward, against the direction of motion), L (to the left of the direction of motion), R (to the right). The full length of the radial bars indicates the probability density function of acceleration in a given direction. For each panel in a-e, the sum of all directions is 100%, but the bars along some of the directions are relatively small. For each bar in a-e, the range of acceleration magnitudes is indicated by the colours, and the probability of each magnitude class is indicated by the radial distance. Colour sections in a-e indicate the frequency distribution of the accelerations in each magnitude class. Note that the radial axis is different between panels a-e, while the colour scale is common to panels a-e. Panel f has a separate legend and a different colour scheme. The first 100 timesteps of the simulation, when the iceberg motion is initialized from rest, are omitted. Software written by Daniel Pereira was used to create the wind rose (https://www.mathworks.com/matlabcentral/fileexchange/47248-wind-rose, last access: November 2025, MATLAB Central File Exchange)

**Figure 7:** Same as Figure 6 but for freely floating MEDIUM icebergs in the Amundsen Sea in the SHORT simulation.

Figure 8: (a) same as Figure 6f but for the contributions of SSH and density gradients to the acceleration due to pressure gradients acting on THICK freely floating icebergs in the box where they were launched (Fig. 1), away from shallow topography and ice-shelves and (b) as in a but for freely floating MEDIUM icebergs.

Figure 9: Simulated motion of an iceberg along Bear Ridge in an Amundsen Sea configuration of NEMO without sediment resistance or solid-body friction along the bottom. The topography, with contours 4 m apart in elevation, and the superimposed iceberg trajectory are shown in the Mollweide projection. The right panel shows the enlarged iceberg trajectory. Most of the iceberg trajectory corresponds to a state of free flotation, and the iceberg's vertical displacement along the topographic slope is only on the order of centimeters.

Figure 10: Maps showing 14 iceberg trajectories at 20 minute time-intervals (different shades of purple corresponding to individual icebergs) over the four-year LONG simulation with THICK icebergs, where the colorbar indicates the seafloor depth along Bear Ridge highlighting regions shallower than 400 m. All icebergs in this test are launched east of 109.9° W and north of 74° S.

Figure 11: Simulated scours along Bear Ridge from the LONG simulation with THICK icebergs. Panel a shows scour depth [m] and b shows the temporal evolution of the scours [days of the simulation].

Figure 12: As Figure 6 and 7 and with the same colorscale but for the case of kinetic grounding of THICK icebergs. The net term in (e) is dominated by the dissipative forces shown in Figure 13. Occasionally, the orientation of the Coriolis force is not strictly to the left of the iceberg motion for numerical reasons: the force is calculated using the projected directions of future motion at each Runge-Kutta stage of the iceberg dynamics algorithm.

Figure 13: As Figure 12 but for the gravitational acceleration and the deceleration due to dissipative forces in the case of kinetic grounding of THICK icebergs: (a) gravitational acceleration present in a small fraction of the iceberg timestep samples (b), the Coulomb solid-body frictional deceleration (c), deceleration due to kinetic sediment resistance. Occasionally, the orientation of the deceleration due to sediment resistance force is not strictly against the iceberg motion for numerical reasons: the force is calculated using the projected directions of future motion at each Runge-Kutta stage of the iceberg dynamics algorithm.

Figure 14: Components of the sediment resistance force [N] as a function of the scour depth $D$ [m] for icebergs with keel width W=90 m and sediment properties typical of the Amundsen Sea. The individual terms and their sum refer to equations 6 through 11.

Figure 15: As Figure 12 and 13 but in the case of static grounding of THICK icebergs on solid basement with (a) showing the net productive force per unit mass excluding gravity, (b) the gravitational force per unit mass, (c) the sediment resistance per unit mass and (d) static Coulomb friction in the case of static grounding of THICK icebergs. The scale is non-linear and shows the order of magnitude. Note that unlike Figures 12-13, the directions are here the cardinal points as the icebergs do not move. In order to show meaningful geographical directions, here we include only icebergs grounded on Bear Ridge and nowhere else. In the case of grounding along the basement, the icebergs in the simulation always come to rest and enter a static regime within a day. The static Coulomb frictional force per unit mass shown here is the maximum achievable magnitude, but the effective friction does not exceed the sum of the other sources of acceleration, which are dominated by gravity.

Figure 16: Locations of THICK icebergs during the course of the (a) pre-existing SIMPLE GROUNDING; (b) the pre-existing SIMPLE GROUNDING scheme and with masked pressure gradient forces along the iceshelf edge; and (c) LONG dissipative simulation with the new grounding scheme. The topography shallower than 560 m is superimposed with gray contours, 20 m apart. The blue line denotes the ice shelf fronts, where many icebergs come to a stop under the pre-existing SIMPLE GROUNDING scheme. Iceberg locations 4 years after launch are shown for the simulation with new grounding scheme only.

Figure B1: As in Figure 11 but in simulations with a range of kinetic Coulomb coefficient values $\mu$, sub-stepping choices, and representations of the topographic curvature effect.

Figure B2: As in Figure B1 but in simulations with zero Coulomb friction and 8-m sediment thickness. Each row represents an individual iceberg. The left column shows scour depths, and the right column shows their evolution in time during the LONG simulation with THICK icebergs.

Figure B3: Asymmetry between the ascent and descent (a); and deceleration and acceleration (b) of an iceberg along a frictionless slope with and without 1075 sub-stepping.

Figure C1: As Figure 6f and 7f in the main text but for sea-ice drag on freely-floating THICK icebergs (dark blue), freely-floating MEDIUM icebergs (cyan), and kinetically-grounded THICK icebergs (purple).