# Peer review of "Modelled dynamics of floating and grounded icebergs, with application to the Amundsen Sea"

_EGUsphere, 2025_

## Referee Comment (RC2)

[referee-annotated manuscript omitted]

---

## Author Comment (AC1)

We thank the editor and both reviewers for their helpful and constructive feedback, and we address their comments.

Below we first offer our response to the editor, followed by our comments addressed to reviewer 1, finally, to reviewer 2.

Sincerely,

Yavor Kostov, lead author

**Responses to the editor**

Justification (visible to authors and reviewers only):

Dear Dr Kostov and co-authors,

Thank you for your relevant, interesting and novel submission to TC / EGUsphere. I believe that it is suitable for further peer review. Below I have made note of some minor issues that I encourage you to address during the peer review process. They are mostly geared toward accessibility and clarity.

I will now send it for peer review.

**Best regards,**

**Felicity McCormack**

We are grateful to the editor for giving us the opportunity to submit our work to peer review. Our response and revisions will address both the editor's comments and the ones listed by each reviewer.

Below we also include our response to the individual issues raised by the editor.

Best regards,

Yavor Kostov, lead author

Here we respond to specific points that the editor makes.

- I suspect that the developments presented in this manuscript should have substantial impact in terms of modelling icebergs. However, the significance could be described more clearly in the introduction, discussion, and conclusion. For example, the end of the first paragraph in the introduction notes that modelling the behaviour of the largest icebergs is particularly challenging. However, the introduction does not elaborate on what those challenges are, which may make it unclear to readers less (or not at all) familiar with iceberg modelling why the developments presented in this manuscript are important. A brief overview of the limitations of previous modelling approaches -- perhaps highlighting which factors (e.g. thermodynamics, biogeochemistry, interaction with underwater infrastructure, influence on polynya activity) are likely to have the largest impact -- would help clarify the motivation and significance of this work.

We are grateful to the editor for the feedback and the recommendation that we should elaborate on the significance and impact of our new developments in the introduction, discussion, and conclusion. We will edit these sections. As suggested, we will elaborate on the challenges in modelling icebergs and on the importance of our developments.

- Line 15: should this be "iceberg acceleration"

We thank the editor, and we agree that we should specify we are referring to "iceberg acceleration" in Line 15.

- Section 3.4. In places it's difficult to follow what is new in the berg scheme and how it differs from the previous schemes or what is commonly done in other berg schemes. It could be helpful for readers to see the updates represented schematically (e.g. some kind of flowchart), including how the bergs interact with ice shelves, or at least an itemised summary of the new processes / parameterisations implemented

We appreciate the editor's comments about clarifying the novelty of the new grounding scheme in Section 3.4. We will specifically list the new processes that we have implemented, and which were missing from the previous grounding schemes in NEMO.

- There are some sentences that contain large whitespaces which makes me wonder whether some terms haven't rendered in the pdf? (e.g. 204, L454)

We are grateful to the editor for pointing out that there are larger than usual whitespaces in the text (e.g., lines 204 and 454). These are typos that do not correspond to any missing text, and we will easily fix them.

- If you've not already done so, please check all colour maps in the coblis color blindness simulator (<a href="https://www.color-blindness.com/coblis-color-blindness-simulator/">https://www.color-blindness.com/coblis-color-blindness-simulator/</a>) and adapt the colour schemes as necessary

We thank the editor for reminding us to double check the figures in the colour blindness simulator one more time, and we will do so.

- The manuscript is quite long. Please consider whether you can reduce the text length for clarity and conciseness or combine some of the figures (e.g. figures 15 and 16)

We furthermore thank the editor for the suggestion to reduce the text length and to combine figures. We will indeed do our best to make the manuscript more concise, and we will merge Figures 15 and 16, as suggested.

**Reviewer #1**

The authors have developed the modelled dynamics of drifting and grounding/grounded icebergs, with close attention to realism, in particular the evidence from scouring. In the former instance, the pressure gradient force for drifting bergs is more correctly separated into barotropic and baroclinic parts. In the latter case, with a focus on the topographic

obstacle that is Bear Ridge in the Amundsen Sea, more extensive improvements to the NEMO-ICB model configuration are outlined. The attention to dynamical detail is impressive, most notably representation of the force balance for a grounded (and ungrounded) berg. The authors outline in considerable detail the additional forces and accelerations, based on clear fundamental physics, with just a degree of uncertainty in the coefficients of Coulomb friction.

The manuscript is succinctly written throughout. The Introduction (Sect. 1) clearly motivates the model development presented here, with a view to the wider system ice-ocean-climate system. Sect. 2 provides thorough background information on the character of seafloor and sediments, or relevance to grounding. Sect. 3 provides a detailed outline of the existing model equations and developments thereof, model configuration and experimental design. In the Results (Sect. 4), well-crafted figures convey a rich level of information, in particular the wind roses that summarise the strength and relative orientation of accelerations and forces, and the summary force balances (given typically small net accelerations). Sect. 5 provides a brief summary and discussion, pointing towards new modelling possibilities now that the basis is provided for more realistic representation of tabular bergs near Antarctica, specifically the consequences of grounding for sea ice, hydrography and even feedback on the calving process. I close with the following technical comments:

Thank you for your time in carefully reviewing our manuscript and expressing your support! We are grateful for all your comments, and we will implement your suggestions promptly.

**Technical Comments:**

 'Equations' 9, 10, 12-16 are actually terms or relations; either refer to these as such in the main text, or formally make these equations; likewise (27) is a set of proportionalities, not equations

Thank you for this helpful suggestion! We will turn the aforementioned relations into equations and refer to (27) as a proportionality.

2. 6 caption: typo - 'small' rather than 'smalls'

Thank you for pointing this out! We will correct it.

3. Line 797: typo - 'or' not 'of'?

Thank you for pointing this out! We will correct the typo.

**Reviewer #2**

In this manuscript, Y Kostov and co-authors present an updated grounding representation for icebergs in the NEMO ocean model, as well as improvements to how iceberg drift is computed.

The paper is very well written and structured, clearly illustrated, and the subject matter is a natural fit for The Cryosphere. I believe this work presents substantive steps forward in the representation of icebergs in models and I am looking forward to seeing how these changes will improve future iceberg modeling efforts.

In light of this I recommend the paper for publication after revisions, with my comments detailed below. (Please note that some of these comments are musings rather than

requests for edits, arising largely because I am fascinated by this topic. Relatedly, I am keenly aware that I refer to my own papers quite a lot in my comments - which is mostly just a consequence of being most familiar with those and not a request for citations).

Thank you for your thorough and constructive comments and suggestions! We appreciate your feedback and the references to relevant papers that we will cite appropriately.

**General Comments:**

- 1) My most substantial comment is that I do wonder whether the paper may benefit from being split into 2 separate articles: one on grounding and one on drift dynamics. My reasons for suggesting this are two-fold:
- i) The paper is quite long and it is at points hard to keep track of all the different pieces (see also a similar comment by the editor).

We thank you and the editor for this comment. We will shorten the paper and we are grateful that you have highlighted particular portions of the text that could be written more concisely.

ii) The paper consists of two fairly independent components: the grounding parameterization and the free drift analysis and improvements. While the grounding work is more developed in the manuscript as it stands I would argue that there is plenty of material to expand the drift analysis into its own paper (without too much extra work). Such a split could streamline the presentation in a number of ways, for example, you wouldn't have to bring in the MEDIUM icebergs at all for the grounding work. I do think a split would also help the impact of the work - other modeling groups may be more likely to pick up on the improvements in grounding when this is presented in a more focused way.

We are more inclined not to split the draft paper into two manuscripts. We think that in order to describe grounding behaviour, we have to understand how freely floating icebergs approach topographic obstacles such as Bear Ridge. In addition, the contrast between the force balance of freely floating and grounded icebergs is itself very revealing. As an alternative to splitting the paper, we will also consider building stronger bridges in the text between the passages focusing on freely floating icebergs and the ones that directly concern grounding. However, if keeping the manuscript intact will delay publication, we will adhere to the editor's advice.

Having made this case, I happily leave it to the authors and the editor what to do about it.

2) There are a few passages where I thought text could be shortened somewhat. I have highlighted those in the attached pdf.

We will go through all passages that you have highlighted and rewrite them more concisely.

3) It would be helpful to early on provide a short discussion of the types of icebergs that get stuck on Bear Ridge with typical sizes and approximate numbers. While reading the paper, I somehow assumed there would be only a handful of large tabular icebergs at a given time, until I got to Appendix A and realized you are mostly talking about ~hundreds of fairly small icebergs.

Thank you for this comment! Yes, indeed, we should point out that these are hundreds of fairly small icebergs, which may still fall into the model's largest size classes.

Relatedly, I would recommend picking one of the images from the timelapse movie (ideally one with very clear sea ice differences on the two sides of the "wall of icebergs", e.g., timestamp 2:15 of the movie), annotate this, and combine it with figure 1, to provide the reader early on with a sense of the general setup. These images are rather striking.

We will combine still frame ~2:15 of the movie with Figure 1. Thank you for this suggestion!

4) The manuscript is largely focused on grounding, however, I'd argue that the subsequent ungrounding is also important. [As a side note: Reading the paper I was wondering whether ungrounding is primarily the result of melting (and potentially capsizing), or rather changes in ocean current/wind direction? This is not a focus of this work, but if you have any insight I'd be interested to hear it.]

Yes, indeed, the complete ungrounding in the model seems to be the result of melting. It is a very relevant question whether basal melting or lateral melting and capsizing dominate. We have not explored the capsizing of grounded icebergs, but we will now analyse such cases in our simulation.

In contrast, winds seem to be the main driver of motion for kinetically grounded icebergs that remain embedded in the sediment.

While I agree with the authors' choice to focus on the novel representation of the grounding process, I do think it would be helpful to also discuss ungrounding and the role of melting. Two things came to mind:

We will expand the text to discuss the mechanism behind ungrounding and the transition to free flotation in a new subsection between 4.2 and 4.3 titled Ungrounding of icebergs.

- First, as far as I know the melt model in NEMO-ICB contains a dependence of basal melt on the relative velocity between the iceberg and the ocean current at the height of the iceberg base (Merino et al, 2016, eq 2). I wonder how this plays out for grounded icebergs in the latest version?

Thank you for raising this question. We will add a clarification that in the new grounding formulation, the relative basal velocity is computed at the height of the iceberg grounding level. This formulation may not be perfect, but it allows even the statically grounded icebergs which have no velocity of their own to melt at the base.

Lines 871/872 make it sounds like this is not the case in the current model formulation?

We will clarify that lines 871/872 refer to the melting of freely floating icebergs.

The dependence on the size of the iceberg is evident as well in the Merino et al. formulation. I'm likely missing something, but maybe this paragraph could be reworded and/or clarified?

We will clarify that the melting in the Merino formulation depends on the relative velocity and directly on  $L^{-0.2}$ . However, we suggest that the relative velocity between the iceberg and the ocean itself depends on the inverse length scale. Hence, we expect the inverse length scale to have a much stronger net impact on the melting rate.

- Second, freely floating icebergs typically erode much faster on the side walls due to wave erosion (~1 m/d) than the base (~0.1 m/d) - see, e.g., Wagner & Eisenman (GRL, 2017, https://doi.org/10.1002/2016GL071645). In that case you might expect that icebergs shrink laterally until the aspect ratio becomes unstable and they become ungrounded by capsizing (as the authors mention). This may be particularly relevant for the smaller icebergs found all over Bear Ridge. However, since for grounded icebergs the relative basal velocity is higher, maybe the thinning is substantially faster than for freely floating ones, which might entail that capsizing isn't that important after all. Would it be easy to check how often ungrounding in the model coincides with capsizing? I appreciate that a detailed analysis of these processes is beyond the scope of this study, but I do think it would be helpful to comment on how melt is represented in the model, and to at least mention some of the considerations above.

Thank you for this comment. We will add a discussion of the relative magnitudes of the lateral and basal melting for grounded icebergs in our new subsection 4.3 Ungrounding of icebergs. Our basal melting formulation for grounded icebergs indeed allows for sustained vertical thinning. We also suspect that the wave erosion may be exaggerated in the model, and we will discuss this further in our new subsection 4.3 Ungrounding of icebergs where we will highlight melting as the dominant mechanism. We will check whether capsizing of grounded icebergs takes place and facilitates the ungrounding or whether basal melting is sufficient to detach icebergs from the bottom.

5) It was my understanding that the original NEMO-ICB used the erroneous capsizing criterion of Bigg et al (2017). We published a correction to this in Wagner et al (Ocean Modeling, 2017, https://doi.org/10.1016/j.ocemod.2017.07.003) and I discussed this briefly with Bob Marsh back then but never followed up. I just want to make sure the capsizing errors have been fixed, if there ever were any.

Compared to Wagner et al. (OM, 2017) and Bigg et al. (2017), the present capsizing criterion uses a different power law, namely:

SQRT( 0.92\*(KeelDepth\*\*2) + 58.32\*KeelDepth ) )

This makes a direct comparison with Wagner et al. (OM, 2017) formulation more difficult.

However, it stands out that in the present NEMO-ICB version, the criterion is applied as a comparison between the iceberg horizontal length and keel depth. Wagner et al. (OM, 2017) suggests that the appropriate comparison should be between the horizontal width and the full thickness rather than the horizontal length and the keel depth. The full thickness swaps with the horizontal width when the iceberg rotates. Analysing and improving the capsizing is beyond the scope of our current manuscript. However, our companion manuscript Abello et al. (2025) introduces an updated rolling criterion in NEMO that is based on the ratio between horizontal width and the full thickness while taking into account the exact issues highlighted in Wagner et al. (OM, 2017).

We will look for cases of capsizing grounded icebergs in our simulation, and if these take place, we will analyse them. We will also point the reader to our companion manuscript Abello et al. (2025), which updates the rolling criterion in NEMO.

**Specific comments:**

A number of mostly minor and technical comments are provided as annotations to the attached pdf.

Thank you for your detailed feedback. We have responded to these comments line by line below.

Dear authors - I am often wrong, and if you think that any of my comments are misguided please reach out to me and I'll be eager to amend my review.

Till Wagner

Line 10: not sure I'd call it ubiquitious - maybe "commonly observed"?

Line 10: Thank you, we will replace it with "commonly observed".

**Lines 12-13, remove text: Strikethrough text**

Lines 12-13, remove text: Thank you, we will shorten the text as suggested.

Line 27: just note that this is given as "2025" elsewhere.

Line 27: Thank you, we will change "in prep." to 2025.

**Lines 30-39: This paragraph could be shortened to 1 or 2 sentences?**

Lines 30-39: Thank you for this suggestion, but we think that it is important to refer the reader to broadly relevant previous literature on icebergs. This entire paragraph is composed of citations of previous publications that we prefer not to remove from our reference list.

Lines 31-32: see also: Duprat, L., Bigg, G. & Wilton, D. Enhanced Southern Ocean marine productivity due to fertilization by giant icebergs. Nature Geosci 9, 219–221 (2016). https://doi.org/10.1038/ngeo2633

Lines 31-32: Thank you, we will cite Duprat et al. (2016) here.

Line 43: there is also a body of literature that looks at iceberg scouring to shed light on paleo processes. 2 examples: Hill, J., Condron, A. Subtropical iceberg scours and meltwater routing in the deglacial western North Atlantic. Nature Geosci 7, 806–810 (2014). https://doi.org/10.1038/ngeo2267 and Starr, A., Hall, I.R., Barker, S. et al. Antarctic icebergs reorganize ocean circulation during Pleistocene glacials. Nature 589, 236–241 (2021). https://doi.org/10.1038/s41586-020-03094-7

Line 43: Thank you, we will also cite the suggested literature on iceberg scours in a paleoclimate context.

Line 44: just a note that, relatedly, Stern et al (2015)

https://doi.org/10.1002/2015JC010805 found that grounded icebergs can act much like islands, in the sense that they can cause upwelling on the downwind side and increased stratification on the upwind side. Which made me wonder - how much of the differences in sea ice cover between east and west of Bear Ridge is due to icebergs mechanically blocking the sea ice advection and how much is due to the icebergs causing different ocean conditions on the two sides? Also, I found it interesting that these differences in turn impact the melt process of the iceberg itself (disclaimer - I'm an n-th author of that paper).

Line 44: Thank you for pointing us to this reference. We will cite it here in the context of modified ocean conditions sustained by grounded icebergs. We avoid speculating whether such a downwelling effect on the eastern side of Bear Ridge contributes significantly to the sea-ice dipole. Bett et al. (2020) found that representing the grounded bergs as a thin ice shelf had the same effect on sea-ice as representing them as land. This may suggest that the impact of the altered ocean conditions on sea-ice is small compared to the direct blocking of sea-ice advection.

Line 62: maybe better in quotation marks? "wall of icebergs"

Line 62: Yes, we will add quotation marks, as you suggest.

Lines 65-67: I would argue that there were important efforts quite a bit earlier - e.g., Mountain (1980) https://doi.org/10.1016/0165-232X(80)90055-5 and Smith and Banke (1983) https://doi.org/10.1016/0165-232X(83)90045-9

Maybe it would be more accurate to say that most currently used iceberg models can be traced back to Bigg et al (1997) and Gladstone et al (2001).

Lines 65-67: Thank you for pointing us to these references. We will cite them here and rephrase the statement about Bigg et al. (1997) and Gladstone et al. (2001).

Interestingly, Mountain (1980) actually assumes that Ekman currents play an important role in driving iceberg motion. So we will also cite this in our Section 4.1 when discussing the ageostrophic contribution to the ocean dynamics around icebergs.

Lines 99-106: paragraph could maybe be shortened

Lines 99-106: Yes, we will shorten this paragraph, as you suggest.

Line 112: Shoal

Line 112: Thank you. We will replace "shallow" with "shoal," as you suggest.

Line 132: in my experience "capsizing" is used more typically in the literature

Line 132: Thank you. We will replace "toppled" with "capsized," as you suggest.

Line 132: note that the standard capsizing criterion that was used by Bigg et al (1997) and most subsequent iceberg modeling studies dated back to Weeks and Mellor (1978) https://doi.org/10.1016/B978-0-08-022916-4.50015-7 and featured some errors that led to icebergs continually capsizing among other things (see also my general comment 5).

However, it stands out that in the present NEMO-ICB version, the criterion is applied as a comparison between the iceberg horizontal length and keel depth. Wagner et al. (OM, 2017) suggests that the appropriate comparison should be between the width and the full thickness rather than the length and the keel depth. The full thickness swaps with the horizontal width when the iceberg rotates. Analysing and improving the capsizing is beyond the scope of our current manuscript. However, our companion manuscript Abello et al. (2025) introduce an updated rolling criterion in NEMO that is based on the ratio between horizontal width and the full thickness while taking into account the exact issues highlighted in Wagner et al. (OM, 2017). We have already cited Abello et al. (2025), but we will also refer to it when discussing the capsizing criterion and its update in NEMO.

Figure 2: Instead of using a schematic for panel (c), could you take an actual representative cross-section example - and you could mark that cross-section on the map of panel a or b?

Figure 2: Thank you for this suggestion, which we have carefully considered. However, individual sections do not necessarily provide a clearer picture compared to this schematic, which is very representative of typical scour shapes.

Line 155: could you mention in this paragraph briefly why you're interested in WBD and shear strength - I was wondering about it until I got quite a bit further down.

Line 155: Thank you, yes we will mention that WBD and shear strength determine the sediment resistance forces acting on grounded icebergs.

**Line 167: Strikethrough text**

Line 167: Thank you for suggesting this correction. We will remove the "s."

Figure 3: It's not clear to me exactly how the partition of the 3 layers plays out in this plot - maybe you could color the markers accordingly or draw approximate boxes around them.

I was also wondering whether different colors (or using different markers) for different cores may be insightful?

Figure 3: Yes, we will make sure that the three layers are differentiated in this figure by drawing boxes around the data markers while using distinct color and line types for the box contours.

**Lines 200-201: are these two sentences needed?**

Lines 200-201: In order to make the text more concise, we will remove these sentences.

**Line 204: Strikethrough text**

Line 204: Thank you for pointing out this typo. We will correct it.

**Line 204: And**

Line 204: Thank you for pointing out this typo. We will correct it.

Line 208: Martin and Adcroft (2010) distinguished between form drag and skin drag. How does NEMO-ICB deal with this?

[As a sidenote: we looked in some detail at the relative importance of form vs skin drag and found that this varies substantially depending on whether you have a high length-to-height ratio or one that is O(1) - take a look if you're interested: https://doi.org/10.1175/JPO-D-20-0275.1

See also a further comment on C\_drag below.

Line 208: Here we will clarify that NEMO-ICB assumes form drag, and we will cite Martin and Adcroft (2010), as well as Wagner et al. (2022).

Line 239: Maybe a short paragraph on the melt representation in NEMO-ICB following here?

Line 239: You suggest adding a short paragraph on the melt representation here. We think this might be more appropriate after Line 365, and we will add it there.

Lines 241-243: Maybe it's worth mentioning that you envisage the icebergs to plough a triangular (v-shaped) trench into the sediment, however, the motion of the iceberg is computed using a perfectly rectangular iceberg. (I've always assumed that these icebergs are approximately flat at the bottom, which makes me think: how do they leave v-shaped scours? Am I missing something obvious?)

Lines 241-243: Thank you! Yes, we will mention that guided by observations, we assume that iceberg keels plough v-shaped trenches into the sediment. Even tabular icebergs are not flat at the bottom but rough.

**Line 256 and Section 2.2: is this the same as WBD? maybe clarify?**

Line 256 and Section 2.2: Yes, thank you for pointing out the inconsistent use of terminology which we will fix. We assume fully saturated sediment, so in our case WBD is the same as saturated density.

Line 267, Eq. 11: could move this term in front of the curly bracket, since it's the same for the two cases?

Line 267, Eq. 11: Yes, we will move the common factor outside the curly brackets.

Lines 274-277: maybe this should be moved to near line 170?

Lines 274-277: Yes, as you suggest, we will move these sentences right after Line 170.

Line 299: you provide a reference and more discussion for this later- I would move it up here and shorten

Line 299: Thank you! We will move the sentences and the Veldhuijsen et al. reference from Lines 449-451 to this part of the text.

Line 321: Strikethrough text

Line 321: We will fix the mistyped sentence, as you suggest.

Line 321: of motion

Line 321: We will fix the mistyped sentence, as you suggest.

Line 321: Strikethrough text

Line 321: We will fix the mistyped sentence, as you suggest.

Line 321: capped?

Line 321: Yes, we will rephrase "limited" to "capped."

Lines 329-330: maybe discuss briefly here what range you tested and that the results are fairly insensitive to this exact value. Looking at Appendix B it seems to me that the effect of mu is saturated once you exceed 0.002, and the main differences arise somewhere in the range  $\,$ mu = [0,0.002]. Did you look at smaller values as well?

Lines 329-330: Thank you for pointing out that we should discuss the Appendix B results here in more detail. We will state that we have done tests with values higher and lower than 0.002, including a case with no Coulomb friction that still includes sediment resistance and gravity. We will refer the reader to Appendix B.

**Line 339 and equation 18: d**

Line 339 and equation 18: Thank you! We will fix the typo.

Line 341: force

Line 341: Thank you! We will add the missing word "force," as you point out.

Lines 344-361: I wonder whether this paragraph could be cut/shortened/moved to the appendix?

Lines 344-361: Thank you for this suggestion, we will mention the curvarture term here, and move most of this paragraph to Appendix B.

Lines 407-419: This paragraph could maybe be summarized in a sentence or two?

Lines 407-419: Thank you for this suggestion. We will shorten this paragraph.

Lines 449-451: move this density discussion up (see earlier comment).

Lines 449-451: Yes, we will move this discussion to Line 299.

Line 471: maybe mention here somewhere how melt and ungrounding happens (+capsizing?) - see my general comment 4.

Line 471: Thank you for this suggestion. Here we will point out that complete un-grounding and the resumption of free flotation is due to the melting process and compare the relative roles of basal melting versus lateral melting and capsizing.

Section 4.1: The following analysis is a really interesting step forward! I've always felt a little uneasy about our approximation ignoring the ageostrophic currents.

I will say that I struggled a bit to follow the argument of the next few pages in detail, I guess at least in part because I'm used to thinking of it in terms of the force balance equations, rather than just the accelerations. And since I think of the drags, for example, as "surface forces" acting on cross-sectional areas and Coriolis and gravity as "body forces" acting on the mass of the iceberg, I found myself doing some cognitive aerobics reconciling the two frameworks. I am not suggesting you recast the whole argument in terms of forces, but maybe you can go over the text once more with a critical eye to make it as clear as possible?

Section 4.1: Thank you for your feedback! We will rephrase this part of the text to make it clearer. For example, when discussing the potential sources of acceleration on grounded icebergs, we will refer to them as "forces per unit mass" in the case of static grounding.

**Line 542, Eq. 22: this reminds me of the discussion in Bigg et al (1996) - see their eq (4)**

Line 542, Eq. 22: Thank you for pointing this out. We have cited Bigg et al. (1996) elsewhere, and we will also cite that reference here. We will point the reader to eq (4) in Bigg et al. (1996).

Lines 590-599: this may speak to my confusio (see comment above) - how exactly is C\_drag obtained here?

Lines 590-599: Thank you for pointing out the need for greater clarity! We will specify that this is form drag, which is calculated as a function of the fluid density and iceberg geometry. We will specify that the values cited here are output directly by the model during the respective simulations.

Lines 590-599: following up on an earlier comment, it appears that C\_drag represents the form drag only? (For skin drag in the limit of thin icebergs the atmospheric and oceanic drag coefficients are comparable - see Wagner et al (2022)). Might be helpful to clarify what C\_drag represents.

Lines 590-599: Thank you once again for your feedback! We will clarify that this is form drag, which is calculated as a function of the fluid density and iceberg geometry. We will cite Wagner et al (2022). We will specify that the values cited here are output directly by the model during the respective simulations.

Lines 710-711: this is largely a repeat of l 696/697. Also, maybe make it clear when you first introduce MEDIUM icebergs that these do not feature in the grounding discussion?

Lines 710-711: We agree that we should remove these lines because re-stating this information here is redundant to Lines 696-697. Moreover, as you suggest, we will point out earlier, in Lines 451-452, that MEDIUM icebergs do not feature in the grounding analysis.

**Line 754: Strikethrough text**

Line 754: Thank you! We will move the quotation marks outside the word "kinetic".

**Line 754: "**

Line 754: Thank you! We will move the quotation marks outside the word "kinetic".

**Line 766: Strikethrough text**

Line 766: We will move the word "only," as you suggest, and we will delete the phrase "the course of."

**Line 766: Strikethrough text**

Line 766: We will move the word "only," as you suggest, and we will delete the phrase "the course of."

**Line 766: only**

Line 766: We will move the word "only," as you suggest, and we will delete the phrase "the course of."

**Line 785, Figure 13 caption: cut?**

Line 785, Figure 13 caption: We will delete the mistyped word "and," as you suggest.

**Lines 797-798: something went wrong here - revise?**

Lines 797-798: Yes, we apologize for accidentally pasting the unnecessary words at the end of the sentence. We will delete them.

**Lines 797-798: Strikethrough text**

Lines 797-798: Yes, we apologize for accidentally pasting the unnecessary words at the end of the sentence. We will delete them.

**Lines 812-813: Maybe this can be combined with the first discussion of mu on l 329? See also my comment there.**

Lines 812-813: Thank you, we will indeed move that to the earlier discussion of mu on Line 329.

**Figure 15: I'm missing something here, sorry - how do you have acceleration in the case of "statically grounded"? Could this be rephrased?**

Figure 15: We accept your suggestion. In the case of statically grounded icebergs, we will rephrase "acceleration" to "force per unit mass."

**Lines 854-896 on page 32: I feel like this page could be shortened?**

Lines 854-896 on page 32: We will do our best to make this text more concise.

Lines 867-869: note that this was studied (and indeed found to be the case) in England et al (2022) https://doi.org/10.1126/sciadv.abd1273 (disclaimer: I'm a co-author) and breakup was implemented in the GFDL model in Huth et al (2023) https://doi.org/10.1029/2021MS002869

In my experience, breakup is a first-order process for large icebergs. I'd be excited to talk more if you're interested in adding something like this to NEMO?

Lines 867-869: Thank you for pointing us to these references, which we will cite here. It is noteworthy that England et al (2022) demonstrate the potential for iceberg fragments to move at a different angle relative to parent icebergs. In England et al. (2022) the Wagner et al. (2017) analytical model is directly imposed on icebergs. At the same time, it would be interesting if a future study explores whether this motion of iceberg fragments appears as an emergent phenomenon when all model forces are applied to the iceberg momentum budget. There is also indeed potential room to improve the iceberg representation in NEMO by introducing fragmentation.

**Lines 872-873: I think it does, and I think the NEMO model accounts for that (?) see my comment earlier.**

Lines 872-873: Yes, we will add a clarification that the melting in NEMO is indeed a function of the relative velocity and the geometry. Our results imply that the iceberg lengthscale impacts the rate of melting not only by changing the geometry of the icebergs but via an impact on the relative velocity. This impact on melting via the relative velocity is an interesting new avenue for exploration in a future study, especially if it naturally emerges from the full set of dynamical and thermodynamical forcing factors applied to simulated icebergs.

**Line 1134: Strikethrough text**

Line 1134: Thank you. We will delete the typo.

---

## Author Response (AR1)

We thank the editor and the reviewers for their helpful input, and we respond to their feedback below.

Sincerely,

Yavor Kostov, lead author

**Responses to the editor**

**Justification (visible to authors and reviewers only):**
**Dear Dr Kostov and co-authors,**

**Thank you for your relevant, interesting and novel submission to TC / EGUsphere. I believe that it is suitable for further peer review. Below I have made note of some minor issues that I encourage you to address during the peer review process. They are mostly geared toward accessibility and clarity.**

**I will now send it for peer review.**

**Best regards,**
**Felicity McCormack**

We thank the editor for allowing our manuscript to advance to the next stage of peer review. Our revisions have addressed both the editor's and the reviewers' feedback.

We enclose responses to each comment below, and we point to specific line-by-line changes in the manuscript.

Best regards,

Yavor Kostov, lead author

**- I suspect that the developments presented in this manuscript should have substantial impact in terms of modelling icebergs. However, the significance could be described more clearly in the introduction, discussion, and conclusion. For example, the end of the first paragraph in the introduction notes that modelling the behaviour of the largest icebergs is particularly challenging. However, the introduction does not elaborate on what those challenges are, which may make it unclear to readers less (or not at all) familiar with iceberg modelling why the developments presented in this manuscript are important. A brief overview of the limitations of previous modelling approaches -- perhaps highlighting which factors (e.g. thermodynamics, biogeochemistry, interaction with underwater infrastructure, influence on polynya activity) are likely to have the largest impact -- would help clarify the motivation and significance of this work.**

We are grateful to the editor for this recommendation. We have now explained in greater detail the modelling challenges in lines 28-32 of the introduction. We have cited an additional source, Duprat et al. (2016) regarding the role of icebergs as a source of iron (line 41). We have also pointed out the importance of properly representing iceberg grounding for the simulation and projections of sea-level rise (lines 71-73). In lines 911-914 of the Summary and Conclusion, we

once again elaborate on the implications of iceberg grounding for the stability of the ice shelves and for the Antarctic contribution to global sea level rise in future climate projections.

**- Line 15: should this be "iceberg acceleration"**

We thank the editor, and we agree that we should specify we are referring to "iceberg acceleration" in Line 14.

**- Section 3.4. In places it's difficult to follow what is new in the berg scheme and how it differs from the previous schemes or what is commonly done in other berg schemes. It could be helpful for readers to see the updates represented schematically (e.g. some kind of flowchart), including how the bergs interact with ice shelves, or at least an itemised summary of the new processes / parameterisations implemented**

We appreciate the editor's comments about clarifying the novelty of the new grounding scheme in Section 3.4. In lines 440-444, we have now specifically listed the new processes that we have implemented, and which were missing from the previous grounding schemes in NEMO.

**- There are some sentences that contain large whitespaces which makes me wonder whether some terms haven't rendered in the pdf? (e.g. 204, L454)**

We thank the editor for pointing out that there are larger than usual whitespaces in the text (e.g., lines 219 and 471). We have fixed them.

**- If you've not already done so, please check all colour maps in the coblis color blindness simulator (https://www.color-blindness.com/coblis-color-blindness-simulator/) and adapt the colour schemes as necessary**

We thank the editor for reminding us to double check the figures in the colour blindness simulator one more time. Following up on that, we have made Figures 1, 8, 16, and C1 more accessible.

**- The manuscript is quite long. Please consider whether you can reduce the text length for clarity and conciseness or combine some of the figures (e.g. figures 15 and 16)**

We have reduced the text length in the new lines 12,110, 112, 114, 168-175, 198-194, 216, 311-333, 358, 428-430, 872-881, 895-897 and 904-905. We have also merged Figures 15 and 16 and their respective captions, as suggested. In the process, we noticed an issue with the new Figure 15a. In that figure, we show the net productive forces (except gravity) acting on icebergs that are statically grounded along the solid basement. Since the icebergs are static, we show cardinal geographical directions. However, previously this figure averaged over icebergs that ground along Bear Ridge and others that ground in sediment and along the bottom elsewhere. In order to keep a consistent focus on grounding on top of Bear Ridge, in the updated figure, we focus only on the subset of icebergs there.

We have shortened many parts of the previous text. However, we should point out that in response to Reviewer #2, we have added a new subsection 3.3 on iceberg melting and capsizing. We have also discussed capsizing once again in lines 846-858, while acting on feedback from Reviewer #2.

**Reviewer #1**

The authors have developed the modelled dynamics of drifting and grounding/grounded icebergs, with close attention to realism, in particular the evidence from scouring. In the former instance, the pressure gradient force for drifting bergs is more correctly separated into barotropic and baroclinic parts. In the latter case, with a focus on the topographic obstacle that is Bear Ridge in the Amundsen Sea, more extensive improvements to the NEMO-ICB model configuration are outlined. The attention to dynamical detail is impressive, most notably representation of the force balance for a grounded (and ungrounded) berg. The authors outline in considerable detail the additional forces and accelerations, based on clear fundamental physics, with just a degree of uncertainty in the coefficients of Coulomb friction.

The manuscript is succinctly written throughout. The Introduction (Sect. 1) clearly motivates the model development presented here, with a view to the wider system ice-ocean-climate system. Sect. 2 provides thorough background information on the character of seafloor and sediments, or relevance to grounding. Sect. 3 provides a detailed outline of the existing model equations and developments thereof, model configuration and experimental design. In the Results (Sect. 4), well-crafted figures convey a rich level of information, in particular the wind roses that summarise the strength and relative orientation of accelerations and forces, and the summary force balances (given typically small net accelerations). Sect. 5 provides a brief summary and discussion, pointing towards new modelling possibilities now that the basis is provided for more realistic representation of tabular bergs near Antarctica, specifically the consequences of grounding for sea ice, hydrography and even feedback on the calving process. I close with the following technical comments:

Thank you carefully reviewing our manuscript and encouraging us! We have implemented your suggestions.

Technical Comments:

1. 'Equations' 9, 10, 12-16 are actually terms or relations; either refer to these as such in the main text, or formally make these equations; likewise (27) is a set of proportionalities, not equations

   Thank you for this helpful suggestion! We have turned the aforementioned relations (now equations 9,10, 12, and 14) into equations and refer to (24) as a set of proportionalities.

2. 6 caption: typo - 'small' rather than 'smalls'

   Thank you for pointing this out! We have corrected the Figure 6 caption.

3. Line 797: typo – 'or' not 'of'?

   Thank you for pointing this out! We have corrected the typo (now lines 814-815).

In this manuscript, Y Kostov and co-authors present an updated grounding representation for icebergs in the NEMO ocean model, as well as improvements to how iceberg drift is computed.

The paper is very well written and structured, clearly illustrated, and the subject matter is a natural fit for The Cryosphere. I believe this work presents substantive steps forward in the representation of icebergs in models and I am looking forward to seeing how these changes will improve future iceberg modeling efforts.

In light of this I recommend the paper for publication after revisions, with my comments detailed below. (Please note that some of these comments are musings rather than requests for edits, arising largely because I am fascinated by this topic. Relatedly, I am keenly aware that I refer to my own papers quite a lot in my comments - which is mostly just a consequence of being most familiar with those and not a request for citations).

Thank you for your detailed and constructive comments and recommendations! We are grateful for your feedback that we have acted upon. We have also cited the additional references to relevant papers that you have provided.

**General Comments:**

1) **My most substantial comment is that I do wonder whether the paper may benefit from being split into 2 separate articles: one on grounding and one on drift dynamics. My reasons for suggesting this are two-fold:**

   **i) The paper is quite long and it is at points hard to keep track of all the different pieces (see also a similar comment by the editor).**

   We thank you and the editor for this comment. We have shortened the paper and we are grateful that you have highlighted particular portions of the text that could be written more concisely.

   We have reduced the text length in the new lines 12,110, 112, 114, 168-175, 198-194, 216, 311-333, 358, 428-430, 872-881, 895-897 and 904-905. We have also merged Figures 15 and 16 and their respective captions, as suggested. In the process, we noticed an issue with the new Figure 15a. In that figure, we show the net productive forces (except gravity) acting on icebergs that are statically grounded along the solid basement. Since the icebergs are static, we show cardinal geographical directions. However, previously this figure averaged over icebergs that ground along Bear Ridge and others that ground in sediment and along the bottom elsewhere. In order to keep a consistent focus on grounding on top of Bear Ridge, in the updated figure, we focus only on the subset of icebergs there.

   We have shortened many parts of the previous text. However, we should point out that in response to your comments, we have added a new subsection 3.3 on iceberg melting and capsizing. We have also discussed capsizing once again in lines 846-858, while acting on your feedback.

   **ii) The paper consists of two fairly independent components: the grounding parameterization and the free drift analysis and improvements. While the grounding work**

is more developed in the manuscript as it stands I would argue that there is plenty of material to expand the drift analysis into its own paper (without too much extra work). Such a split could streamline the presentation in a number of ways, for example, you wouldn't have to bring in the MEDIUM icebergs at all for the grounding work. I do think a split would also help the impact of the work - other modeling groups may be more likely to pick up on the improvements in grounding when this is presented in a more focused way.

We decided not to split the draft paper into two manuscripts. Here is once again our reasoning. In order to describe grounding behaviour, we have to understand the dynamics of freely floating icebergs as they approach topographic obstacles such as Bear Ridge. The contrast between the force balance of freely floating and grounded icebergs is itself very illuminating. Instead of splitting the paper, we have strengthened a few of the bridges in the text between passages focusing on freely floating icebergs and the ones that directly concern grounding (Lines 540-542 and 692).

Having made this case, I happily leave it to the authors and the editor what to do about it.

2) There are a few passages where I thought text could be shortened somewhat. I have highlighted those in the attached pdf.

We have gone through all passages that you have highlighted and rewritten them more concisely. For example, we have reduced the text length in the new lines 12,110, 112, 114, 168-175, 198-194, 216, 311-333, 358, 428-430, 872-881, 895-897 and 904-905.

3) It would be helpful to early on provide a short discussion of the types of icebergs that get stuck on Bear Ridge with typical sizes and approximate numbers. While reading the paper, I somehow assumed there would be only a handful of large tabular icebergs at a given time, until I got to Appendix A and realized you are mostly talking about ~hundreds of fairly small icebergs.

Thank you for this comment! Yes, indeed, we have now pointed out that these are hundreds of fairly small icebergs in the Amundsen Sea Embayment (Lines 50-52): "in shelf seas such as the Amundsen Sea Embayment, where there are hundreds of icebergs at any time and almost 90% of them are smaller than 2 km$^2$ (Mazur et al., 2019)."

Relatedly, I would recommend picking one of the images from the timelapse movie (ideally one with very clear sea ice differences on the two sides of the "wall of icebergs", e.g., timestamp 2:15 of the movie), annotate this, and combine it with figure 1, to provide the reader early on with a sense of the general setup. These images are rather striking.

We have combined still frame ~2:15 of the movie with Figure 1 as a new panel. Thank you for this suggestion!

**4) The manuscript is largely focused on grounding, however, I'd argue that the subsequent ungrounding is also important. [As a side note: Reading the paper I was wondering whether ungrounding is primarily the result of melting (and potentially capsizing), or rather changes in ocean current/wind direction? This is not a focus of this work, but if you have any insight I'd be interested to hear it.]**

Winds seem to be the main driver of motion for kinetically grounded icebergs that remain embedded in the sediment (Figure 12).

We have added a new subsection 3.3 on iceberg melting and capsizing. We have also discussed capsizing once again in lines 846-858, while acting on your feedback.

In our study we have kept the old NEMO iceberg capsizing criterion based on the ratio between horizontal length and keel depth. Our companion manuscript, Olive Abello et al. (2025) use an updated capsizing criterion that compares width and thickness, and is consistent with the careful mathematical arguments laid out in Wagner et al. (2017 OM).

Simulated icebergs grounded on Bear Ridge experience faster lateral than basal melting. However, when the old capsizing criterion is active, the icebergs' width decreases to zero before its length can cross the critical threshold based on the *keel* thickness (Figure R1 in this response). In contrast, when the criterion in Olive Abello et al. (2025) is used, grounded icebergs do capsize (Figure R2, top). In that case, they unground (Figure R2, bottom) and resume flotation, only to become marginally grounded once again (Figure R2, bottom).

[Figure]

Figure R1. Example of a simulated iceberg that is grounded along Bear Ridge a month into the simulation and remains grounded. Using the old capsizing criterion comparing length and keel thickness, the iceberg melts away before it can capsize.

[Figure]

Figure R2. Example of a simulated iceberg that is grounded along Bear Ridge a month into the simulation. We have zoomed on particular months of the simulation. Top: using the Olive Abello (2025) capsizing criterion, the iceberg capsizes. Bottom: the scour depth reached by the iceberg keel at any point in time. After capsizing, the iceberg briefly resumes free flotation. However, its shallower keel quickly becomes marginally grounded.

**While I agree with the authors' choice to focus on the novel representation of the grounding process, I do think it would be helpful to also discuss ungrounding and the role of melting. Two things came to mind:**
We have added a new subsection 3.3 on iceberg melting and capsizing. We have also discussed capsizing once again in lines 846-858, while acting on your feedback.

**- First, as far as I know the melt model in NEMO-ICB contains a dependence of basal melt**

**on the relative velocity between the iceberg and the ocean current at the height of the iceberg base (Merino et al, 2016, eq 2). I wonder how this plays out for grounded icebergs in the latest version?**

Thank you for raising this question. We have added a clarification that in the new grounding formulation, the relative basal velocity is computed at "the deepest ocean level above the sediment" (line 373-374). This formulation may not be perfect, but it allows even the statically grounded icebergs which have no velocity of their own to melt at the base.

**Lines 871/872 make it sounds like this is not the case in the current model formulation?**

We have clarified in lines 890-894 that "the formulas in NEMO explicitly represent the direct effect of iceberg lengthscale on basal melting through a -0.2 power law, while the potentially large impact of horizontal size via changes in relative velocity is an emergent phenomenon."

**The dependence on the size of the iceberg is evident as well in the Merino et al. formulation. I'm likely missing something, but maybe this paragraph could be reworded and/or clarified?**

We have clarified in lines 890-894 that "the formulas in NEMO explicitly represent the direct effect of iceberg lengthscale on basal melting through a -0.2 power law, while the potentially large impact of horizontal size via changes in relative velocity is an emergent phenomenon."

Hence, we expect the inverse length scale to have a much stronger net impact on the melting rate than an estimate based solely on the -0.2 power law.

**- Second, freely floating icebergs typically erode much faster on the side walls due to wave erosion (~1 m/d) than the base (~0.1 m/d) - see, e.g., Wagner & Eisenman (GRL, 2017, https://doi.org/10.1002/2016GL071645). In that case you might expect that icebergs shrink laterally until the aspect ratio becomes unstable and they become ungrounded by capsizing (as the authors mention). This may be particularly relevant for the smaller icebergs found all over Bear Ridge. However, since for grounded icebergs the relative basal velocity is higher, maybe the thinning is substantially faster than for freely floating ones, which might entail that capsizing isn't that important after all. Would it be easy to check how often ungrounding in the model coincides with capsizing? I appreciate that a detailed analysis of these processes is beyond the scope of this study, but I do think it would be helpful to comment on how melt is represented in the model, and to at least mention some of the considerations above.**

Thank you for this comment. We have added a description of melting and capsizing in our new subsection 3.3 Representation of iceberg melting and capsizing in NEMO and their role in ungrounding. In addition, we have discussed results from Amundsen Sea simulations in the context of melting, capsizing, and grounding in lines 846-858:

"In order to explore the potential for capsizing, we have extended our four-year LONG simulation with THICK icebergs up to 20 years by applying a repeat cycle of the same four-year surface boundary conditions. We see that in our extended LONG simulation, the rate of lateral melting indeed exceeds the rate of basal melting, with a potentially unrealistically large contribution of wave erosion to the former. One may expect that this causes widespread capsizing of grounded icebergs. However, the pre-existing NEMO capsizing criterion incorrectly compares horizontal length rather than horizontal width against vertical thickness. As a result, the excessive lateral melting of individual grounded icebergs eventually reduces the horizontal

width to zero within 17 years and completely destroys the iceberg before the horizontal length decreases enough for the iceberg to capsize (not shown). Abello et al. (2025) correct the unphysical capsizing criterion in NEMO, a change that will be implemented in new model configurations. On the other hand, eliminating the relevant bias in the horizontal wave erosion remains an important outstanding issue left to future studies."

**5) It was my understanding that the original NEMO-ICB used the erroneous capsizing criterion of Bigg et al (2017). We published a correction to this in Wagner et al (Ocean Modeling, 2017, https://doi.org/10.1016/j.ocemod.2017.07.003) and I discussed this briefly with Bob Marsh back then but never followed up. I just want to make sure the capsizing errors have been fixed, if there ever were any.**

Compared to Wagner et al. (OM, 2017) and Bigg et al. (2017), the present capsizing criterion uses a different power law, namely:

SQRT( 0.92*(KeelDepth**2) + 58.32*KeelDepth ) )

This makes a direct comparison with Wagner et al. (OM, 2017) formulation more difficult.

However, it stands out that in the present NEMO-ICB version, the criterion is applied as a comparison between the iceberg horizontal length and keel depth. Wagner et al. (OM, 2017) suggests that the appropriate comparison should be between the horizontal width and the full thickness rather than the horizontal length and the keel depth. The full thickness swaps with the horizontal width when the iceberg rotates. Analysing and improving the capsizing is beyond the scope of our current manuscript. However, our companion manuscript Abello et al. (2025) introduces an updated rolling criterion in NEMO that is based on the ratio between horizontal width and the full thickness while taking into account the exact issues highlighted in Wagner et al. (OM, 2017).

As we say above, lines 846-858 now state:

"In order to explore the potential for capsizing, we have extended our four-year LONG simulation with THICK icebergs up to 20 years by applying a repeat cycle of the same four-year surface boundary conditions. We see that in our extended LONG simulation, the rate of lateral melting indeed exceeds the rate of basal melting, with a potentially unrealistically large contribution of wave erosion to the former. One may expect that this causes widespread capsizing of grounded icebergs. However, the pre-existing NEMO capsizing criterion incorrectly compares horizontal length rather than horizontal width against vertical thickness. As a result, the excessive lateral melting of individual grounded icebergs eventually reduces the horizontal width to zero within 17 years and completely destroys the iceberg before the horizontal length decreases enough for the iceberg to capsize (not shown). Abello et al. (2025) correct the unphysical capsizing criterion in NEMO, a change that will be implemented in new model configurations. On the other hand, eliminating the relevant bias in the horizontal wave erosion remains an important outstanding issue left to future studies."

**Specific comments:**

**A number of mostly minor and technical comments are provided as annotations to the attached pdf.**

Thank you for your detailed feedback. We have responded to these comments line by line below and implemented your suggestions in the text.

**Dear authors - I am often wrong, and if you think that any of my comments are misguided please reach out to me and I'll be eager to amend my review.**

**Till Wagner**

**Line 10: not sure I'd call it ubiquitious - maybe "commonly observed"?**

Thank you, we have replaced it with "commonly observed" (Line 10).

**Lines 12-13, remove text: Strikethrough text**

Thank you, we have shortened the text as suggested (Lines 12-13).

**Line 27: just note that this is given as "2025" elsewhere.**

Thank you, we have changed "in prep." to 2025 (Line 26).

**Lines 30-39: This paragraph could be shortened to 1 or 2 sentences?**

Lines 33-42: Thank you for this suggestion, but we think that it is important to refer the reader to broadly relevant previous literature on icebergs. This entire paragraph is composed of citations of previous publications that we prefer not to remove from our reference list.

**Lines 31-32: see also: Duprat, L., Bigg, G. & Wilton, D. Enhanced Southern Ocean marine productivity due to fertilization by giant icebergs. Nature Geosci 9, 219–221 (2016). https://doi.org/10.1038/ngeo2633**

Thank you, we have cited Duprat et al. (2016) here (Line 41).

**Line 43: there is also a body of literature that looks at iceberg scouring to shed light on paleo processes. 2 examples: Hill, J., Condron, A. Subtropical iceberg scours and meltwater routing in the deglacial western North Atlantic. Nature Geosci 7, 806–810 (2014). https://doi.org/10.1038/ngeo2267 and Starr, A., Hall, I.R., Barker, S. et al. Antarctic icebergs reorganize ocean circulation during Pleistocene glacials. Nature 589, 236–241 (2021). https://doi.org/10.1038/s41586-020-03094-7**

Thank you, we have cited the suggested literature on iceberg scours in a paleoclimate context (Lines 46-48).

**Line 44: just a note that, relatedly, Stern et al (2015) https://doi.org/10.1002/2015JC010805 found that grounded icebergs can act much like islands, in the sense that they can cause upwelling on the downwind side and increased stratification on the upwind side. Which made me wonder - how much of the differences in sea ice cover between east and west of Bear Ridge is due to icebergs mechanically blocking the sea ice advection and how much is due to the icebergs causing different ocean conditions on the two sides? Also, I found it interesting that these differences in turn impact the melt process of the iceberg itself (disclaimer - I'm an n-th author of that paper).**

Thank you for pointing us to this reference. We have cited it here (Line 50) in the context of modified ocean conditions sustained by grounded icebergs. We avoid speculating whether such a downwelling effect on the eastern side of Bear Ridge contributes significantly to the sea-ice dipole. Bett et al. (2020) found that representing the grounded bergs as a thin ice shelf had the same effect on sea-ice as representing them as land. This may suggest that the impact of the altered ocean conditions on sea-ice is small compared to the direct blocking of sea-ice advection. We now point this out more clearly in Lines 53-55.

**Line 62: maybe better in quotation marks? "wall of icebergs"**

Lines 36, 69, 451: Yes, we have added quotation marks, as you suggest.

**Lines 65-67: I would argue that there were important efforts quite a bit earlier - e.g., Mountain (1980) https://doi.org/10.1016/0165-232X(80)90055-5 and Smith and Banke (1983) https://doi.org/10.1016/0165-232X(83)90045-9**

**Maybe it would be more accurate to say that most currently used iceberg models can be traced back to Bigg et al (1997) and Gladstone et al (2001).**

Thank you for pointing us to these references. We have cited them here (lines 74-76) and rephrased the statement about Bigg et al. (1997) and Gladstone et al. (2001) in lines 74-76.

Interestingly, Mountain (1980) actually assumes that Ekman currents play an important role in driving iceberg motion. So we have also cited this in our Section 4.1 (line 555-556) when discussing the ageostrophic contribution to the ocean dynamics around icebergs.

**Lines 99-106: paragraph could maybe be shortened**

Lines 109-116: Yes, we have made this paragraph more concise, as you suggested.

**Line 112: Shoal**

Line 123: Thank you. We have replaced "shallow" with "shoal," as you suggest.

**Line 132: in my experience "capsizing" is used more typically in the literature**

Lines 144, 157: Thank you. We have replaced "toppled" with "capsized," as you suggest.

**Line 132: note that the standard capsizing criterion that was used by Bigg et al (1997) and most subsequent iceberg modeling studies dated back to Weeks and Mellor (1978) https://doi.org/10.1016/B978-0-08-022916-4.50015-7 and featured some errors that led to icebergs continually capsizing among other things (see also my general comment 5).**

A direct comparison between the existing NEMO capsizing criteria and the Wagner et al. (OM, 2017) formulation is difficult because they follow different power laws. However, it stands out that in the present NEMO-ICB version, the criterion is applied as a comparison between the iceberg horizontal length and keel depth. Wagner et al. (OM, 2017) suggests that the appropriate comparison should be between the width and the full thickness rather than the length and the keel depth. The full thickness swaps with the horizontal width when the iceberg rotates. Analysing and improving the capsizing is beyond the scope of our current manuscript. However, our companion manuscript Olive Abello et al. (2025) introduce an updated rolling criterion in NEMO that is based on the ratio between horizontal width and the full thickness while taking into account the exact issues highlighted in Wagner et al. (OM, 2017). We have already cited Olive Abello et al. (2025), but we also refer to it when discussing the capsizing criterion and its update in NEMO (Lines 362-382 and 846-858).

**Figure 2: Instead of using a schematic for panel (c), could you take an actual representative cross-section example - and you could mark that cross-section on the map of panel a or b?**

Figure 2: Thank you for this suggestion, which we have carefully considered. However, individual sections do not necessarily provide a clearer picture compared to this schematic, which is very representative of typical scour shapes.

**Line 155: could you mention in this paragraph briefly why you're interested in WBD and shear strength - I was wondering about it until I got quite a bit further down.**

Thank you, yes, we now mention that WBD and shear strength determine the sediment resistance forces acting on grounded icebergs (Line 168).

**Line 168: Strikethrough text**

Thank you for suggesting this correction. We have removed the "s." (Line 184).

**Figure 3: It's not clear to me exactly how the partition of the 3 layers plays out in this plot - maybe you could color the markers accordingly or draw approximate boxes around them.**

**I was also wondering whether different colors (or using different markers) for different cores may be insightful?**

Figure 3: Yes, we have made sure that the three layers are differentiated in this figure by drawing boxes around the data markers while using distinct color and line types for the box contours.

**Lines 200-201: are these two sentences needed?**

In order to make the text more concise, we have removed these sentences (Line 216).

**Line 204: Strikethrough text**

Line 219 Thank you for pointing out this typo. We have corrected it.

**Line 204: And**

Line 219: Thank you for pointing out this typo. We have corrected it.

**Line 208: Martin and Adcroft (2010) distinguished between form drag and skin drag. How does NEMO-ICB deal with this?**

**[As a sidenote: we looked in some detail at the relative importance of form vs skin drag and found that this varies substantially depending on whether you have a high length-to-height ratio or one that is O(1) - take a look if you're interested: https://doi.org/10.1175/JPO-D-20-0275.1**

**See also a further comment on C_drag below.**

Lines 223 and 543-544: Here we have clarified that NEMO-ICB assumes form drag, and we have cited Martin and Adcroft (2010), as well as Wagner et al. (2022).

**Line 239: Maybe a short paragraph on the melt representation in NEMO-ICB following here?**

Line 362: You suggest adding a short paragraph on the melt representation here. We think this might be more appropriate after Line 362, and we have added it there, as a new subsection 3.3.

**Lines 241-243: Maybe it's worth mentioning that you envisage the icebergs to plough a triangular (v-shaped) trench into the sediment, however, the motion of the iceberg is computed using a perfectly rectangular iceberg. (I've always assumed that these**

**icebergs are approximately flat at the bottom, which makes me think: how do they leave v-shaped scours? Am I missing something obvious?)**

Thank you! Yes, we have mentioned that guided by observations, we assume that iceberg keels plough v-shaped trenches into the sediment (Lines 260-262). Even tabular icebergs are not flat at the bottom but rough (Lines 139-140), typically with a v-shaped protrusion. We state that our assumption about the keel shape is based on the indirect evidence provided by iceberg scours (Lines 260-262). In addition, in line 871 we explain that future radar sounding observations may reveal the distribution of iceberg keel shapes.

**Line 256 and Section 2.2: is this the same as WBD ? maybe clarify ?**

Lines 168, 170, 188, 292 in Sections 2.2 and 3.2: Yes, thank you for pointing out the inconsistent use of terminology which we have fixed. We assume fully saturated sediment, so in our case WBD is the same as saturated density. We now use consistent terminology in these lines.

**Line 267, Eq. 11: could move this term in front of the curly bracket, since it's the same for the two cases?**

Line 283, Eq. 11: We have moved the common factor outside the curly brackets, as you suggest.

**Lines 274-277: maybe this should be moved to near line 170 ?**

Yes, as you suggest, we have moved these sentences to lines 173-175.

**Line 299: you provide a reference and more discussion for this later- I would move it up here and shorten**

Thank you! We have moved the sentences and the Veldhuijsen et al. reference to Lines 313-315 of the new text.

**Line 321: Strikethrough text**

We have fixed the mistyped sentence, as you suggest (Line 331).

**Line 321: of motion**

We have fixed the mistyped sentence, as you suggest (Line 331).

**Line 321: Strikethrough text**

We have fixed the mistyped sentence, as you suggest (Line 331).

**Line 321: capped?**

Line 334: Yes, we have rephrased "limited" to "capped."

**Lines 329-330: maybe discuss briefly here what range you tested and that the results are fairly insensitive to this exact value. Looking at Appendix B it seems to me that the effect of mu is saturated once you exceed 0.002, and the main differences arise somewhere in the range mu = [0,0.002]. Did you look at smaller values as well?**

Lines 336-342: We have referred to the Appendix B results here in more detail. We state that we have done tests with values higher and lower than 0.002, including a case with no Coulomb friction that still includes sediment resistance and gravity. We point the reader to Appendix B.

**Line 339 and equation 18: d**

Line 333 and equation 15: Thank you! We have fixed the typo.

**Line 341: force**

Line 353: Thank you! We have added the missing word "force," as you point out.

**Lines 344-361: I wonder whether this paragraph could be cut/shortened/moved to the appendix?**

Lines 356-361: Thank you for this suggestion, we now briefly mention the curvature term here and elaborate only in the Appendix B.

**Lines 407-419: This paragraph could maybe be summarized in a sentence or two?**

Lines 423-431: Thank you for this suggestion. We have shortened this paragraph.

**Lines 449-451: move this density discussion up (see earlier comment).**

Yes, we have moved this discussion and the Veldhuijsen et al. reference to Lines 313-315 of the new text.

**Line 471: maybe mention here somewhere how melt and ungrounding happens (+capsizing?) - see my general comment 4.**

Lines 488-489: Thank you for this suggestion. Here we point out that complete un-grounding and the resumption of free flotation can be due to the melting process and compare the relative roles of basal melting versus lateral melting and capsizing.

**Section 4.1: The following analysis is a really interesting step forward! I've always felt a little uneasy about our approximation ignoring the ageostrophic currents.**

**I will say that I struggled a bit to follow the argument of the next few pages in detail, I guess at least in part because I'm used to thinking of it in terms of the force balance equations, rather than just the accelerations. And since I think of the drags, for example, as "surface forces" acting on cross-sectional areas and Coriolis and gravity as "body forces" acting on the mass of the iceberg, I found myself doing some cognitive aerobics reconciling the two frameworks. I am not suggesting you recast the whole argument in terms of forces, but maybe you can go over the text once more with a critical eye to make it as clear as possible?**

Section 4.1: Thank you for your feedback! We have rephrased this part of the text to make it clearer. For example, when we discuss the atmospheric terms in the momentum budget, we point out that they are an external source of iceberg acceleration (Lines 586-587). The question then becomes what balances these external forces (Lines 586-587). When discussing the drag terms, we specify that they are form drag (lines 543-544, 599, 610). When discussing actual values of the drag terms, we point out that they are direct output from the model (Line 613). When referring to the geostrophic component of ambient ocean flow, we suggest that it is indeed a "dominant" component (Line 664) without implying that Ekman transport is negligible.

**Line 542, Eq. 22: this reminds me of the discussion in Bigg et al (1996) - see their eq (4)**

Line 563, Eq. 19: Thank you for pointing this out. We have cited Bigg et al. (1996) elsewhere, and we also cite that reference here. We point the reader to eq (4) in Bigg et al. (1996).

**Lines 590-599: this may speak to my confusio (see comment above) - how exactly is C_drag obtained here?**

Lines 598-600: Thank you for pointing out the need for greater clarity! We now specify that this is form drag (reiterating this in lines 543-544), which is calculated as a function of the fluid density and iceberg geometry. And in lines 613-614, we specify that the values cited here are output directly by the model during the respective simulations.

**Lines 590-599: following up on an earlier comment, it appears that C_drag represents the form drag only? (For skin drag in the limit of thin icebergs the atmospheric and oceanic drag coefficients are comparable - see Wagner et al (2022)). Might be helpful to clarify what C_drag represents.**

Lines 598-600: Thank you once again for your feedback! We have clarified that this is form drag (reiterating this in lines 543-544), which is calculated as a function of the fluid density and iceberg geometry. We cite Wagner et al (2022). We specify that the values cited here are output directly by the model during the respective simulations.

**Lines 710-711: this is largely a repeat of l 696/697. Also, maybe make it clear when you first introduce MEDIUM icebergs that these do not feature in the grounding discussion?**

Line 731: We agree and remove these lines because re-stating this information here is redundant to previous lines. Moreover, as you suggest, we point out earlier, in Lines 467-469, that MEDIUM icebergs do not feature in the grounding analysis.

**Line 754: Strikethrough text**

Line 773: Thank you! We have moved the quotation mark outside the word "kinetic".

**Line 754: "**

Line 773: Thank you! We have moved the quotation mark outside the word "kinetic".

**Line 766: Strikethrough text**

Line 785: We have moved the word "only," as you suggest and deleted the phrase "the course of."

**Line 766: Strikethrough text**

Line 785: We have moved the word "only," as you suggest and deleted the phrase "the course of."

**Line 766: only**

Line 785: We have moved the word "only," as you suggest and deleted the phrase "the course of."

**Line 785, Figure 13 caption: cut?**

Figure 13 caption: We deleted the mistyped word "and," as you suggest.

**Lines 797-798: something went wrong here - revise?**

Lines 813-815: Yes, we apologize for accidentally pasting the unnecessary words at the end of the sentence. We have deleted them.

**Lines 797-798: Strikethrough text**

Lines 814-815: Yes, we apologize for accidentally pasting the unnecessary words at the end of the sentence. We have deleted them.

**Lines 812-813: Maybe this can be combined with the first discussion of mu on l 329? See also my comment there.**

Thank you, we have indeed moved that to the earlier discussion of mu in Lines 338-342.

**Figure 15: I'm missing something here, sorry - how do you have acceleration in the case of "statically grounded" ? Could this be rephrased?**

Figure 15 and Lines 831-836: We accept your suggestion. In the case of statically grounded icebergs, we rephrase "acceleration" to "force per unit mass."

**Lines 854-896 on page 32: I feel like this page could be shortened?**

We have now made the whole Summary and Conclusions section more concise. At the same time, on your request (Revewer #2), we had to expand the discussion of melting in NEMO and its dependence on iceberg lengthscale.

**Lines 867-869: note that this was studied (and indeed found to be the case) in England et al (2022) https://doi.org/10.1126/sciadv.abd1273 (disclaimer: I'm a co-author) and breakup was implemented in the GFDL model in Huth et al (2023) https://doi.org/10.1029/2021MS002869**

**In my experience, breakup is a first-order process for large icebergs. I'd be excited to talk more if you're interested in adding something like this to NEMO ?**

Lines 885-886: Thank you for pointing us to these references, which we cite here. It is noteworthy that England et al (2022) demonstrate the potential for iceberg fragments to move at a different angle relative to parent icebergs. In England et al. (2022) the Wagner et al. (2017) analytical model is directly imposed on icebergs. At the same time, it would be interesting if a future study explores whether this motion of iceberg fragments appears as an emergent phenomenon when all model forces are applied to the iceberg momentum budget. There is also indeed potential room to improve the iceberg representation in NEMO by introducing fragmentation. WE have cited Huth et al. (2022) in lines 892-894.

**Lines 872-873: I think it does, and I think the NEMO model accounts for that (?) see my comment earlier.**

Lines 890-892: Yes, we include a clarification that the melting in NEMO is indeed a function of the relative velocity and the geometry. Our results imply that the iceberg lengthscale impacts the rate of melting not only by changing the geometry of the icebergs but via an impact on the relative velocity. This impact on melting via the relative velocity is an interesting new avenue for exploration in a future study, especially if it naturally emerges from the full set of dynamical and thermodynamical forcing factors applied to simulated icebergs.

**Line 1134: Strikethrough text**

Line 934: Thank you. We have deleted the typo.

---

## Author Response (AR2)

Response to the editor's comments by Yavor Kostov and co-authors.

Dear Dr Kostov and co-authors,

Thank you very much for your considered editing of the manuscript in response to the reviewers' comments. The revised manuscript will be suitable for publication in the Cryosphere. However, I would ask you to please consider implementing the below minor changes to help with readability. Once these are addressed, I would be happy to recommend for publication.

Best regards,
Felicity McCormack

Dear Dr McCormack,

We thank you for patiently reviewing our manuscript and for recommending the Minor Revisions with detailed suggestions. We have implemented everything that you outline, and we respond to each of your comments below.

Best regards,

Yavor Kostov and co-authors
* * *
- L29: perhaps expand this sentence to describe what is the problem with representing the icebergs as point particles

Thank you for this suggestion. We have expanded the sentence in Line 29 with the text "particularly challenging, especially when they are represented as point particles because this ignores the impact of spatial variability in the ocean properties along the horizontal and vertical extent of the keels, as well as the interaction with bottom topography."

- L29-L32: slight editing of language to improve clarity would be good here

Thank you for this comment. In Lines 30-34, we now say, "The modelling approach of Stern et al. (2017) does capture the effect of the ocean and the atmosphere over the entire breadth and depth of large icebergs. Similarly, Martin and Adcroft (2010), as well as Merino et al. (2016), partially mitigate the issue of representing icebergs in NEMO as point particles by imposing that the latter respond to spatially averaged ambient properties, and that iceberg keels are aware of bottom topography."

- L74-81: Here, it would be helpful to add a sentence or two at the end of this paragraph describing what is the problem with current approaches to modelling icebergs. I.e. what aspects of the previous attempts are lacking? What don't we represent well, particularly w.r.t. Iceberg grounding / ungrounding? That then motivates the following paragraph that talks to what the approach of this paper is; i.e. how it addresses the knowledge gap.

We are grateful for this helpful comment. We have described the knowledge gap, as you suggest. In Lines 83-88, we now say, "Since the developments of Merino et al. (2016), the keels of simulated icebergs interact with shallow bottom topography as an obstacle. However, icebergs in NEMO are unable to ground realistically and cannot remain trapped in the sediment. They stop and bounce off topographic obstacles instead. In turn, this has prevented the further development of algorithms simulating the blocking of sea-ice by grounded icebergs. The present work, which updates iceberg grounding in the model, is therefore a stepping-stone towards a future improvement of the interaction between icebergs and sea-ice in NEMO." We think that this highlights what has been lacking and motivates the following paragraph, as you suggest.

- Signposting around section 2. R2 raised the idea to separate this manuscript into 2 papers: one on grounding and one on drift dynamics. I agree with the authors that it's better to keep these aspects together, but I would recommend some more signposting that helps the reader to understand why certain ideas are introduced and how they relate to the aims of the paper. Section 2 in particular should include some more signposting, as per my below comments:

Thank you for your detailed suggestions. We have added the signposting that you request, and we address each of your comments below.

-- L103: Here, it would be helpful to motivate why you include section 2 with a couple of sentences. I.e. section 2 provides an overview of the dynamics related to iceberg grounding (from observations) that are essential to understand if we are to model them correctly (as per section 3).

Thank you! In lines 111-112, we now say, "Section 2 thus provides an overview of the relevant real-world conditions, which are essential to understand and model iceberg grounding as per Section 3."

-- Sections 2.1-2.3. Can you start these sections with a couple of sentences that describe why we need to understand these processes, and how neglect of these processes in previous attempts to model icebergs impacts the accuracy / reliability of the modelling?

We appreciate this suggestion. We have added introductory sentences to the beginnings of these sections (Lines 118-124 and 179-181) to explain why we need an understanding of these properties and how that impacts the model's ability to simulate grounded icebergs and their residence times.

-- L125:128: the sentences "Here, we manually map more than 60 scours from existing... calibrate the modelled scouring in Section 4." could be moved to the start of the section after a sentence on why iceberg scours are important to map. E.g. "Iceberg scours are a ubiquitous

feature of the Amundsen Sea sector, providing insight into the dynamics and history of iceberg grounding in the past. In this section, we manually map more than 60 scours..." Then proceed to describe where iceberg scours are found, what they tell us about the dynamics, and their morphology.

Thank you! We have indeed made these changes and the rearrangement that you propose. Lines 118-124 now read, "Iceberg scours (also called ploughmarks) are widespread in two Amundsen Sea regions and imply that thick icebergs can remain embedded in the bottom sediment for extended periods of time, a feature absent from previous versions of NEMO. Sediment scours also provide indirect evidence for the shape and size of the iceberg keels that interact with bottom topography, and this information is needed for the proper simulation of grounding in models. Here, we manually map more than 60 scours from existing multibeam-bathymetric data on the eastern flank and central part of Bear Ridge as a representative population of modern scours. We describe their morphology qualitatively with the aim of characterising modern iceberg grounding events, and then we provide metrics on their dimensions to calibrate the modelled scouring in Section 4."

-- L168-172: As for section 2.1, modifying the first couple of sentences of section 2.2 to be clear on why sediment density and strength is important and the aim of this section would be good.

Following your helpful suggestion, we have indeed added new sentences and modified existing ones in the beginning of Section 2.2 (Lines 179-185) to highlight the aim of the section and the importance of these sediment properties, "In order to simulate the residence time of icebergs embedded in the bottom sediment, a model needs realistic values for the saturated density and shear strength, properties which determine resistance forces acting on grounded icebergs. In this section we present observations of saturated density and shear strength characteristic of the Amundsen Sea. Data from sediment cores recovered from the basin (Figure 3; Smith et al., 2011; Clark et al., 2024) provide a range of values for both the saturated density and shear strength."

- L180-181: remove second "to" → "...at or close to (<25 km) to the ice sheet..."

Done (Line 194).

- L380-382: recommend to add clarification that the new rolling criteria detailed in Olivé Abelló et al. (2025) is not also employed in this manuscript

Thank you for pointing this out. We have added a clarification in lines 396-400: "Our work and the Olivé Abelló et al. (2025) companion study test each of the new updates to the NEMO iceberg algorithms separately. Our study introduces the new representation of grounding but relies on the old rolling criterion. In comparison, Olivé Abelló et al. (2025) use the new rolling criterion combined with the old grounding algorithms. These coordinated efforts will contribute to the improved representation of icebergs in future versions of NEMO."

**Editor decision: Publish subject to minor revisions (review by editor)** by Felicity McCormack

**Public justification (visible to the public if the article is accepted and published)**:
Dear Dr Kostov and co-authors,
Apologies for the delay in my response.
Based on the reviewers' comments and recommendations, I agree that this manuscript is suitable for publication pending minor revisions. Please ensure that all reviewer feedback, as well as the points raised in my earlier response, are fully addressed when you upload your revised manuscript.

Best regards,
Felicity McCormack

Dear Dr McCormack,

We once again thank you for your time and effort. We appreciate all suggestions and have fully implemented them, as indicated in our line-by-line response above.

Kind regards,

Yavor Kostov and co-authors

---

## Author Response (AR3)

Dear Dr McCormack,

We thank you for your positive decision and for your guidance during the peer review process. We are also grateful to the reviewers for their helpful feedback.

We are submitting the accepted version of the manuscript with formatted references, consistent unit notation, and figure panel labels that comply with the Copernicus requirements. No further changes have been made.

Kind regards,

Yavor Kostov

On behalf of all co-authors